# EXPLORING MULTI-GRAINED CONCEPT ANNOTATIONS FOR MULTIMODAL LARGE LANGUAGE MODELS

## ABSTRACT

Multimodal Large Language Models (MLLMs) excel in vision–language tasks by pre-training solely on coarse-grained concept annotations (*e.g.*, image captions). We hypothesize that integrating fine-grained concept annotations (*e.g.*, object labels and object regions) will further improve performance, as both data granularities complement each other in terms of breadth and depth in concept representation.

We introduce a new dataset featuring **M**ultimodal **M**ulti-**G**ra**i**ned **C**oncept annotations (**MMGiC**) for MLLMs. In constructing MMGiC, we explore the impact of different data recipes on multimodal comprehension and generation. Our analyses reveal that multi-grained concept annotations integrate and complement each other, under our structured template and autoregressive discrete framework.

We definitively show that multi-grained concepts do facilitate MLLMs to better locate and learn concepts, aligning vision and language at multiple granularities. We further validate our hypothesis by investigating the comparison and collaboration between MMGiC and image–caption data on 12 multimodal comprehension and generation benchmarks, *e.g.*, their appropriate combination achieve 3.95% and 2.34% accuracy improvements on POPE and SEED-Bench. Code, data and models will be made openly available.

## 1 INTRODUCTION

With the rapid development of Large Language Models (LLMs) (Brown et al., 2020; Chowdhery et al., 2023; Touvron et al., 2023a;b) and Visual Foundation Models (VFMs) (Radford et al., 2021; Rombach et al., 2022; Dehghani et al., 2023), researchers have started to explore the potential of unifying them into Multimodal Large Language Models (MLLMs) to perform various Vision–Language (VL) tasks, such as image captioning, visual question answering, and text-to-image generation. MLLMs, such as Emu (Sun et al., 2023c), SEED-LLaMA (Ge et al., 2023b) and LaVIT (Jin et al., 2023), follow a similar framework that integrates the capabilities of LLMs and VFMs under an autoregressive objective of predicting the next visual or textual token, showing impressive performance on VL tasks.

Despite their success, existing MLLMs typically do not make full use of *concepts* in VL learning, relying on coarse-grained concept annotations (*e.g.*, image captions) but ignoring fine-grained concept annotations (*e.g.*, object labels and object regions). This leads to superficial and incomplete understanding of concepts, limiting VL alignment. Specifically, by "concept" we mean an abstraction and generalization of a group of things having common characteristics (Goguen, 1969; Carey, 2000; Blouw et al., 2016). Concepts can be categorized into concrete and abstract concepts by whether they can be sensed by five human senses (Shevade et al., 2005; Connell et al., 2018). Concrete concepts, such as objects, attributes, and relationships, are not only easy to collect and annotate, but also semantically consistent when expressed across modalities (Chen et al., 2019; Xu et al., 2020; Xie et al., 2020). Hence, many traditional Vision–Language Models (VLMs) combine coarse- and fine-grained concept annotations to better learn cross-modal consistent concrete concepts, thus improving VL alignment (Li et al., 2020; Zeng et al., 2021; Shen et al., 2022; Menon & Vondrick, 2023). However, they rely on additional components and loss functions to leverage different-grained concept annotations (*e.g.*, bounding box prediction), and optimize the multimodal comprehension ability of VLMs at different granularities **separately** through multitask learning. Moreover, they **seldom** explore the potential of multi-grained concept annotations in multimodal generation tasks, such as text-to-image generation, let alone exploring both multimodal comprehension and generation tasks under the same framework.

As prior work demonstrate that concepts are crucial for VL alignment and combining coarse- and fine-grained concept annotations can better learn concepts, we argue that existing MLLMs should make better use of concepts by incorporating multi-grained concept annotations into their training. To this end, we first merge four public, object detection datasets to construct a new multimodal dataset, MMGIC. Our goal is not to replace or surpass existing image–caption datasets, but to address the lack of datasets with multi-grained concept annotations for MLLMs and explore their potential. Different from previous VLM work, we 1) provide multimodal annotations for images, including both textual forms (caption, labels and label descriptions) and visual form (object regions), to **enrich** multi-grained concept annotations; 2) design a structured template to integrate multimodal multi-grained concept annotations into image–text interleaved documents, to **leverage** the complex context processing capability of MLLMs; 3) instead of additional components or loss functions used in VLMs, train MLLMs with an autoregressive discrete framework and predict the next visual or textual token in a multimodal discrete token sequence, to **improve** multimodal comprehension and generation ability **across** multiple granularities **simultaneously**. This can reuse existing LLM training regimes and ensure the generality and applicability of MMGIC to different MLLM frameworks. The key contributions are as follows:

- We introduce MMGIC, a new dataset with multimodal multi-grained concept annotations (Sec. 2). Under a general MLLM framework (Sec. 3), we show that MMGIC can help MLLMs better locate and learn concepts, thus aligning vision and language across multiple granularities (Sec. 4).

- We explore different data recipes for multi-grained concept annotations (Sec. 4.1 & 4.4). Our analyses show that multi-grained annotations can integrate and complement each other to help MLLMs ground concepts in the textual annotations to corresponding regions in images, thus improving the ability to understand and generate concepts.

- Through the evaluation of 12 benchmarks for multimodal comprehension and generation in both pre-training (Sec. 4.2) and supervised fine-tuning stages (Sec. 4.3), we explore the comparison and collaboration between MMGIC and coarse-grained image–caption data. We find that they each have their own strengths in depth and breadth of concept representation, and that appropriate curriculum learning strategies can effectively combine their strengths to further improve performance.

## 2 MMGIC DATASET

To be clear, our goal is not to supplant existing image–caption datasets, but to build a multimodal dataset with multi-grained concept annotations to address the lack of such datasets, and then explore its potential in MLLMs. We now introduce its collection, pre-processing, complement and construction.

### 2.1 COLLECTION AND PRE-PROCESSING

In this work, we focus on concrete concepts, especially objects, attributes of objects, and relationships between objects. They are fundamental elements in VL learning and widely annotated in object detection datasets. Therefore, we collect four public large-scale human-annotated object detection datasets, including Open Images (Kuznetsova et al., 2020), Objects365 (Shao et al., 2019), V3Det (Wang et al., 2023a), and Visual Genome (Krishna et al., 2017). Images in these datasets are uploaded to Flickr by real-world users and collected by dataset providers. They typically show complex scenes with multiple objects, and are annotated with fine-grained category labels and object bounding boxes. Comparing with widely-used coarse-grained image–caption datasets, fine-grained object annotations provided in these datasets can help MLLMs locate and learn concepts in images.

**Object Annotation Pre-processing.** Fine-grained object annotations includes category labels and bounding box coordinates for each object region. To accommodate varying aspect ratios of bounding boxes and the requirement for a square input image, we crop a new larger square region $S_i$ that contains the original object region $R_i$, $R_i \subseteq S_i$, with their centers aligned as closely as possible. We then update the annotations of $S_i$ by integrating the category label of $R_i$ with the category labels of surrounding object regions. Notably, instead of transforming bounding box coordinates into tokens in the text (Chen et al., 2021; Liu et al., 2023c; Peng et al., 2023) or visual markers in the image (Shtedritski et al., 2023; Yang et al., 2023; Yao et al., 2024), for each object, MMGIC directly provides visual tokens of the cropped region $S_i$ and textual tokens of fine-grained category labels and location descriptions (Figure 1 ③ ). Fine-grained cropped regions can help MLLMs locate and align concepts in images and in annotations at a detailed level.

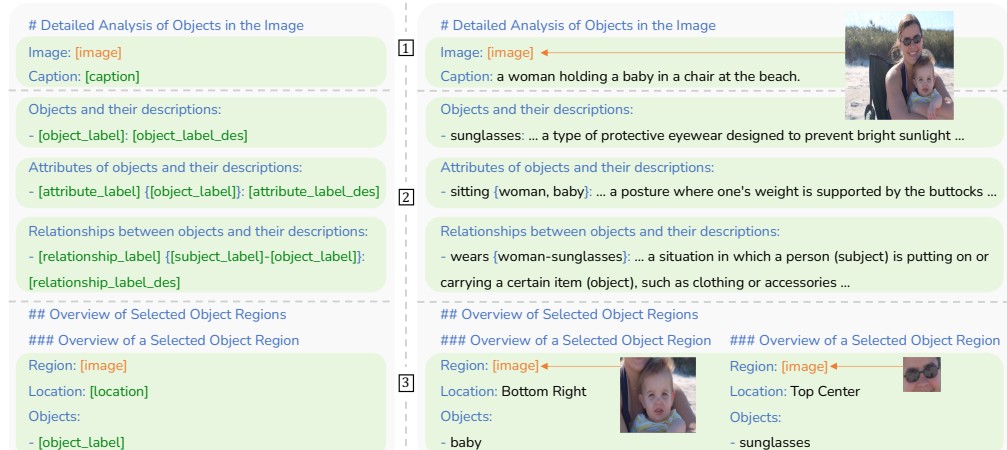

Figure 1: Structured template *(Left)* and data example *(Right)* of MMGiC. Different colored text indicates template text, image placeholders, annotation placeholders and multi-grained concept annotations, respectively. Each image–text interleaved data sample will be tokenized into discrete tokens.

## 2.2 MULTI-GRAINED CONCEPT ANNOTATION COMPLEMENT

We follow LAION-COCO (Schuhmann et al., 2023) to automatically synthesize captions for all images (Figure 1 ☐1 ) by BLIP-2 (Li et al., 2023b) and CLIP (Radford et al., 2021), since partial images are not annotated with captions. We do not use or synthesize captions for object regions to avoid introducing hallucinations.

**Label Description Generation.** Label descriptions are corresponding concept descriptions of concrete concepts in the image, which convey understanding about a concept by visually observed details and relevant knowledge. Previous works (Shen et al., 2022; Yao et al., 2022; Menon & Vondrick, 2023) successfully improve concept understanding by introducing label descriptions from WordNet (Miller, 1992) and LLMs (Brown et al., 2020). Inspired by them, we design prompt templates and several human-annotated examples for each type of category labels (object, object attribute, and relationship between objects), and then generate label-description pairs (Figure 1 ☐2 ) by GPT-4 (Achiam et al., 2023). We manually check them to ensure quality. Moreover, for better differentiation of polysemous category labels, we manually check them based on the definitions in WordNet, and update them based on the specific data samples, *e.g.*, "batter" → "batter (ballplayer)" or "batter (cooking)".

## 2.3 CONSTRUCTION

Above steps transform four object detection datasets into MMGiC, a multimodal dataset with multi-grained concept annotations. Furthermore, we carefully design a structured template to integrate multi-grained concept annotations into an image–text interleaved document. As shown in Figure 1 *(Left)*, the structured template consists of: ☐1 coarse-grained image-annotation part: each image is annotated with a short and general description of the whole image; ☐2 fine-grained image-annotation part: concrete concepts (objects, attributes and relationships) present in the image are annotated with corresponding category labels and label descriptions; ☐3 fine-grained object-annotation part: each object in the image is annotated with a cropped object region, object labels in the region, and a location description. A data example of MMGiC is shown in Figure 1 *(Right)*.

Different from previous VLM work that provide multiple **isolated** annotations for each image, MMGiC provides richer multimodal multi-grained concept annotations in both textual forms (caption, labels and label descriptions) and visual form (object regions) for each image, and integrates them into an image–text interleaved document by our structured template. This can **leverage** MLLMs' complex context processing capability to facilitate VL alignment **across** multiple granularities **simultaneously** under our MLLM framework. In a nutshell, MMGiC fills the gap in the MLLM field for datasets with multi-grained concept annotations. It contains 3.5M unique images, 23.9M unique object

regions, and 61.8M category label–description pairs. Based on MMGIC, we explore and analyse different data recipes for multi-grained concept annotations (Sec. 4.1 & 4.4), and further compare MMGIC with image–caption data (Sec. 4.2 & 4.3). More data details are shown in Appendix F.

# 3 FRAMEWORK

We introduce a general MLLM framework and its two training stages. Our goal is not to develop new frameworks, training objectives or benchmark SOTAs, but to explore the potential of multi-grained concept annotations for MLLMs under the general framework.

## 3.1 AN AUTOREGRESSIVE DISCRETE MLLM FRAMEWORK

Based on previous works (Ge et al., 2023b; Jin et al., 2023), we standardize a framework consisting of several visual modules and a LLM with an extended VL vocabulary (Fig. 2). It is trained with an autoregressive objective to generate predictions of the next token in a discrete sequence of image–text interleaved tokens, and can support our exploration in multimodal comprehension and generation.

**Visual Modules.** Inherited from LaVIT (Jin et al., 2023), the visual modules consist of a visual encoder, a visual tokenizer, a visual decoder and a diffusion model. The visual encoder is a pre-trained vision transformer (Dosovitskiy et al., 2020; Sun et al., 2023b), which encodes an image into a sequence of visual embeddings. The visual tokenizer quantizes these embeddings into a sequence of discrete visual tokens by a visual codebook (van den Oord et al., 2017). The visual decoder reconstructs predicted visual tokens into a sequence of visual embeddings, which are then taken as the condition of the diffusion model (Sohl-Dickstein et al., 2015; Ho et al., 2020; Rombach et al., 2022) to progressively generate target image pixels from a Gaussian noise. Overall, visual modules can tokenize an image into a sequence of discrete visual tokens as input and decode predicted visual tokens back into an image.

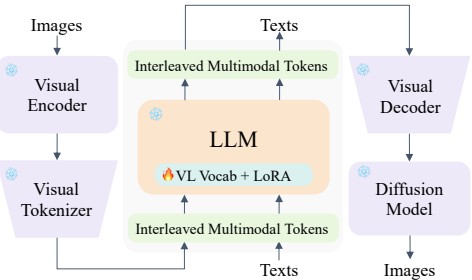

Figure 2: Illustration of our general MLLM framework. Only the LLM are loaded and partially fine-tuned during training.

**LLM with an Extended VL Vocabulary.** The LLM is inherited from LLaMA-2-7B (Touvron et al., 2023b), and its vocabulary is extended to support both textual and visual tokens. Since visual tokens in the VL vocabulary correspond one-to-one with visual latent codes in the visual codebook, instead of initializing visual token embeddings with the distribution of original textual token embeddings (Ge et al., 2023b; Jin et al., 2023), we directly initialize them with visual latent codes and a projection matrix (More details in Appendix D.1). The LLM can process a sequence of interleaved visual and textual tokens, and predict the next visual or textual token in an autoregressive manner.

To pursue simplicity, efficiency, and scalability (Ge et al., 2023b), instead of additional components or loss functions used in VLMs, we train MLLMs with a single autoregressive objective. This allows for a fair comparison between MMGIC and coarse-grained image–caption data under the same framework, and ensures the generality and applicability of MMGIC to different MLLM frameworks. More importantly, our framework facilitates VL alignment **across** multiple granularities **simultaneously** through MMGIC and MLLMs' complex context processing capability. In all training stages, we follow common practice (Ge et al., 2023a; Zhu et al., 2023a) to **freeze** most of the parameters and only tune partial parameters of the LLM: a VL vocabulary, additional LoRA modules (Hu et al., 2021), norm layers and a language modeling head layer, to greatly improve efficiency. We pre-tokenize images into discrete visual token sequences and do not load all visual modules during training since they are frozen.

## 3.2 TRAINING STAGE 1: PRE-TRAINING (PT)

Similar to LLMs that tokenize text-only documents into discrete textual token sequences, for each data sample in MMGIC shown in Fig. 1 *(Right)* or an image–caption pair, our framework tokenize it

into a discrete token sequence consisting of interleaved visual and textual token sequences. To help the LLM distinguish between two types of sequences, following LaVIT, we add two special tokens `[IMG]` and `[/IMG]` to the vocabulary, and insert them before and after each visual token sequence, respectively. Our framework is trained with an autoregressive objective to maximize the likelihood of predicting the next visual or textual token. Details of training settings are provided in Appendix E.1.

### 3.3 TRAINING STAGE 2: SUPERVISED FINE-TUNING (SFT)

To align pre-trained MLLMs with natural language instructions, following previous works (Sun et al., 2023c;a; Liu et al., 2023c;b; 2024a; Zhu et al., 2023a; Hu et al., 2024a), we collect 1.21M samples from several public datasets for supervised fine-tuning, including multimodal instruction datasets (Liu et al., 2023b; Yu et al., 2023c; Zhang et al., 2023; Chen et al., 2023; LAION, 2024b; Brooks et al., 2023; Zhang et al., 2024) and text-only instruction datasets (Taori et al., 2023; Steven Tey, 2023), and 1M samples from an aesthetic image–caption dataset (LAION, 2024a). We also play back 1M samples from MMGIC to avoid forgetting the knowledge learned in the pre-training stage. Following LLaVA (Liu et al., 2023c;b; 2024a), all datasets are transformed into a unified format, which consists of a general system message and multiple instruction–answer pairs. Only answer tokens are accounted in loss calculation. Details of instruction data are provided in Appendix H.

## 4 EXPERIMENT

Based on MMGIC and a general MLLM framework, we can **explore** the potential of multi-grained concept annotations for MLLMs on various VL benchmarks in both pre-training and SFT stages. Specifically, we follow previous works (Ge et al., 2023b; Liu et al., 2023c; Zhu et al., 2023a) to focus on the zero-shot multimodal comprehension and generation capabilities, including image captioning (COCO (Chen et al., 2015), NoCaps (Agrawal et al., 2019)), text-to-image generation (COCO (Chen et al., 2015), VIST (Huang et al., 2016)), visual question answering (VQAv2 (Goyal et al., 2017), GQA (Hudson & Manning, 2019), VizWiz (Gurari et al., 2018)), comprehensive multi-choice benchmarks (POPE (Li et al., 2023c), MME (Yin et al., 2023), MMBench (Liu et al., 2023d), ScienceQA (Lu et al., 2022b), SEED-Bench (Li et al., 2023a)). More evaluation details are provided in Appendix E.2.

### 4.1 DATA RECIPES FOR MULTI-GRAINED CONCEPT ANNOTATIONS

As mentioned in Sec. 2.3 and Fig. 1, multi-grained concept annotations include four components: coarse-grained image captions (C), fine-grained category labels (L), label descriptions (D) and object regions (R). Hence, we design four data recipes to investigate the importance of each component on zero-shot image captioning and image generation tasks and find the best recipe. As shown in Table 1:

- {0, 1, 2}-th rows: simply appending category labels to image captions does not help and even hurts the performance on both tasks. MLLMs may struggle to understand the association between images, captions and category labels, leading to confusion and treating category labels as additional noise. Whereas, label descriptions can strengthen the association between images and annotations, to help MLLMs align the concepts represented by labels with the concepts in the image, thus mitigating the performance drop. As shown in Figure 3 *(Top)*, MMGIC(C) incorrectly identifies "accordion" as "electronic keyboard". While for MMGIC(CLD), label descriptions can help MLLMs better understand labels by **visual details** "pleated bellows and keyboard, box-like" and **relevant knowledge** "portable".
- {2, 3}-th rows: object regions can further **complement** other annotations to help MLLMs better locate and align concepts in images and in annotations at a fine-grained level, thus significantly improving the performance on both tasks. As shown in Figure 3 *(Bottom)*, MMGIC(CLD) incorrectly identifies "laying" as "sitting" and "on top of a toilet" as "on the floor next to a toilet". While for MMGIC(CLDR), object regions can help MLLMs **ground** concepts in the textual annotations to corresponding regions in the image, especially in terms of instance interaction and spatial relationship here, which is also consistent with meso analyses in Section 4.4.

We further analyse generated images by MLLMs pre-trained with different data recipes in Figure 4 *(Left)*. The input prompt and the label description of "bagel" are shown in the top and generated images are shown in the bottom. We found that while MMGIC(C) could accurately generate "on top of a white plate on top of a table", it fails to correctly generate "a bagel", let alone "a blueberry bagel".

Table 1: Zero-shot evaluation of different data recipes for MMGIC. C: image caption; L: category labels; D: label descriptions; R: object regions; CLIP-T/I: image–text or image-image similarity via CLIP; ↓: lower is better, otherwise higher is better. The best results are **bold**.

| | Data Recipes | | | | Image Captioning | | Image Generation | | | | |
|---|---|---|---|---|---|---|---|---|---|---|---|
| | | | | | COCO | NoCaps | MS-COCO-30K | | | VIST | |
| | C | L | D | R | CIDEr | CIDEr | FID (↓) | CLIP-T | CLIP-I | FID (↓) | CLIP-I |
| 0 | ✓ | | | | 113.64 | 99.11 | **7.20** | 30.81 | 71.62 | 67.61 | 62.22 |
| 1 | ✓ | ✓ | | | 113.67 | 96.80 | 10.52 | 30.43 | 71.09 | 71.67 | 61.96 |
| 2 | ✓ | ✓ | ✓ | | 116.02 | 98.61 | 8.92 | 30.89 | 71.60 | 51.89 | 64.30 |
| 3 | ✓ | ✓ | ✓ | ✓ | **118.30** | **102.01** | 7.36 | **31.57** | **72.24** | **35.33** | **66.10** |

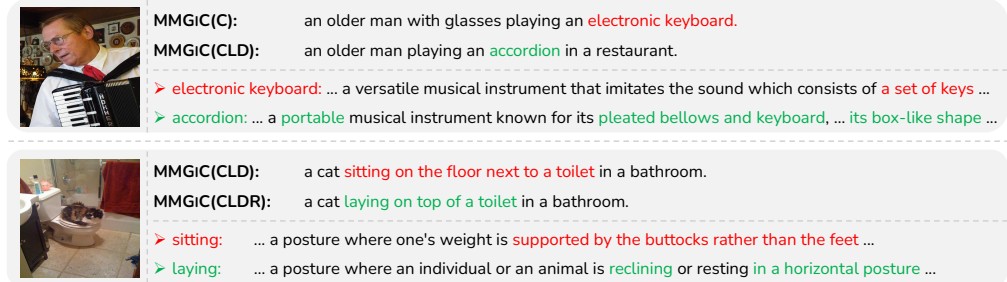

MMGIC(C):    an older man with glasses playing an electronic keyboard.
MMGIC(CLD):    an older man playing an accordion in a restaurant.

➢ electronic keyboard: ... a versatile musical instrument that imitates the sound which consists of a set of keys ...
➢ accordion: ... a portable musical instrument known for its pleated bellows and keyboard, ... its box-like shape ...

MMGIC(CLD):    a cat sitting on the floor next to a toilet in a bathroom.
MMGIC(CLDR):    a cat laying on top of a toilet in a bathroom.

➢ sitting:    ... a posture where one's weight is supported by the buttocks rather than the feet ...
➢ laying:    ... a posture where an individual or an animal is reclining or resting in a horizontal posture ...

Figure 3: Comparison of generated captions by MLLMs pre-trained with different data recipes. MMGIC(C), MMGIC(CLD) and MMGIC(CLDR) denote the {0, 2, 3}-th data recipes in Table 1, respectively. The bottom right of each example shows associated label–description pairs from MMGIC.

Label descriptions in MMGIC(CLD) can provide more **visual details** about both "bagel" and "blueberry", which helps MLLM better understand and generate the concept of "bagel" with the feature of "round bread product", and also add four "blueberries" to th "bagel". Furthermore, with the help of object regions in MMGIC(CLDR), MLLMs can correctly understand and generate a "bagel" with the feature of "its hole in the center". We also show some emergent abilities of MLLMs pre-trained with MMGIC in Figure 4 *(Right)*. Notably, MMGIC does **not** contain any samples about image editing and in-context image synthesis. Leveraging image–text interleaved documents with multi-grained concept annotations, the top two examples show that MLLMs can precisely understand editing instructions and perform appropriate editing, while the bottom example shows that MLLMs can synthesize image precisely based on the image–text interleaved sequences. More analyses are provided in Appendix C.2.

In summary, multi-grained concept annotations are not **isolated** from each other. With our structured template and MLLM framework, they can **integrate** into multimodal documents and **complement** each other to help MLLMs better learn concepts, thus improving the ability to understand and generate concepts. We take the 3-rd data recipe as the default data recipe and denote it simply as MMGIC.

## 4.2 COMPARISON AND COLLABORATION BETWEEN MMGIC AND IMAGE–CAPTION DATA

MMGIC is constructed from public object detection datasets mainly covers common concepts. Widely-used coarse-grained image–caption data, *e.g.*, Conceptual Captions (Sharma et al., 2018) and LAION-5B (Schuhmann et al., 2022a), typically cover diverse, scalable but noisy concepts. To explore the potential of MMGIC, we then investigate the **comparison** and **collaboration** between MMGIC with image–caption data. To strike a balance between increasing concept breadth, reducing data noise and improving efficiency, we collect several large-scale public image–caption datasets (Ordonez et al., 2011; Sharma et al., 2018; Changpinyo et al., 2021; Schuhmann et al., 2022a;b; Sun et al., 2024), and follow LLaVA (Liu et al., 2023c) to first filter them by the frequency of noun-phrases extracted by spaCy from their given synthesized captions, and then automatically synthesize captions same as MMGIC. We name this dataset as IC, where 52M unique images are collected and selected, almost 15 times more than MMGIC. More details are provided in Appendix G. As shown in Table 2:

Figure 4: Comparison of generated images by MLLMs pre-trained with different data recipes *(Left)* and image editing and multimodal in-context image synthesis examples *(Right)*.

Table 2: Zero-shot evaluation of comparison and collaboration between MMGIC and IC. IC-PART: select the same number of samples as MMGIC from IC; MMGIC+IC: joint training; MMGIC → IC: train on MMGIC first and then on IC. The best results are **bold** and the second-best are underlined.

| | Training Data | Image Captioning | | Image Generation | | | | |
| | | COCO | NoCaps | MS-COCO-30K | | | VIST | |
| | | CIDEr | CIDEr | FID ($\downarrow$) | CLIP-T | CLIP-I | FID ($\downarrow$) | CLIP-I |
|---|---|---|---|---|---|---|---|---|
| 0 | IC-PART | 95.74 | 85.00 | 11.62 | 29.94 | 68.21 | 64.03 | 63.41 |
| 1 | IC | 104.15 | 92.24 | 7.65 | 31.40 | 70.27 | 41.65 | 65.06 |
| 2 | MMGIC | 118.30 | 102.01 | 7.36 | 31.57 | **72.24** | **35.33** | **66.10** |
| 3 | MMGIC+IC | 106.45 | 92.98 | **7.11** | 31.65 | 70.93 | 36.54 | 65.89 |
| 4 | MMGIC → IC | 105.62 | 93.77 | 7.29 | 31.57 | 70.26 | 37.45 | 65.88 |
| 5 | IC → MMGIC | 120.84 | **105.59** | 7.13 | **31.96** | 71.54 | 37.13 | 65.62 |
| 6 | MMGIC+IC → MMGIC | **121.22** | 105.33 | 7.22 | 31.91 | 71.75 | 36.23 | 65.79 |

- {0, 1, 2}-th rows: comparing with IC, MMGIC can help MLLMs achieve **significantly** better performance on both tasks even with a much **smaller** number of samples, which demonstrates the effectiveness of multi-grained concept annotations in concept understanding and generation.

- {3, 4, 5, 6}-th rows: naturally, we try to improve the performance and concept breadth of MLLMs by exploring the collaboration between MMGIC and IC. However, simply joint training MMGIC and IC achieves better performance than IC, but significantly worse than MMGIC, especially on image captioning and VIST. We then follow the curriculum learning strategy (McCann et al., 2019) to train them in different orders. Interestingly, training on IC first and then on MMGIC achieves significantly better performance than all the above strategies on image captioning and partial metrics of image generation. This is consistent with recent findings (Hu et al., 2024b; Li et al., 2024a) that training with **high-quality** data **later** in the pre-training phase leads to better performance. Moreover, considering that the noise in IC still cause a slight performance drop on VIST ({2, 5}-th rows), we first jointly train on MMGIC and IC to alleviate the effect of noise in IC, and then on MMGIC, eventually achieving the best average performance (6-th row).

Overall, by exploring comparison and collaboration between MMGIC with large-scale coarse-grained image–caption data in the pre-training stage, we demonstrate that MMGIC achieves better performance than IC on both tasks, and their appropriate collaboration can further **improve** the average performance. We present {1, 2, 6}-th rows as three baselines, and denote them as MLLM-{IC, MMGIC, MMGIC & IC}. More discussions about collaboration strategies are provided in Appendix B.2.

## 4.3 EVALUATION ON DOWNSTREAM VISION–LANGUAGE BENCHMARKS AFTER SFT

To further explore the potential of MMGIC dataset on various downstream VL benchmarks, we then perform SFT (Section 3.3) on our three baselines. Technically, different training datasets, training settings, framework, etc., lead to non-comparable and unfair comparisons of our baselines with existing MLLMs. Hence, we show SOTA MLLMs in gray as upper bound references, and their computation and data resources are extremely expensive and large (well over 10 times that of our work).

Table 3: Zero-shot evaluation on multimodal comprehension benchmarks after SFT. MLLMs in Group a are for comprehension only, while MLLMs in Group b are for both comprehension and generation. MMB: MMBench; SQA$^I$: ScienceQA-IMG; SEED$^I$: SEED-Bench-IMG; *: w/o SFT.

| Model | Image Captioning | | VQA | | | Multi-Choice Benchmark | | | | |
| | COCO | NoCaps | VQAv2 | GQA | VizWiz | POPE | MME | MMB | SQA$^I$ | SEED$^I$ |
|---|---|---|---|---|---|---|---|---|---|---|
| *SOTA MLLMs as upper bound references, with more training data or trainable parameters, not comparable* | | | | | | | | | | |
| a   LLaVA-1.5-7B | | | 78.50 | 62.00 | 50.00 | 85.90 | 1826.80 | 65.20 | 66.80 | 65.80 |
|     Emu-I-14B | 120.40 | 108.80 | 62.00 | 46.00 | 38.30 | | | | | 58.00 |
|     Emu2-Chat-37B | | | 84.90 | 65.10 | 54.90 | | | 62.40 | | 68.90 |
| b   VL-GPT-I-7B | 133.70 | | 67.20 | 51.50 | 38.90 | | | | | |
|     LaVIT-v2-7B* | 133.30 | 112.00 | 68.30 | 47.90 | 41.00 | | | | | |
|     SEED-LLaMA-I-8B | 124.50 | 97.78 | 66.20 | 52.24 | 55.10 | 79.92 | 1497.53 | 52.58 | 60.24 | 51.50 |
| *Our comparable baselines* | | | | | | | | | | |
| 0   MLLM-IC | 108.13 | 92.71 | 70.28 | 56.02 | 52.62 | 81.14 | 1646.71 | 59.54 | 65.94 | 58.41 |
| 1   MLLM-MMGIC | 119.35 | 104.19 | 70.13 | 56.84 | 51.14 | 83.25 | 1636.47 | 58.51 | 65.79 | 60.03 |
| 2   MLLM-MMGIC & IC | **122.31** | **106.97** | **70.57** | **56.97** | **52.66** | **85.09** | **1668.19** | **59.88** | **66.24** | **60.75** |

Table 4: Zero-shot evaluation on multimodal generation benchmarks after SFT. MLLMs in Group a are for generation only, while MLLMs in Group b are for both comprehension and generation.

| Model | MS-COCO-30K | | | VIST | |
| | FID ($\downarrow$) | CLIP-T | CLIP-I | FID ($\downarrow$) | CLIP-I |
|---|---|---|---|---|---|
| *SOTA MLLMs as upper bound references, not comparable* | | | | | |
| a   KOSMOS-G (Pan et al., 2023) | 10.99 | | | | |
|     GILL (Koh et al., 2024) | 12.20 | | | | 64.10 |
|     Emu2-Gen-37B (Sun et al., 2023a) | | 29.70 | 68.60 | | |
| b   VL-GPT-I-7B (Zhu et al., 2023a) | 11.53 | | | | |
|     LaVIT-v2-7B* (Jin et al., 2023) | 7.10 | 31.93 | 71.06 | 34.76 | 68.41 |
|     SEED-LLaMA-I-8B (Ge et al., 2023b) | 16.66 | 29.52 | 69.22 | 43.69 | 65.21 |
| *Our comparable baselines* | | | | | |
| 0   MLLM-IC | 8.11 | 30.90 | 70.72 | 38.19 | 65.37 |
| 1   MLLM-MMGIC | **6.79** | **31.63** | **72.44** | **34.32** | **67.66** |
| 2   MLLM-MMGIC & IC | 7.29 | 31.54 | 72.03 | 34.39 | 67.19 |

**Zero-shot Multimodal Comprehension.** As shown in Table 3, MLLM-MMGIC & IC achieves the best performance on all 10 benchmarks compared to the other two baselines, and even outperforms some SOTA MLLMs with more training data or full-param training or larger LLMs on some benchmarks. Moreover, even with less than 4M pre-training data compared to MLLM-IC with 52M pre-training data, MLLM-MMGIC significantly outperforms MLLM-IC on the benchmarks that inspect **in-depth** understanding of common concrete concepts, *e.g.*, COCO, NoCaps, POPE, SEED-Bench. In contrast, MLLM-IC outperforms MLLM-MMGIC on benchmarks that require a **broader** understanding of concrete concepts, *e.g.*, VizWiz, MME, MMBench. More importantly, MLLM-MMGIC & IC can effectively **combine** the strengths of both in terms of **depth** and **breadth** of concept representation and further improve performance, *e.g.*, 3.95% and 2.34% accuracy improvements on POPE and SEED-Bench. We further analyse in terms of dataset statistics and concept overlap in App. C.1.

**Zero-shot Multimodal Generation.** As shown in Table 4, for two text-to-image generation benchmarks that focus on common concrete concepts, MLLM-MMGIC achieves the best performance, and matches or even outperforms some SOTA MLLMs on some metrics. This demonstrates that fine-grained category labels, label descriptions and object regions can help MLLMs better **learn** and **generate** concepts. Besides, the noise introduced by IC (discussed in Section 4.2) in the pre-training stage may not be well alleviated by SFT, thus MLLM-MMGIC & IC achieves the second-best performance. More results and analyses on image editing are provided in Appendix C.2.

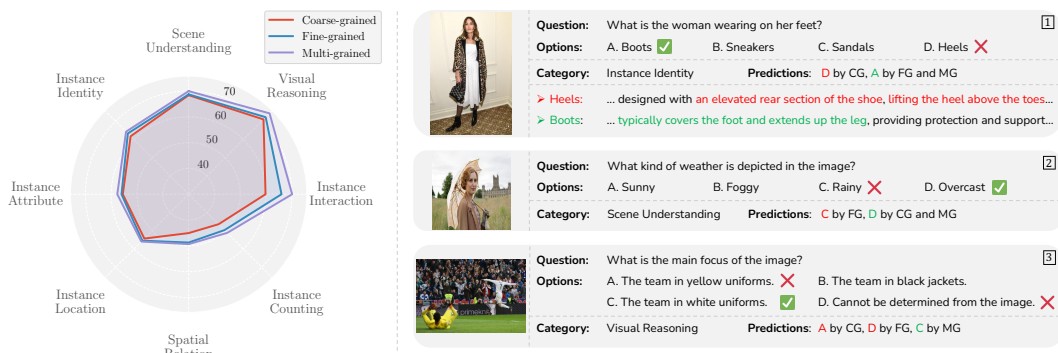

Figure 5: Analysis on 8 dimensions of SEED-Bench-IMG. Left: the performance of MLLM-MMGIC trained with different-grained concept annotations from MMGIC. Right: corresponding case studies. CG, FG, and MG denote MLLMs trained with coarse-, fine-, and multi-grained concept annotations from MMGIC, respectively. ✅ denote the ground truth; ✗ denote incorrect prediction(s).

## 4.4 THE IMPACT OF DIFFERENT-GRAINED CONCEPT ANNOTATIONS

To delve into the potential of multi-grained concept annotations for MLLMs, we conduct meso analyses on SEED-Bench (Li et al., 2023a), a large-scale multi-choice benchmark for multimodal comprehension. We follow common practice (Ge et al., 2023b) to select 9 image evaluation dimensions, *i.e.*, about 14K questions, and denote it as SEED-Bench-IMG. We ignore the "*Text Recognition*" dimension due to noisy data and small scale (only 85), leading to result fluctuations. "*Scene Understanding, Visual Reasoning*" dimensions focus on the holistic understanding and cross-modal reasoning of the image, while the other 6 dimensions focus on in-depth understanding of concrete concepts in the image. "Coarse-grained" (CG) means that only image captions from MMGIC are used. "Fine-grained" (FG) uses category labels, label descriptions and object regions from MMGIC. "Multi-grained" (MG) means that both above are used.

**Quantitative Analysis.** We first explore the impact of different-grained concept annotations from MMGIC on each evaluation dimension of SEED-Bench-IMG in Figure 5 *(Left)*. For overall accuracy, FG significantly improves performance over CG by 1.39 points, and multi-grained can further improve by 1.4 points. For each evaluation dimension:

- FG provides **deeper** understanding of concepts than CG. Compared with coarse-grained image captions, fine-grained concept annotations can help MLLMs better **understand** and **locate** concepts in images, and **recognize** relationships between concepts, especially on the "*Instance Identity, Spatial Relation, Instance Counting, Instance Interaction, Visual Reasoning*" dimensions.

- MG can facilitate **collaboration** between concept annotations of different granularities, thus fully **integrating** each other's strengths and achieving further improvements in all dimensions, especially on the "*Scene Understanding, Instance Attribute, Instance Counting, Instance Interaction, Visual Reasoning*" dimensions. This demonstrates that our structured template for MMGIC can help MLLMs better utilize multi-grained concept annotations to learn concepts and thus promote vision–language alignment **across** multiple granularities **simultaneously**.

**Qualitative Analysis.** To better analyse the advantages of different-grained concept annotations in MMGIC for MLLMs, we provide corresponding qualitative analysis in Figure 5 *(Right)*. More case studies are provided in Appendix C.4.

- Example ①  "Instance Identity": FG provides **deeper** understanding of concepts than CG. While CG provides a holistic description of the image, it cannot help MLLMs distinguish between "Heels" and "Boots". However, FG provides visual details "lifting the heel above the toes" and "covers the foot and extends up the leg" by label–description pairs, which help MLLMs distinguish between similar concepts. Object region-annotation pairs further help MLLMs better locate and learn these concepts. Hence, FG helps **capture** visual details and **identify** concepts correctly.

- Example ⟨2⟩ "Scene Understanding": CG provides more **holistic** understanding of the image than FG. While the lady in the image is holding an "umbrella", her surroundings show that the sky is covered with clouds, and it is not raining. MLLMs trained with FG seem to be too focused on the "umbrella" and ignore the overall scene of the image, while CG can help MLLMs predict the correct answer by better understanding the **global context**.
- Example ⟨3⟩ "Visual Reasoning": MG can **combine** the advantages of CG and FG to improve visual reasoning. This image shows a football match with a player in "yellow uniforms" lying on the ground and a player in "white uniforms" celebrating as the audience cheers him on. Rich objects and details confuse MLLMs trained with FG to confirm the main focus of the image, while MLLMs trained with CG perceive the visually more prominent "yellow" player as the main focus. However, with our structured template and MLLM framework, MG enables different-grained concept annotations to **integrate** and **complement** each other, thus better understanding and reasoning about the image from both **global context** and **local details**. MLLMs trained with MG correctly identify the "white" player as the main focus of the image.

We also conduct experiments on self-generated annotations in evaluation, MMGIC directly as SFT data, the impact of the nature of image–text interleaved, text-only performance in Appendix C.3 & C.

## 5 RELATED WORK

MLLMs have emerged recently including ones (Sun et al., 2023c; Ge et al., 2023b; Jin et al., 2023; Zhu et al., 2023a; Dong et al., 2024) that propose multimodal generalists capable of both multimodal comprehension and generation. However, existing MLLMs often overlook the importance of concepts and only utilize coarse-grained concept annotations, *e.g.*, image captions, which may limit vision–language alignment. Factually, many traditional VLMs recognized the importance of concepts in vision–language learning. To better utilize concepts in vision–language learning and improve performance, they incorporated fine-grained concept annotations into coarse-grained image captions. For example, object labels (Li et al., 2020; Zhang et al., 2021), label descriptions (Shen et al., 2022; Yao et al., 2022; Menon & Vondrick, 2023; Li et al., 2024b), region descriptions (Zeng et al., 2021) and object regions (Zeng et al., 2021; Li et al., 2022b). Especially, Oscar (Li et al., 2020) appends fine-grained object labels detected in the image after the coarse-grained image caption to simplify semantic alignment between vision and language. X-VLM (Zeng et al., 2021) proposes to align visual concepts (images and object regions) with coarse-grained image captions and fine-grained object labels and object region descriptions in multi-granularity. In this paper, different from previous VLM work, we collect and construct rich multimodal multi-grained concept annotations in MMGIC dataset. Without additional components or loss functions, we design a structured template to leverage the advantages of MLLMs under an autoregressive discrete framework. Through evaluation of 12 multimodal comprehension and generation benchmarks, as well as the comparison, combination, and analysis of MMGIC and image–caption data, for the first time, we explore and demonstrate the potential of MMGIC in MLLMs.

## 6 CONCLUSION AND FUTURE WORK

We introduce a new multimodal dataset, MMGIC, providing multi-grained concept annotations in both textual and visual form. MMGIC allows us to measure the potential of appropriate use of concepts. With our structured template and autoregressive discrete framework, multi-grained concept annotations can integrate and complement each other to help MLLMs better locate and learn concepts, thereby aligning vision and language across multiple granularities simultaneously. Furthermore, MMGIC and coarse-grained image–caption data each have their own strengths in depth and breadth of concept representation, and appropriately combining them can effectively integrate their strengths to further improve performance. We hope to inspire future research to further explore the potential of multi-grained concept annotations by incorporating more different types of annotations, scaling up data by automatic synthesis, scaling up (concrete and even abstract) concepts, and other VL tasks.

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

This appendix is organized as follows.

## A    BROADER IMPACT

In this paper, we introduce MMGIC, a new dataset with multimodal multi-grained concept annotations for MLLMs. We first collect, pre-process and complement four public large-scale human-annotated object detection datasets. With a well-designed structured template, we transform these datasets into our MMGIC dataset to address the lack of such datasets in the MLLM field and support our exploration. Under an autoregressive discrete framework, we explore the potential of MMGIC for MLLMs by evaluations and analyses on various downstream vision–language benchmarks. We demonstrate that MMGIC can provide multi-grained concept annotations to help MLLMs better locate and learn concepts, thus facilitating vision–language alignment across multiple granularities simultaneously. We will open-source the models and the code of the complete pipeline from data processing, model training to evaluation for facilitating more reproducible research. We hope that our work can inspire more research on multi-grained concept annotations for MLLMs.

We do not anticipate a specific negative impact, but, as with any multimodal large language model (MLLM), we recommend to exercise caution. MMGIC dataset is constructed from four public widely-used object detection datasets and will not bring any potential ethical or societal issues. We will follow the usage rules and copyright of original datasets. Newly generated data parts will follow the usage rules and copyrights of the corresponding open-source or closed-source models. We will require researchers who use our code and models to follow the principles of positive AI research to avoid model abuse and negative societal impact.

## B    LIMITATION AND FUTURE WORK

### B.1    AUTOMATIC ANNOTATION SYNTHESIS

Nowadays, both LLMs and MLLMs can continuously improve the performance and generalization ability by scaling up training data. Although our MMGIC dataset, based on several human-annotated object detection datasets and our well-designed data pipeline, can provide high-quality multi-grained concept annotations, it is still limited by the scale of the original datasets. A natural idea is to automatically synthesize a variety of different types of annotations for any image. This is a promising direction for future research, which can scale up to more concepts and continuously improve MLLMs.

Recent works (Peng et al., 2023; Pan et al., 2023; Wang et al., 2023b; Rasheed et al., 2023; Li et al., 2023d; Yang et al., 2023; Li et al., 2024c) have proposed to automatically synthesize various

annotations for large-scale web images with external open/close source models and complicated pipelines. KOSMOS-2 (Peng et al., 2023) extracted noun phrases from image captions and detected bounding boxes from images to synthesize Grounded image–text pairs (GRIT) to improve the grounding capability of MLLMs. KOSMOS-G (Pan et al., 2023) synthesized image captions and segment object regions for images to construct compositional image generation instruction tuning data to improve the zero-shot subject-driven image generation capability of MLLMs. All-Seeing (Wang et al., 2023b) and GLaMM (Rasheed et al., 2023) designed complicated pipelines to combine multiple external models to synthesize various annotations for large-scale web images from SA-1B (Kirillov et al., 2023), including detailed image captions, object labels, object regions, question-answer pairs about objects in the image, scene graphs, and so on, to improve the image/region-level captioning and grounding capability of MLLMs. SoM (Yang et al., 2023) synthesized semantic segmentation masks for images to improve the image segmentation and grounding capability of MLLMs.

Although these works have shown that automatically synthesized annotations can improve the performance of MLLMs, cascading multiple external models and complicated pipelines may introduce noise and bias to the synthesized annotations. Existing works introduced complex data pipelines and costly manual reviews (especially for pre-training data exceeding 1M) to reduce noise, but their released data still contains quite a bit of hallucinations, noise, and redundancy. Moreover, these works mainly focus on exploring the potential of automatically synthesized annotations in the multimodal comprehension or generation capability of MLLMs, and have not explored both under a unified framework at the same time. Among them, All-Seeing and GLaMM are similar to MMGIC proposed in this paper. Both of them synthesize various annotations for images, including image captions, category labels and object regions provided in MMGIC. They don't provide label descriptions but provide visual question–answering pairs or scene graphs, or other annotations. However, the final data forms of them are similar to the related works in traditional VLMs, where each image is paired with isolated different types of annotations. They need to introduce additional components and loss functions to utilize different annotations, such as object region annotations, to optimize the model's comprehension capability at different granularities separately. Besides, their experiments mainly explore the object recognition and segmentation, image/region-level captioning and grounding capability of MLLMs, which are actually downstream tasks that obviously benefit from their synthesized/annotated different types of annotations.

In contrast, in this paper, from the perspective of concept annotations, we explore the potential of multi-grained concept annotations for MLLMs for the first time, in both multimodal comprehension and multimodal generation. We mainly focus on general multimodal benchmarks instead of specific benchmarks that fine-grained annotations are good at (*e.g.*, object recognition and grounding benchmarks). We want to evaluate and analyze the potential of multi-grained concept annotations for MLLMs in a more comprehensive, general and fair way, by comparing and collaborating with widely-used coarse-grained image–caption data.

We believe that the automatic synthesis of multi-grained concept annotations for any image is a promising direction for future research. In addition, incorporating more different types of annotations, including not only various annotations found in existing works, but also annotations for text-rich images and table/chart images, would be valuable for future research. We have explored the potential of multi-grained concept annotations for MLLMs in both multimodal comprehension and generation, including image captioning, text-to-image generation, visual question answering, multi-choice benchmarks, and image editing. It is interesting to explore more benchmarks on object recognition and segmentation, grounding capability and other vision–language tasks in the future. Finally, in this work, we focus on concrete concepts, especially objects, attributes of objects, and relationships between objects. It is very interesting to explore more concrete concepts and also abstract concepts, such as emotions, events, and so on, in the future.

## B.2 COLLABORATION STRATEGY OF MMGIC AND IMAGE–CAPTION DATA

In Section 4.2, we investigate comparison and collaboration between MMGIC and coarse-grained image–caption dataset IC on 4 benchmarks for multimodal comprehension and generation. Simply joint training with MMGIC and IC cannot bring performance improvements compared to training with MMGIC only, and even leads to obvious performance degradation on COCO, NoCaps and VIST. Considering that IC is $\sim 15$ times larger than MMGIC, we also try to balance the two datasets

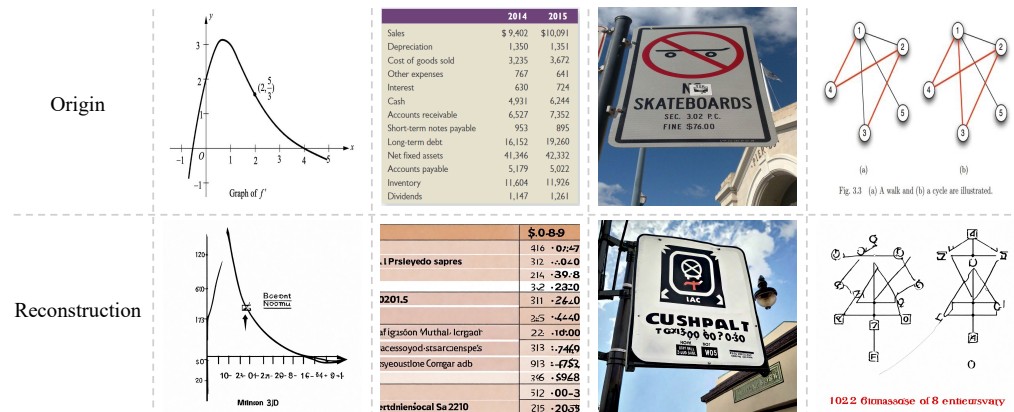

Figure 6: Visualization of the original and reconstructed images by the visual tokenizer inherited from LaVIT (Jin et al., 2023) used in this paper. Text-rich and chart images cannot be well reconstructed.

by using $2 \sim 4$ times of MMGIC[1]. However, repeating MMGIC, *e.g.*, 4MMGIC+IC, introduces only minor performance fluctuations and results in lower average performance.

Therefore, we follow the curriculum learning strategy (McCann et al., 2019) to train them in different orders and achieves significantly better average performance by training on IC first and then on MMGIC. This is consistent with recent findings (Hu et al., 2024b; Li et al., 2024a) that training with high-quality data late in the pre-training phase leads to better performance. Moreover, considering that the noise in IC still cause a slight performance drop on VIST ($\{2, 5\}$-th rows in Table 2), and jointly training on MMGIC and IC outperforms IC on both tasks ($\{1, 3\}$-th rows in Table 2). Hence, we first jointly train on MMGIC and IC to alleviate the effect of noise in IC, and then on MMGIC, eventually achieving the best average performance (6-th row in Table 2). Similarly, further increasing MMGIC repetitions, *e.g.*, 3MMGIC+IC → MMGIC, only yields minor performance fluctuations and leads to noticeable average performance drops. We speculate that repetition of MMGIC leads to overfitting of MMGIC and insufficient learning of IC, resulting in performance declines and fluctuation. We believe that the collaboration strategy of MMGIC and image–caption data is an interesting research question about data mixing (Liu et al., 2024b), data repetition (Muennighoff et al., 2024) and curriculum learning (Soviany et al., 2022). Automatic annotation synthesis for large-scale image–caption data we discussed in Appendix B.1 is also a promising solution to scale up to more concepts and continuously improve MLLMs without suffering from the problem of data duplication.

### B.3  A MLLM Framework for Multimodal Comprehension and Generation

**Parameter-Efficient Training.**   In this work, we standardize a generative vision–language framework based on existing MLLMs, which consists of a LLM and several visual modules. We follow previous works (Ge et al., 2023a; Zhu et al., 2023a) to freeze all visual modules and most of the LLM parameters during training to greatly improve efficiency. Although the performance of the framework is competitive on various vision–language benchmarks, it is limited by the frozen visual modules and parameter-efficient fine-tuning of the LLM. In the future, we will explore full-parameter fine-tuning of the LLM and visual modules to further improve the performance of the framework.

**Visual Tokenizer.**   Unlike traditional visual tokenizers like VQ-VAE (van den Oord et al., 2017) that reconstruct from original image pixels, SEED (Ge et al., 2023a) followed BEiT v2 (Peng et al., 2022) to train the visual tokenizer by reconstructing from visual embeddings with high-level semantics, and found that the latter strategy can better capture high-level semantics instead of low-level image details in the former strategy, which is more effective for multimodal comprehension. As the concurrent work of SEED, LaVIT also found that high-level semantics matters for downstream tasks, and further supported dynamic sequence length varying from the image. Compare SEED and LaVIT, we take the visual tokenizer from LaVIT, which can better preserve image details and semantic information.

---

[1]We follow the advice from Muennighoff et al. (2024) to repeat data no more than 4 times.

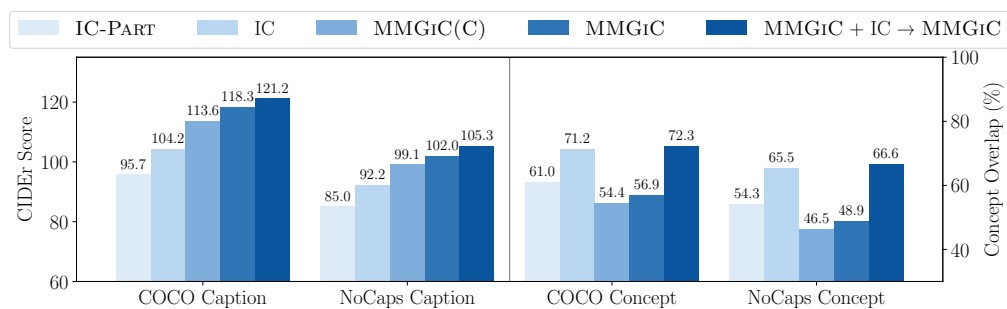

Figure 7: Comparison of performance (CIDEr Score) and concept overlap (%) for different training data settings on image captioning datasets COCO and NoCaps.

However, no matter which visual tokenizer is used, the vector quantization process is a trade-off between the cost/difficulty of learning and the capacity of representing visual information. For example, as shown in Figure 6, the visual tokenizer of LaVIT we used in this work cannot well reconstruct the original image by the visual decoder and diffusion model, especially text-rich images and chart images. Chameleon (Team, 2024) also found such limitations of traditional visual tokenizers (reconstructed from original image pixels) on heavy OCR-related tasks.

We also conduct experiments on downstream VL benchmarks containing text-rich images and table/chart images to further validate the limitation of the visual tokenizer. Our MLLM-MMGıC & IC achieves 32.0 accuracy on MMMU (Yue et al., 2023) validation set, where the human performance, the best model performance and random performance are $88.6, 59.4, 22.1$, respectively; and achieves 26.0 accuracy on MathVista (Lu et al., 2024b) testmini set, where the human performance, the best model performance and random performance are $60.3, 63.9, 17.9$, respectively. In the future, we will explore more advanced visual tokenizers to better capture high-level semantics and low-level image details for multimodal comprehension on general, text-rich and table/chart images.

**LLM Part.** We conduct experiments with LLaMA-2-7B as the LLM part in our framework. LLaMA-2-7B is widely used in the LLM and MLLM fields due to its good performance and generalizability, ensuring the reliability, reproducibility and scalability of our experimental results. Especially, most of the existing SOTA MLLMs (e.g., EMU (Sun et al., 2023c), LaVIT (Jin et al., 2023), SEED-LLaMA (Ge et al., 2023b), DreamLLM (Dong et al., 2024), VL-GPT (Zhu et al., 2023a)) all use the LLaMA series or its variants to initialize the LLM part of their models. Since our research aims to explore the potential of MMGıC as a new data paradigm for MLLMs, we follow this common practice to use LLaMA-2-7B in our framework. It will be interesting to explore more advanced LLMs in the future to further improve the performance of our framework, considering recent findings that the model size scaling of LLM is more effective than image encoder in yielding improved performance (Li et al., 2024a).

## C  MORE EXPERIMENTAL RESULTS AND ANALYSIS

### C.1  CONCEPT OVERLAP

In Section 4.3, we find that MMGıC and IC have their own strengths in depth and breadth of concept representation. To further verify our findings, inspired by K-LITE (Shen et al., 2022), we investigate the concept overlap between different training data settings and downstream datasets in terms of dataset statistics. The concept overlap is computed as the percentage of concepts in a downstream dataset that are covered by the training data. We select two image captioning datasets, COCO and NoCaps, and directly take noun chunks extracted from their ground-truth captions as concepts. We select the $\{0, 1, 2, 6\}$-th data recipes in Table 2 and also include MMGıC(C) from $\{0$-th data recipe in Table 1 for better comparison. As shown in Figure 7, we can see that:

- Comparing IC-PART and IC: with the number of image–caption pairs increasing from less than 4M to almost 52M, the concept overlap increases significantly (10.2% and 11.2%) and the performance also improves obviously (8.5 and 7.2 points) on both datasets.

Table 5: Zero-shot evaluation on the image editing benchmark DreamBench (Ruiz et al., 2023) after SFT. DINO: cosine similarity between generated and real images via DINO (Caron et al., 2021). The best results are **bold** and the second-best are underlined.

| | Model | DreamBench | | |
|---|---|---|---|---|
| | | CLIP-T | CLIP-I | DINO |
| *SOTA MLLMs as upper bound references, not comparable* | | | | |
| a | KOSMOS-G (Pan et al., 2023) | 28.70 | 84.70 | 69.40 |
| | Emu2-Gen-37B (Sun et al., 2023a) | 28.70 | 85.00 | 76.60 |
| b | SEED-LLaMA-I-8B (Ge et al., 2023b) | 27.06 | 79.27 | 54.42 |
| *Our comparable baselines* | | | | |
| 0 | MLLM-IC | 27.50 | 82.92 | 67.41 |
| 1 | MLLM-MMGIC | **27.69** | 82.62 | 67.36 |
| 2 | MLLM-MMGIC & IC | 27.62 | **83.06** | **68.30** |

- Comparing MMGIC(C) and MMGIC: the performance improvement (4.7 and 2.9 points) may not come from improved concept overlap (only 2.5% and 2.4%). This provides side evidence that fine-grained concept annotations in MMGIC can integrate and complement coarse-grained concept annotations in MMGIC(C), to help MLLMs learn concepts deeply and thus improve performance.
- Comparing MMGIC and MMGIC+IC → MMGIC: MMGIC can collaborate with IC to achieve higher concept overlap (15.4% and 17.7%). Appropriate curriculum learning strategies can effectively integrate the strengths of both in terms of depth and breadth of concept representation, thus improving performance on both datasets (2.9 and 3.3 points).
- Comparing IC-PART, IC and MMGIC(C): although they are all image–caption pairs and captions are all synthesized by the same pipeline (Appendix F.4), MMGIC(C) can achieve better performance on both datasets with lower concept overlap and the same number of image–caption pairs as IC-PART. We attribute this to the fact that the images in MMGIC(C) have higher quality, *i.e.*, contain more concepts and are more complex than IC-PART and IC collected from the web. Specially, the number of unique noun chunks in the three training data are: 1.7M, 10.9M, and 0.4M, respectively, while the average number of unique noun chunks per image are: 3.0, 3.0, and 8.8, respectively. This is consistent with recent findings (Dai et al., 2024; Li et al., 2024a) that the impact of data quality is greater than data scale.

## C.2 ANALYSIS ON IMAGE EDITING

To further investigate how MMGIC can better help MLLMs utilize concepts to align vision and language and improve the capability of generating concepts we conduct both qualitative and quantitative analyses on image editing task.

Although MMGIC does not contain any samples about image editing, with our structured template, each sample of MMGIC can be seen as an image–text interleaved document with an image, multiple cropped regions from the image, textual annotations, and also template instruction text. Such design can help MLLMs better understand each region in the image, as well as the relationships within and between regions, thus better learning and generating concepts, and aligning images and instructions. We hypothesize that this may help MLLMs learn more diverse image generation abilities.

**Quantitative Analysis.** To verify this, we include about 0.32M image editing instruction samples from Instructpix2pix (Brooks et al., 2023) and MagicBrush (Zhang et al., 2024) during the SFT stage, and evaluate the performance of our baselines on the image editing benchmark DreamBench (Ruiz et al., 2023), as shown in Table 5. Compared MMGIC with IC, IC performs better on image (subject) fidelity (CLIP-I and DINO), and MMGIC performs better on text fidelity (CLIP-T, i.e., adherence to editing instructions). We attribute this to the fact that the structured template and rich multimodal annotations in MMGIC can help MLLMs better understand and follow editing instructions, while the massive image–caption pairs and higher concept breadth in IC can help MLLMs better generate

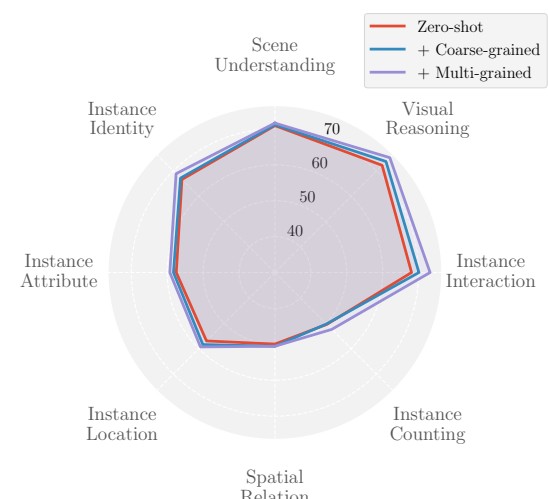

Figure 8: Detailed evaluation of MLLM-MMGIC & IC with different evaluation strategies on 8 dimensions of SEED-Bench-IMG.

and edit images. Furthermore, we also find that the collaboration of MMGIC and IC combines the advantages of both, and achieves better average performance on both image and text fidelity. Compared with existing SOTA MLLMs, MMGIC & IC not only significantly outperforms SEED-LLaMA (Ge et al., 2023b) (using more pre-training data, SFT data, and full-param fine-tuning), but also achieves comparable performance to generation-only MLLMs (KOSMOS-G (Pan et al., 2023) and Emu2-Gen-37B (Sun et al., 2023a)) trained with far more image generation and editing data.

**Qualitative Analysis.** We also show some qualitative examples of image editing from MLLMs pre-trained with MMGIC in Figure 4 *(Right)*. The top two examples show that MMGIC can precisely understand editing instructions and perform appropriate editing, while the bottom example shows that MMGIC can synthesize image precisely based on the image–text interleaved sequences. Considering that MMGIC dataset does not contain any samples about above two emergent abilities of image editing and multimodal in-context image synthesis, we hypothesize that integrating multi-grained concept annotations into image–text interleaved documents through our structured template can help MLLMs better understand and locate objects in the image, to more accurately edit the objects in the image, and synthesize images based on the image–text interleaved sequences.

### C.3 SELF-GENERATED MULTI-GRAINED CONCEPT ANNOTATIONS IN EVALUATION

Recent works (Sun et al., 2023c; Wu et al., 2023b; Mitra et al., 2023) have proposed to ask MLLMs to answer questions with the help of image annotations (*e.g.*, captions, or scene graphs) generated by themselves or external models during evaluation. Inspired by these works, we naturally ask MLLMs trained with MMGIC to self-generate coarse- and fine-grained concept annotations for images, like annotations provided in MMGIC. Considering the inference efficiency and avoiding noise, we only generate category labels as fine-grained concept annotations. Specifically, we explore different evaluation strategies on SEED-Bench-IMG in Fig. 8. After training with MMGIC, fine-grained concept annotations can help MLLMs generate better coarse-grained image captions to improve performance, especially on the "*Instance Location, Instance Interaction, Visual Reasoning*" dimensions. Furthermore, self-generate multi-grained concept annotations can integrate the advantages of both granularities, achieving 1.68 points improvement in overall accuracy compared with the zero-shot evaluation, especially on the "*Instance Identity, Instance Counting, Instance Interaction, Visual Reasoning*" dimensions. This demonstrates that multi-grained concept annotations can help MLLMs better understand and reason about concepts in the image during evaluation.

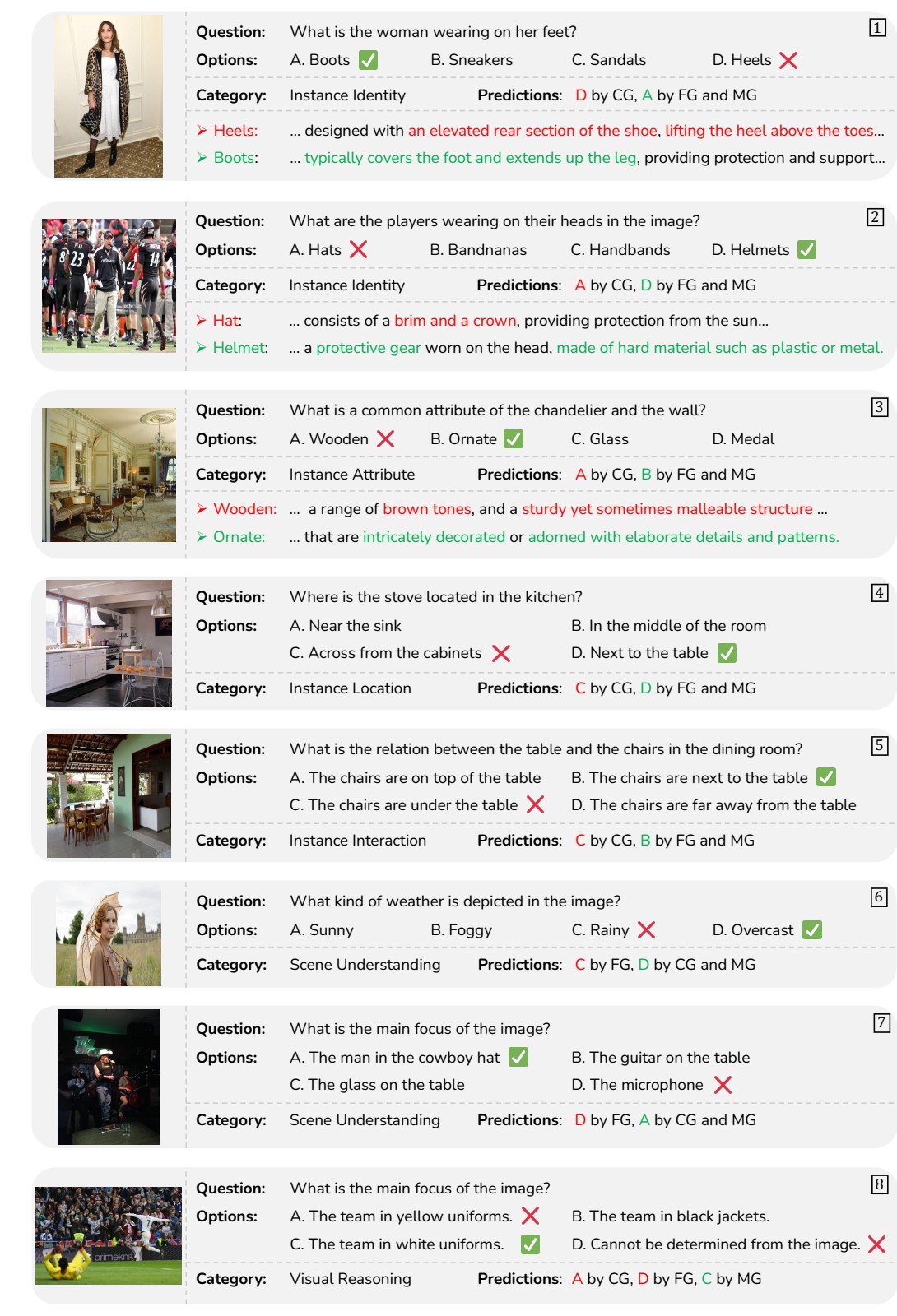

Figure 9: Case study of MLLM-MMGIC trained with different-grained concept annotations from MMGIC on SEED-Bench-IMG. CG, FG, and MG denote MLLMs trained with coarse-, fine-, and multi-grained concept annotations from MMGIC, respectively. ✅ denote the ground truth; ✗ denote incorrect prediction(s). The bottom right of the first three examples show the associated label–description pairs from MMGIC.

Table 6: Zero-shot evaluation on multimodal comprehension and generation benchmarks after SFT. MMB: MMBench; SQA$^I$: ScienceQA-IMG; SEED$^I$: SEED-Bench-IMG. The best results are **bold**.

| Model | Image Captioning | | VQA | | | Multi-Choice Benchmark | | | | | MS-COCO-30K | | | VIST | |
|---|---|---|---|---|---|---|---|---|---|---|---|---|---|---|---|
| | COCO | NoCaps | VQAv2 | GQA | VizWiz | POPE | MME | MMB | SQA$^I$ | SEED$^I$ | FID ($\downarrow$) | CLIP-T | CLIP-I | FID ($\downarrow$) | CLIP-I |
| 0 MLLM-IC | 108.13 | 92.71 | 70.28 | 56.02 | 52.62 | 81.14 | 1646.71 | 59.54 | 65.94 | 58.41 | 8.11 | 30.90 | 70.72 | 38.19 | 65.37 |
| 1 + MMGIC | **119.03** | **105.49** | **70.33** | **56.67** | **52.63** | **82.31** | **1656.29** | **59.64** | **66.04** | **59.60** | **7.65** | **31.26** | **71.66** | **34.51** | **66.61** |

## C.4 Qualitative Analysis of the Impact of Different-Grained Concept Annotations on MLLMs

We have provided the quantitative results of MLLM-MMGIC with different-grained concept annotations from MMGIC on each evaluation dimension of SEED-Bench-IMG in Section 4.4. Here, we further provide corresponding qualitative analysis to better analyse the advantages of coarse-, fine-, and multi-grained concept annotations in MMGIC for MLLMs. As shown in Figure 9, we can see that:

- Examples $\boxed{1}$ and $\boxed{2}$ from the "Instance Identity" dimension: while CG can provide a holistic description of the whole image by the image caption, MLLMs trained with CG cannot distinguish between "Boots" and "Heels" in $\boxed{1}$, and "Helmets" and "Hats" in $\boxed{2}$. However, FG can provide visual details "covers the foot and extends up the leg" and relevant knowledge "protective gear" by label–description pairs, which can help MLLMs better understand these concepts. Object regions can further help MLLMs better locate and learn these concepts. Therefore, MLLMs trained with FG or MG can capture these local details and identify concepts correctly.

- Example $\boxed{3}$ from the "Instance Attribute" dimension: after locating the "chandelier" and "wall" in the image, MLLMs trained with FG or MG can recognize the shared attribute "Ornate" correctly by the visual details "decorated with elaborate details and patterns".

- Example $\boxed{4}$ from the "Instance Location" dimension: MLLMs trained with FG or MG can identify and locate the "stove" in the image, and then correctly analyse its surrounding objects and the spatial relationships between them.

- Example $\boxed{5}$ from the "Instance Interaction" dimension: the interaction between the table and the chairs requires the model to correctly understand the local details of their positions and spatial relationships. The image shows that the table is placed around the chairs, rather than completely under the table. Such details can be correctly captured by MLLMs trained with FG or MG.

- Example $\boxed{6}$ from the "Scene Understanding" dimension: while the lady in the image is holding an "umbrella", her surroundings show that the sky is covered with clouds, but it is not raining, which implies that the weather is "overcast". MLLMs trained with FG seem to be too focused on the "umbrella" to the overall scene of the image, while MLLMs trained with CG or MG can predict the correct answer by better understanding the global context of the image.

- Example $\boxed{7}$ from the "Scene Understanding" dimension: similarly, MLLMs trained with FG pay too much attention to the "microphone" in the image. In fact, "the man in the cowboy hat" is standing prominently in the scene. The lighting, his position, and his stance all draw attention to him as the main subject.

- Example $\boxed{8}$ from the "Visual Reasoning" dimension: the image shows a football match, where a player in the "white uniform" is celebrating, and a player in the "yellow uniform" is lying on the ground. The audience in the background is cheering for the player in the "white uniform" and taking pictures of him. Since the image contains rich objects and details, MLLMs trained with FG have difficulty confirming the main focus of the image, while MLLMs trained with CG believe that the visually more prominent "yellow" player is the main focus. However, MMGIC with our structured template can help MLLMs combine the advantages of CG and FG, make different-grained concept annotations complement each other, and better understand and reason about the image from both global scene and local details. Hence, MLLMs trained with MG can correctly identify the "white" player as the main focus of the image.

Table 7: Zero-shot evaluation on multimodal generation benchmarks. For each block, the best result are **bold**.

| | Training Data | MS-COCO-30K | | | VIST | |
|---|---|---|---|---|---|---|
| | | FID ($\downarrow$) | CLIP-T | CLIP-I | FID ($\downarrow$) | CLIP-I |
| 0 | MMGIC(C) | **7.20** | 30.81 | 71.62 | 67.61 | 62.22 |
| 1 | MMGIC | 7.36 | **31.57** | **72.24** | **35.33** | **66.10** |
| 2 | MMC4-Pairs (Zhu et al., 2023b) | **16.55** | **29.32** | **69.13** | 101.68 | 60.43 |
| 3 | MMC4 (Zhu et al., 2023b) | 17.81 | 28.68 | 68.48 | **40.31** | **66.00** |

## C.5 MMGIC AS INSTRUCTION FINE-TUNING DATA

If we regard the template text of our structured template as multiple instructions and multi-grained concept annotations as responses to the instructions, then MMGIC can be regarded as image–text interleaved instruction fine-tuning data. It can be directly used in the SFT stage of existing MLLMs pre-trained with image–caption data only. As shown in Table 6, the 0-th row corresponds to the performance of MLLM-IC, an MLLM that pre-trained with only image–caption data, IC, and fine-tuned with our default SFT data in Section 3.3 (1.21M samples from public instruction datasets and 1M samples from an aesthetic image–caption dataset). To explore the potential of MMGIC directly as instruction fine-tuning data, we add 1M samples from MMGIC to the SFT stage of MLLM-IC. MLLM-IC+MMGIC in the 1-th row achieves better performance on all benchmarks, especially on the benchmarks that deeply inspect common concrete concepts, *e.g.*, COCO, NoCaps, POPE, SEED-Bench, MS-COCO-30K, and VIST. This demonstrates MMGIC can be used not only as pre-training data, but also as instruction fine-tuning data to help MLLMs learn concepts better, which further ensure the generality of MMGIC across different MLLM frameworks.

## C.6 IMPACT OF THE NATURE OF IMAGE–TEXT INTERLEAVED ON VIST

Unlike MS-COCO-30K (Chen et al., 2015) that generates a new image based on a single caption, VIST (Huang et al., 2016) (Visual Storytelling) is a multimodal generation benchmark that requires MLLMs to generate a new image based on interleaved image–caption context within the same story. Compared with image–caption data, MMGIC has the nature of image–text interleaved, which may help MLLMs better understand the image–text context in VIST. Hence, in Table 7, we compare the performance of MMGIC(C) and MMGIC ($\{0, 3\}$-th data recipes in Table 1) on both MS-COCO-30K for simple text-to-image generation and VIST for in-context image synthesis. In addition, we also pre-train MLLMs with MMC4[2] (Zhu et al., 2023b) and MMC4-Pairs (split each MMC4 document into multiple individual image–text pairs) to ablate the impact of multi-grained concept annotations.

Comparing the top two rows in Table 7, after introducing the nature of image–text interleaved into our MMGIC, we find that most metrics increase, especially on VIST. However, for MMC4-Pairs and MMC4 in the bottom two rows, in-context image synthesis performance remains increases but text-to-image generation performance decreases. The results demonstrate that the nature of image–text interleaved can greatly benefit in-context image synthesis but may not be beneficial for simple text-to-image generation. We hypothesize that the multi-grained concept annotations in our MMGIC can work with the nature of image–text interleaved to help MLLMs better learn concepts and utilize concepts to align vision and language, thus improving performance on both benchmarks.

## C.7 EVALUATION ON Q-BENCH AND HALLUSIONBENCH

In this paper, we mainly focus on benchmarks that inspect the comprehension and generation capabilities of MLLMs on concrete concepts. Here, we further evaluate on two new benchmarks,

---

[2]We use a subset of MMC4, *i.e.*, MMC4-core-fewer-faces, which has a moderate sample size of 22.4M images and 5.5M image–text interleaved training samples, which is similar to the sample size of our MMGIC. Following the processing pipeline of MM-Interleaved (Tian et al., 2024), we process MMC4, and then convert each data sample into a discrete token sequence similar to our MMGIC.

Table 8: Zero-shot evaluation on Q-Bench and HallusionBench after SFT.

| | Model | Q-Bench | HallusionBench |
|---|---|---|---|
| a | LLaVA-1.5-7B (Liu et al., 2023b) | 58.70 | 27.60 |
| b | SEED-LLaMA-I-8B (Ge et al., 2023b) | 47.22 | 32.26 |
| 0 | MLLM-IC | 56.66 | 32.58 |
| 1 | MLLM-MMGIC | 57.59 | 30.37 |
| 2 | MLLM-MMGIC & IC | **58.26** | **33.39** |

Table 9: Evaluation on the MMLU (Hendrycks et al., 2021) benchmark with standard 5-shot setting and Accuracy as the metric.

| | Training Setting | MMLU |
|---|---|---|
| 0 | LLaMA-2-7B-base | 46.22 |
| 1 | + MMGIC | 29.98 |
| 2 | + MMGIC + SFT | **46.50** |

Q-Bench (Wu et al., 2023a) and HallusionBench (Liu et al., 2023a). They focus on low-level vision abilities (clearness, brightness, etc.), visual illusion and language hallucination (especially abstract concepts), respectively. As shown in Table 8, we can see that for the low level vision abilities in Q-Bench, although MMGIC mainly focus on objects, attributes and relations in images, it still helps MLLM achieve better performance than IC. We attribute this to the fact that concepts like clearness and brightness are concrete concepts that can also benefit from multi-grained concept annotations (*e.g.*, captions, label descriptions ) in MMGIC. As for HallusionBench, IC achieves significantly better performance than MMGIC since it contains more concepts, especially abstract concepts. Most importantly, the combination of MMGIC and IC can achieve the best performance on both benchmarks, which demonstrates that their combination can effectively integrate the strengths of both in terms of depth and breadth of concept representation, thus improving performance on both benchmarks.

## C.8 TEXT-ONLY PERFORMANCE

We follow common practice (Sun et al., 2023c; Ge et al., 2023b; Jin et al., 2023) to evaluate the zero-shot performance of MLLMs on multimodal comprehension and generation benchmarks. Notable, some recent works also explored the text-only performance of MLLMs. For example, LaVIT (Jin et al., 2023) and MM1 (McKinzie et al., 2024) found that training MLLMs only with multimodal data could result in a significant degradation of the text-only performance. To address this issue, they mixed a substantial amount of text-only data, *e.g.*, 66% text-only and 33% multimodal data, thus preserving the original text capability of the model. Besides, VILA (Lin et al., 2023) observed that while multimodal pre-training will hurt text-only performance, by simply incorporating some high-quality text-only instruction data during the SFT stage, the model can achieve similar text-only performance to the initial LLM.

To improve efficiency, we follow the approach of VILA to incorporate text-only instruction data, *i.e.*, ShareGPT(Steven Tey, 2023) and Alpaca (Taori et al., 2023), into our default SFT data in Section 3.3. We also conduct experiments on the de-facto text-only comprehensive multi-choice benchmark, MMLU (Hendrycks et al., 2021), to quickly evaluate the text-only performance of our models. As shown in Table C.8, we can see that there is indeed a significant performance drop after pre-training with MMGIC. However, after SFT, the MMLU performance of the model recovers well and is similar to the original LLM.

## C.9 ATTENTION MAP ANALYSIS

In Section 4.4, Figure 5 and 9, we provide qualitative analysis of the impact of different-grained concept annotations in MMGIC on the performance of MLLMs. To further understand the behavior of MLLMs trained with different-grained concept annotations, we visualize the attention maps of the

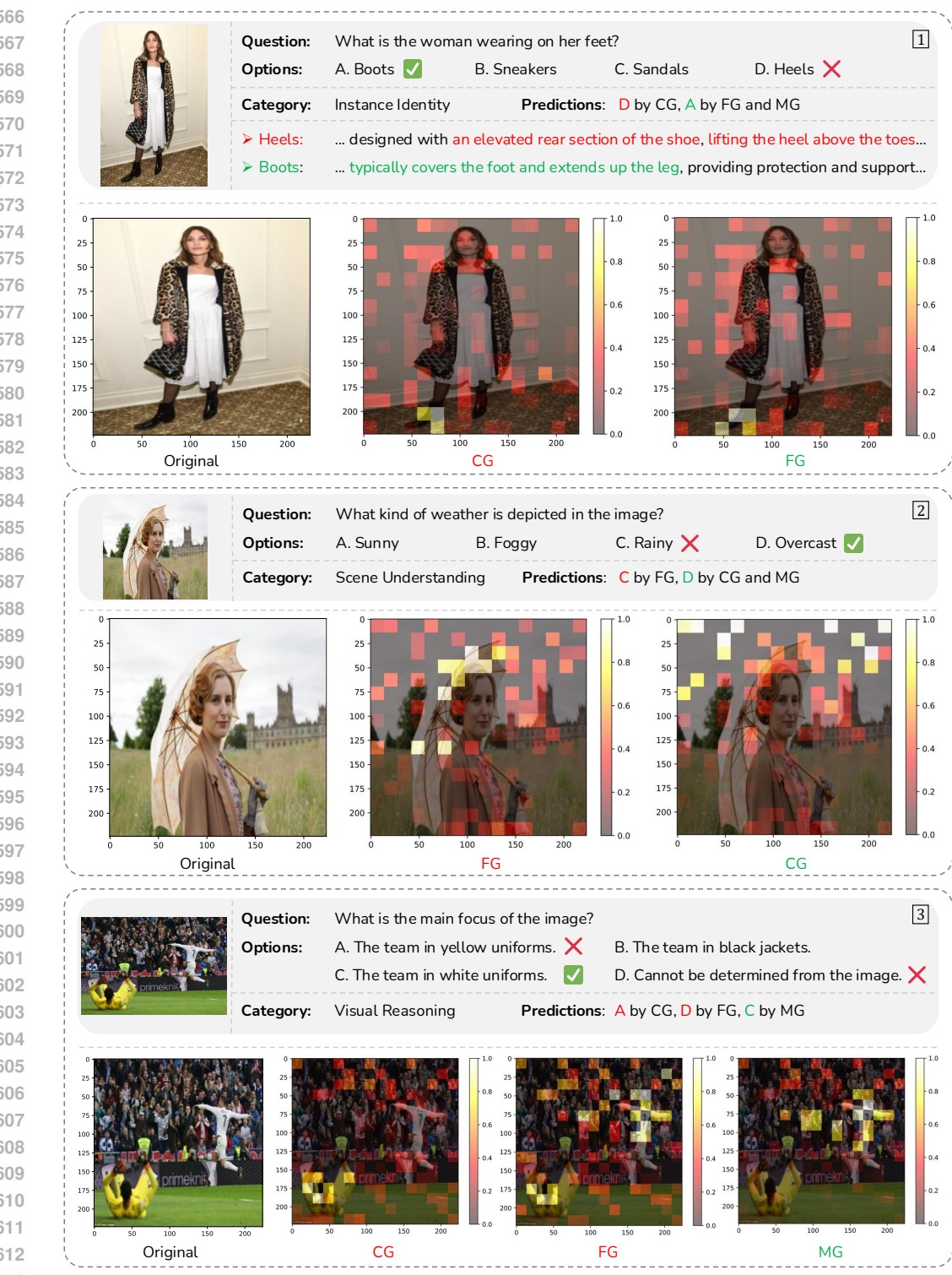

Figure 10: Attention map of cases on SEED-Bench-IMG from MLLM-MMGIC trained with different-grained concept annotations in MMGIC. CG, FG, and MG denote MLLMs trained with coarse-, fine-, and multi-grained concept annotations from MMGIC, respectively. Cases are the same as in Figure 5. ✅ denote the ground truth; ✗ denote incorrect prediction(s). The bottom of each example consists of an original image and two or three attention maps that shows the attention from the predicted answer token to the image tokens. In each attention map, the brighter the color, the higher the attention weight. The image patch with only gray shading means that it is not selected by the visual tokenizer (inherited from LaVIT (Jin et al., 2023)).

cases of Figure 5 on SEED-Bench-IMG, which shows the attention from the predicted answer token to the image tokens. In Figure 10, we can see that:

- Example $\boxed{1}$ "Instance Identity": Compared with CG (only focusing on the heel), FG can better understand the differences between different types of shoes in the options, to allocate attention more accurately to the image tokens related to shoes (not only the heel but also the toe and the upper part of the shoe), achieving the correct answer by capture the **local details** of the shoes.
- Example $\boxed{2}$ "Scene Understanding": Compared with FG (only focusing on the umbrella), CG can better understand the overall scene of the image, to allocate attention more accurately to the image tokens related to the sky and the clouds, achieving the correct answer by better understanding the **global context**.
- Example $\boxed{3}$ "Visual Reasoning": CG seems to be too focused on the player in the yellow uniform, while FG seems to be distracted by the audience, the player in the white uniform and the player the yellow uniform. However, MG can better combine the advantages of CG and FG, to allocate attention more accurately to the image tokens related to the player in the white uniform and the surrounding audience, achieving the correct answer (since the audience is cheering and taking pictures of the player in the white uniform) by better understanding and reasoning about the image from both **global context** and **local details**.

## D  DISCUSSION

### D.1  DIFFERENCE BETWEEN OUR FRAMEWORK WITH EXISTING MLLM FRAMEWORKS

This paper mainly focuses on MLLMs that are multimodal generalists capable of both multimodal comprehension and generation. Existing MLLMs (Sun et al., 2023c; Jin et al., 2023; Ge et al., 2023b; Zhu et al., 2023a; Sun et al., 2023a; Yu et al., 2023a; Lu et al., 2022a; 2024a) are typically trained with an autoregressive objective, *i.e.*, generate the prediction of the next token in a multimodal sequence containing discrete textual tokens, and discrete visual tokens or continuous visual embeddings. MLLMs process images in different ways. For example, EMU (Sun et al., 2023c;a) and VL-GPT (Zhu et al., 2023a) transform images into continuous visual embeddings by visual encoders, and then adopt a separate regression head to predict visual embeddings. In contrast, CM3Leon (Yu et al., 2023a), Unified-IO (Lu et al., 2022a; 2024a) and SEED-LLaMA (Ge et al., 2023b) tokenize images into discrete visual tokens by visual tokenizers, and then predict visual tokens by the extended vocabulary with the unified language model head. Technically, different training datasets, training settings (*e.g.*, full- or partial-param), framework, evaluation settings (*e.g.*, image resolution), etc., lead to non-comparable and unfair comparisons of our baselines with existing MLLMs. Their computation (more training data and fully fine-tune the LLM params) and data resources are extremely expensive and large (well over 10 times that of our work). Our autoregressive discrete framework for MLLMs is based on SEED-LLaMA and LaVIT, which consists of several visual modules and a LLM with an extended vision–language vocabulary, as shown in Figure 2. Next, we describe the differences between our framework and these two MLLMs.

- **Visual Modules:** our visual modules are inherited from LaVIT (Jin et al., 2023), which consists of a visual encoder, a visual tokenizer, a visual decoder and a diffusion model. The differences between the two MLLMs mainly lie in the design of the latter three modules. The visual tokenizer of SEED-LLaMA uses the Casual Q-Former from BLIP-2 (Li et al., 2023b) to compress visual representations into a fixed-length sequence of visual embeddings, while LaVIT is more flexible and designs a dynamic visual tokenizer with token selector and token merger, which can obtain a more flexible and effective sequence of visual embeddings. For the discrete visual token sequence predicted by the LLM, SEED-LLaMA uses an MLP as the visual decoder to compress and reconstruct it into the image embedding (1 token) and directly generate images through the frozen unCLIP-SD, while LaVIT uses multiple transformer blocks as the visual decoder to reconstruct the discrete visual token sequence into a continuous feature map by learned queries (256 tokens), and then uses the conditional denoising U-Net to progressively recover image pixels from a Gaussian noise with the feature map as the condition, which can retain more image information generated by the LLM, thus achieving better image generation results. We freeze all visual modules and pre-tokenize images into discrete visual token sequences, and only load vision modules to generate actual images when evaluate on downstream image generation tasks. This can greatly reduce the training cost and improve efficiency. Notably, the vision modules in

our framework, though inherits from LaVIT, only performs "Stage 1 Tokenizer Training," while LaVIT's vision module also performs "Stage 2: Unified Vision–Language Pre-training."

- **Extended VL Vocabulary:** as we stated in Section 3.1, SEED-LLaMA and LaVIT directly learn new visual token embeddings and initialize them with the distribution of original textual token embeddings. Visual tokens in the VL vocabulary correspond one-to-one with visual latent codes in the visual codebook. Considering the large difference in embedding dimensions between visual token embeddings in the VL vocabulary (4096) and visual latent codes in the visual codebook (32), we follow the factorized embedding parameterization in ALBERT (Lan et al., 2020) to replace visual token embeddings $|V| \times 4096$ with two smaller embedding matrices $|V| \times 32, 32 \times 4096$, where $|V|$ is the number of visual tokens in the VL vocabulary. The former matrix is directly initialized with visual latent codes in the visual codebook. In our preliminary experiments, we found that our initialization method of the VL vocabulary can significantly improve the performance of multimodal comprehension tasks, and has similar performance on multimodal generation tasks.

- **Training Objective:** SEED-LLaMA and our framework both treat the multimodal sequence as a discrete sequence of image–text interleaved tokens, and train the LLM with an autoregressive objective to generate predictions of the next token. To mitigate the loss of detailed information caused by vector quantization, LaVIT designs two different training objectives for image-to-text and text-to-image cases. For image-to-text, LaVIT is similar to EMU and uses the continuous visual features of images as the condition to predict all discrete textual tokens and calculate loss for textual tokens only; for text-to-image, LaVIT is similar to SEED-LLaMA and our framework, which tokenizes images into discrete visual token sequences and calculates loss for both textual and visual tokens. To pursue simplicity, efficiency, and scalability, our framework is similar to SEED-LLaMA, which tokenizes images into discrete visual token sequences, so that different types of multimodal training data can be converted into discrete sequences of image–text interleaved tokens, and trained with a unified autoregressive objective to fully learn the extended vision–language vocabulary of the LLM.

- **Training Data and Settings:** For our framework, we use MMGiC with 4M samples and IC with 52M image–caption pairs for pre-training; we use 1.21M multimodal instruction samples, 1M aesthetic image–caption pairs and 1M samples from MMGiC for supervised fine-tuning; during pre-training and SFT stages, our framework freezes all parameters of vision modules and most parameters of the LLM, and only tunes partial parameters of the LLM (<8%): the extended VL vocabulary, additional LoRA modules, norm layers and a language model head layer. Existing MLLMs typically use more pre-training and fine-tuning data, and fully fine-tune all LLM parameters and partial vision module parameters. Take SEED-LLaMA as an example, uses more pre-training data (∼770M vs <60M), SFT data (∼145M vs <4M), and full-param fine-tuning. Its computation and data resources are extremely expensive and large (well over 10 times that of our work). As shown in Table 3 and 4, compared with SEED-LLaMA, our baseline MLLM-MMGiC & IC requires significantly less data (∼1/15) and partial-param fine-tuning (<8%), achieving significant advantages on most benchmarks and comparable performance on COCO and VizWiz. Comparing with LaVIT, even though our framework does not convert images into continuous visual features to trade off lower loss of image representation and better multimodal comprehension performance, we still achieve significant advantages on three VQA benchmarks, and comparable performance on other benchmarks.

It should be emphasized that it is difficult to compare our baselines with existing MLLMs fairly, and the purpose of this paper is not to develop new frameworks, training objectives or benchmark SOTAs, but to explore the potential of multi-grained concept annotations for MLLMs. To this end, we explore different data recipes for multi-grained concept annotations and investigate the comparison and collaboration between MMGiC with widely-used coarse-grained image–caption data. Evaluations and in-depth analyses on 12 benchmarks for multimodal comprehension and generation are conducted in both pre-training and supervised fine-tuning stages.

**Comparison with Grounding MLLMs.** Fine-grained object region annotations are crucial for grounding VLMs/MLLMs. They usually convert bounding box coordinates into discrete tokens in text or visual markers in images. Take KOSMOS-2 (Peng et al., 2023) as an example, it falls into the former category, utilizing object bounding box coordinates through special discrete position tokens placed after corresponding texts in the caption. Unlike KOSMOS-2, MMGiC converts object bounding box coordinates into corresponding object regions (cropped from images), transforming

pure textual object-level annotations into multimodal annotations. Structured templates integrate different granularities into unified image–text interleaved documents, enhancing MLLM's ability to locate concepts in textual annotations to corresponding regions in images. With the unique complex context processing capabilities of MLLMs and the LM autoregressive loss, our framework can explicitly utilize multi-grained annotations in MMGIC to optimize multimodal alignment across multiple granularities simultaneously, improving concept understanding and generation.

## D.2  APPLICABILITY OF MMGIC TO EXISTING MLLMs

Our framework combines the advantages of LaVIT and SEED-LLaMA to allow efficient exploration on various multimodal tasks, using only LM autoregressive loss without extra modules and loss functions for MMGIC, maintaining the framework's generality and efficiency. Compared to the current image–text pairs or image–text interleaved documents used by MLLMs, MMGIC can be seen as image–text interleaved documents with textual (caption, label, label description) and visual (object region) multi-grained concept annotations. They are well-integrated through our carefully designed structured templates, forming image–text interleaved documents that can be directly used by various MLLMs. Therefore, both our MMGIC dataset and our framework exhibit good generality. MMGIC do not impose any requirements or limitations on the model architecture or loss function of MLLMs. It can be directly applied to various MLLMs, whether they process images as continuous vectors (e.g., EMU, LaVIT, VL-GPT) or discrete tokens (e.g., CM3Leon, SEED-LLaMA, Chameleon). Our experiments on 12 multimodal comprehension and generation benchmarks also ensure the reliability and credibility of our conclusions, making sure that MMGIC can be quickly applied to other MLLMs' training. Overall, our model framework and training loss follow CM3Leon and SEED-LLaMA's spirit of unified discrete sequence modeling, offering simplicity, efficiency, and scalability. We did not make any special architectural or loss design modifications for MLLMs, ensuring good generalization and reliability of our framework, training loss, dataset, and experimental explorations.

Moreover, our focus is to explore the potential of multi-grained concept annotations for MLLMs. Directly fine-tuning existing SOTA MLLMs cannot fully demonstrate this point. Thus, we explore comparison and collaboration between MMGIC and IC starting from the multimodal pre-training phase. Unfortunately, all SOTA MLLMs use the LAION-5B dataset (and its subsets) for pre-training. However, the official data download channels were closed by the LAION team due to safety reviews (from December 19, 2023). This prevents us from retraining existing SOTA MLLMs from the pre-training stage in a controlled environment to directly prove MMGIC's superiority over IC. Therefore, we collected ∼250M image–caption pairs from other widely used and recognized datasets and selected ∼52M as the IC dataset for our experiments. Through the evaluation of three baselines across 12 multimodal understanding and generation benchmarks, we demonstrate the potential of MMGIC for MLLMs. Combining MMGIC and IC can leverage their respective advantages in terms of depth and breadth of concept representation, further improving MLLM performance.

## D.3  ANNOTATION SYNTHESIS IN MMGIC

MMGIC is constructed from several public human-annotated object detection datasets, which covers many common concepts but currently cannot scale up to cover more concepts in the real world. We only automatically synthesize image captions for all images with the de-facto captioning model BLIP-2 (Li et al., 2023b) and ranking model CLIP (Radford et al., 2021), and synthesize label descriptions for all labels with the strong GPT-4 (Achiam et al., 2023). For synthesized captions, we randomly sample 1000 captions to manually check the quality, and find that the quality is acceptable. For synthesized label descriptions, we carefully design prompt templates and human-annotated in-context examples, and manually check all synthesized label descriptions to ensure the quality. Besides, with the help of WordNet, we check category labels, and update the polysemous labels based on the specific data samples for better differentiation, *e.g.*, "batter" → "batter (ballplayer)" or "batter (cooking)".

In this paper, we want to use and check existing human-annotated annotations as much as possible, and minimize the automatic generation of annotations, thus reducing noise that interferes with our exploration of multi-grained concept annotations for MLLMs. Hence, we take the above strategies to ensure the quality of synthesized annotations (captions and label descriptions). Moreover, in MMGIC, fine-grained attribute and relationship labels are only available in partial images from the original datasets. Although we try to automatically generate fine-grained attributes and relationship

annotations with external open-source models based on image captions (and images), after careful manual checks, the synthesized fine-grained annotations are found to be of low quality, with non-negligible hallucinations and noise, which are very likely to negatively impact MLLMs. Thus, only high-quality manually annotated fine-grained annotations were used in MMGIC to avoid unnecessary hallucinations and noise. We focus our limited resources and effort on carefully handling existing data with meticulous manual checks and in-depth experimental exploration to ensure the quality of MMGIC and the reliability of our exploration.

To further ensure that such partially missing fine-grained annotations do not introduce additional bias, we compare MMGIC with only object labels and label descriptions, as well as MMGIC with all category labels and label descriptions, *i.e.*, MMGIC(CLD). We find that the performance is still improved by the partially available fine-grained attribute and relationship labels. More importantly, as shown in Section 4.4 and Figure 5 *(Left)*, compared to MMGIC with only image captions, multi-grained concept annotations in MMGIC can still help MLLMs achieve significant improvements of 2.15 and 10.31 points in the corresponding "Instance Attribute" and "Instance Interaction" dimensions, respectively. Similar results are also observed in Appendix C.3 and Figure 8, where MLLMs trained with MMGIC can self-generate fine-grained category labels during the inference stage to further improve the performance by 1.79 and 5.16 points in the corresponding "Instance Attribute" and "Instance Interaction" dimensions, respectively. Above experimental results demonstrate that partially missing fine-grained attribute and relationship labels do not affect the effectiveness of MMGIC and the reliability of our exploration.

## D.4 DIFFERENCE BETWEEN MMGIC AND EXISTING IMAGE–TEXT INTERLEAVED DATASETS

MMC4 (Zhu et al., 2023b) and OBELICS (Laurençon et al., 2024) are two widely-used open-source large-scale image–text interleaved datasets. They are constructed from massive HTML documents. MINT-1T (Awadalla et al., 2024) further enhances the scale and diversity, uniquely including data from PDFs and ArXiv documents. These datasets are large-scale, diverse, and also noisy. They are important for MLLMs to reason across image and text modalities, and have been used to further improve the capabilities and performance of MLLMs, especially in image–text interleaved scenarios. Our proposed MMGIC can also be seen as image–text interleaved documents with higher-quality and better image–text alignment, which has multi-grained concept annotations to help MLLMs better locate and learn concepts, thereby promoting vision–language alignment across multiple granularities simultaneously. It is a promising direction to automatically synthesize multi-grained concept annotations for these web-scale datasets.

## D.5 IMAGE-INDEPENDENT LABEL DESCRIPTION GENERATION

In Section 2.2 and Appendix F.5, we describe the process of generating label descriptions for category labels in MMGIC. Considering that our description generator, GPT-4, do not see the images both during training and annotation synthesis, such general image-independent generation of label descriptions may introduce hallucinations and noise. In this paper, to ensure the quality of synthesized label descriptions and avoid hallucinations, we particularly consider the following aspects:

- **Effectiveness of Synthesized Label Descriptions.** As mentioned in Section 2.2, many previous works use WordNet and LLM to obtain general image-independent label descriptions, and successfully improve the model's understanding of concepts corresponding to these labels. The experimental results in Table 1 of Section 4.1 also show that label descriptions can bring significant performance improvements.

- **Feasible Synthesis for Common Concepts.** Whether in previous works (Menon & Vondrick, 2023) or in the four open-source datasets used in this paper, since the target labels are **common** concepts, LLM can generate label descriptions with good generality and rich visual information even without seeing the image.

- **Disambiguate Polysemous Labels.** We also use WordNet to disambiguate the polysemous labels in MMGIC, such as "batter" → "batter (ballplayer)" or "batter (cooking)", to further reduce the ambiguity in the label descriptions, improve the generality and quality.

- **Careful Quality Control.** We not only carefully manual-check all generated label descriptions to ensure their quality and avoid introducing additional noise or hallucinations, but also carefully

Table 10: Training hyperparameters and cost for both pre-training and supervised fine-tuning (SFT) stages. MMGIC and IC refer to two pre-training datasets.

| Training hyperparameters | | | |
|---|---|---|---|
| Training Data | MMGIC | IC | SFT Data |
| Optimizer | AdamW | AdamW | AdamW |
| Learning Rate | 2e-4 | 2e-4 | 4e-4 |
| Weight Decay | 0.05 | 0.05 | 0.05 |
| Training Epochs | 1 | 1 | 1 |
| Warmup Ratio | 0.1 | 0.1 | 0.05 |
| Min learning rate ratio | 0.1 | 0.1 | 0.1 |
| Learning Rate Scheduler | Cosine | Cosine | Cosine |
| Batch Size | 512 | 512 | 256 |
| Maximum Token Length | 2048 | 2048 | 2048 |
| Training Cost | | | |
| GPU Device | 32×NVIDIA A100-80G | | |
| Training Time | ~11h | ~20h | ~5h |

design prompt templates and manual annotation examples (in Appendix F.5) to guide and improve the quality of the annotations generated by LLM. Interestingly, we find that the powerful GPT-4 can even automatically identify invalid noisy labels in the labels with the customized prompt templates, helping us to clean up these invalid annotations.

# E EXPERIMENTAL DETAILS

## E.1 TRAINING DETAILS

The training hyperparameters and cost are shown in Table 10.

**Training Loss.** Similar to LLMs (Brown et al., 2020; Touvron et al., 2023b) and MLLMs (Ge et al., 2023b; Jin et al., 2023) under the autoregressive discrete framework, as we discussed in Section 3.2, we train our MLLMs with an autoregressive objective to maximize the likelihood of predicting the next visual or textual token in a multimodal discrete token sequence as follows:

$$L = \sum_{i=1}^{|u|} \log P\left(u_i \mid u_1, \ldots, u_{i-1}\right)$$

where $u$ denotes the multimodal token sequence, and $u_i$ denotes the $i$-th token in the sequence. We can divide the multimodal token sequence into two parts: visual tokens and textual tokens. In the MLLM training, we find that the loss of the visual tokens $L_V$ is larger than the loss of the textual tokens $L_T$. To balance the loss of the visual and textual tokens, we introduce a loss scale ratio $\alpha = 0.1$ to scale the visual token loss. The final loss is defined as $L = \alpha \cdot L_V + L_T$.

**Image-First and Text-First.** Our pre-training data consists of image–caption pairs (IC) and image–text interleaved documents (MMGIC). We follow common practice (Alayrac et al., 2022) to randomly select the image-first or text-first data template for each data sample as shown in Section F.6. The difference between the two templates is the order of appearance of the image and text in the sample, which determines the condition of each token during autoregressive training. For the multimodal instruction data we collect from public datasets, we follow their original data formats and do not change the order of appearance of the image and text in the sample.

**Training Parameters.** In all training stages, we only tune partial parameters of the LLM: the VL vocabulary, additional LoRA modules (Hu et al., 2021), norm layers and a language model head

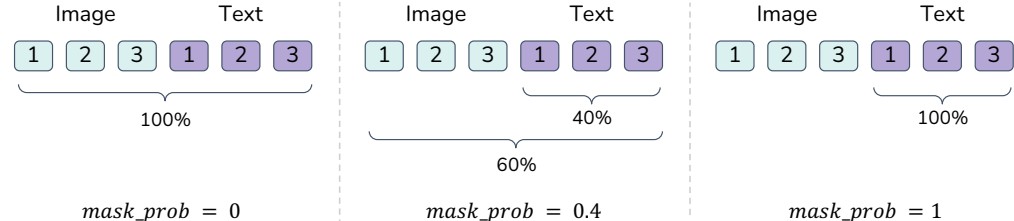

Figure 11: A brief illustration of the hyperparameter $mask\_prob$ in the pre-training stage when the template is image-first.

Table 11: Evaluation benchmarks, eval splits and eval metrics for the pre-training stage.

| Dataset | Task Description | Eval Split | Metric |
|---------|------------------|------------|--------|
| COCO | Scene description | `test` | CIDEr |
| NoCaps | Scene description | `test` | CIDEr |
| MS-COCO | Text-Conditional Image Synthesis | `val (30K)` | FID, CLIP-I, CLIP-T |
| VIST | Contextual image synthesis | `val` | FID, CLIP-I |

layer, to greatly improve efficiency. All data are pre-tokenized, and we don't load all visual modules during training.

**Loss Calculation Strategy.** Moreover, in SFT stage, we follow LLaVA (Liu et al., 2023c) to only calculate loss for the answer tokens. In the pre-training stage, we find that existing MLLMs typically calculate loss for all tokens or only for the textual tokens when visual tokens as the condition, and vise versa. We conduct experiments to investigate the impact of different loss calculation strategies on the performance of MLLMs. Take the image-first template as an example, we introduce a hyperparameter $mask\_prob$ to control the probability of only computing the loss for the textual tokens in each sample. As shown in Figure 11, when $mask\_prob = 0$, we calculate the loss for all tokens in each sample; when $mask\_prob = 0.4$, we calculate the loss for the textual tokens only in $40\%$ of the samples, and calculate the loss for all tokens in the rest $60\%$ of the samples; when $mask\_prob = 1$, we calculate the loss for the textual tokens only in all samples. $mask\_prob$ can be seen as a sample-level mask ratio of modality tokens, which controls the difficulty of the pre-training task. Higher $mask\_prob$ means easier pre-training task since more tokens are masked out. We conduct experiments with $0.1$ as the step size to investigate the impact of $mask\_prob = [0, 1]$ on the performance of MLLMs on image captioning and image generation benchmarks. Interestingly, the situation is more complicated than we thought. For image captioning tasks, the performance increases with the increase of $mask\_prob$, but the performance suddenly drops at $mask\_prob = 1.0$, even lower than the performance at $mask\_prob = 0.0$. For image generation tasks, the performance decreases first and then increases with the increase of $mask\_prob$ (the inflection point is $mask\_prob = 0.5$, the performance is better than that at $mask\_prob = 0.0$), and then decreases again (the inflection point is $mask\_prob = 0.8$). To balance the performance of comprehension and generation tasks, we choose $mask\_prob = 0.9$ as the final hyperparameter. Since this is not the focus of this paper, we do not conduct in-depth research and analysis on this, but we believe that this is an interesting phenomenon worthy of further research, and we encourage future researchers to conduct in-depth research on this.

E.2 EVALUATION DETAILS

**Evaluation Benchmarks and Metrics.** We show the evaluation benchmarks and their corresponding data splits and metrics in Table 11 and Table 12 for the pre-training and supervised fine-tuning stages, respectively.

**Evaluation Strategy.** In this paper, we evaluate the performance of MLLMs on both multimodal comprehension and generation tasks, including image captioning, text-to-image generation, visual

Table 12: Evaluation benchmarks, eval splits and eval metrics after SFT.

| Dataset | Task Description | Eval Split | Metric |
|---------|------------------|------------|--------|
| COCO | Scene description | test | CIDEr |
| NoCaps | Scene description | test | CIDEr |
| VQAv2 | Scene understanding QA | test | VQA Acc |
| GQA | Scene understanding QA | test | VQA Acc |
| VizWiz | Scene understanding QA | test | VQA Acc |
| POPE | Visual Hallucination | - | F1-score |
| MME | Multimodal Comprehension | test | MME score |
| MMBench | Multimodal Comprehension | dev | Acc |
| ScienceQA-IMG | Multimodal Comprehension | test | Acc |
| SEED-Bench-IMG | Multimodal Comprehension | test | Acc |
| MS-COCO | Text-Conditional Image Synthesis | val (30K) | FID, CLIP-I, CLIP-T |
| VIST | Contextual Image Synthesis | val | FID, CLIP-I |
| DreamBench | Subject-driven Image Editing | From KOSMOS-G | DINO, CLIP-I, CLIP-T |

question answering, multi-choice benchmarks, and image editing. We follow standard evaluation strategies and dataset splits to ensure the comparability of our experimental results. We implement the generation of visual and textual tokens in the evaluation stage based on the vLLM library, and generate the final image through the Diffusers library. Notably, multi-choice benchmarks have different designs and implementations in task instruction templates and evaluation strategies. The impact of different task instruction templates on the model's performance is significant and cannot be ignored. Therefore, we manually design at least 8 task instruction templates for each dataset and select the best performance as the final result to ensure the stability of the experimental results. The evaluation strategy is more complicated, and some benchmarks, such as MMBench (Liu et al., 2023d), also introduce an external LLM (GPT-4 (Achiam et al., 2023)) as the choice extractor. To pursue simplicity and efficiency, we directly select the highest probability of the candidate option tokens, *i.e.*, "Yes" and "No" for POPE (Li et al., 2023c), "A", "B", "C", "D", *etc*. for other benchmarks, as the final answer. We don't introduce any external LLMs in our experiments since it will increase the cost and leads to unstable results (Liu et al., 2023a) compared with the above strategy. Similarly, we don't evaluate on benchmarks that use GPT-4 as the default evaluator, since it will increase the cost and has strong bias on models that tuned with GPT-4 synthetic data (Lin et al., 2023).

## F    DETAILS OF MMGIC DATASET

In this section, we provide the details for constructing MMGIC dataset. The statistics of MMGIC dataset are shown in Table 13.

### F.1    DATASET COLLECTION

We collect four large-scale object detection dataset, including Open Images (Kuznetsova et al., 2020), Objects365 (Shao et al., 2019), V3Det (Wang et al., 2023a) and Visual Genome (Krishna et al., 2017) for MMGIC.

BigDetection (Cai et al., 2022) has found that Open Images and Objects365 are biased towards certain scale (small or large) of objects, while the improved object annotations for these two datasets in BigDetection are balanced across object scales. Therefore, we use the improved object annotations of Open Images and Objects365 from BigDetection. For Open Images, we merge the BigDetection object annotations with the attribute and relationship annotations of Open Images. Specifically, for each object region in BigDetection, we calculate the $IoU$ with each Open Images object region in the same image, and select the Open Images object region with the largest $IoU$ to pair with it. We only keep the relationship and attribute annotations in Open Images where subject object and the object object are both successfully paired. We do not use the LVIS (Gupta et al., 2019) subset in BigDetection since all images in LVIS come from COCO (Lin et al., 2014), which is our downstream

Table 13: Statistics of MMGıC dataset and four object detection datasets underlying it. The last 4 columns are based on the final tokenized dataset and the other columns are before tokenization. The # Concepts column denotes the number of unique category label–description pairs in each dataset. Note that Visual Genome and V3Det are repeated 3 times to keep data balance in the final dataset.

| | Dataset | # Regions | # Images | # Concepts | # Visual Tokens | # Textual Tokens | # Used Regions | # Samples |
|---|---|---|---|---|---|---|---|---|
| 0 | Open Images | 7.16M | 1.44M | 642 | 0.70B | 0.47B | 6.21M | 1.44M |
| 1 | Objects365 | 15.71M | 1.78M | 600 | 1.31B | 0.97B | 12.58M | 1.78M |
| 2 | V3Det | 1.18M | 0.18M | 12,976 | 0.26B | 0.21B | 2.30M | 0.53M |
| 3 | Visual Genome | 1.50M | 0.10M | 20,830 | 0.28B | 0.28B | 2.77M | 0.31M |
| 4 | **MMGıC** | **25.55M** | **3.48M** | **35,048** | **2.54B** | **1.92B** | **23.86M** | **4.06M** |

evaluation dataset. For V3Det and Visual Genome, we use all their original annotations for further pre-processing.

Since BigDetection dataset does not involve any changes to the positions of the objects in the Open Images and the objects in the same image are a subset of the objects in the Open Images, the matching process is correct and appropriate. We use $IoU$ instead of exact matching because BigDetection uses coordinates to provide object positions, whereas Open Images use ratios, where using exact match may introduce errors.

For V3Det, there are 60 images that were not successfully downloaded, and we simply discarded these images.

### F.2 DATASET PRE-PROCESSING

**Open Images, Objects365 and V3Det.** There are 3 steps to pre-process these three datasets. Notice that we directly use the train and dev splits from the original datasets and don't use the test set.

1. **Clean labels**: to remove the noise that may be introduced by long-tail distribution, we count the number of occurrences of each label in each dataset, and then select a threshold to remove low-frequency labels (typically noise). For V3Det, we remove object labels with frequency less than 3. For Open Images and Objects365, we don't remove any label since each label appears many times and has high quality.

2. **Clean objects and object-related information**: we remove objects that have illegal coordinates, exceed the image range or the corresponding label is already removed. For Open Images, we also remove the attribute and relationship annotations corresponding to these illegal objects.

3. **Clean images**: since the de-facto visual encoder (*i.e.*, Vision Transformer, ViT) requires a square input image, we resize all images to a fixed resolution of $224 \times 224$ pixels. Following standard practices (Zeng et al., 2023), to improve the data quality, we first remove images with short edges less than 224 pixels or aspect ratios greater than 3.0 or less than 0.33, to prevent the image from changing too much after resizing. We also remove the images without any objects.

**Visual Genome.** Visual Genome is a dataset trying to connect structured image concepts to language. Each image is annotated with: 1) image-level info, *i.e.*, image captions; 2) region-level info, including objects in the region and the region description; 3) object-level info, including object label and location, the attributes of objects, the relationships between objects, with all the labels mapped to WordNet synsets. We use the image-level info and object-level info for further pre-processing which require the following steps. [3]

---

[3] We found that region-level annotations may not be suitable for constructing MMGıC since: 1.many regions only contains a single object; 2.the annotations of the objects in region may not be exactly in the region; 3.the region description may not be exactly matched with the region itself. Hence, We use $v1.4$ version of the dataset which includes image-level and object-level annotations except the attribute of objects. Since we found that the objects in $v1.2$ is the subset of that in $v1.4$, we use the attribute annotations in $v1.2$

1. **Clean images**: since all images need to be resized to $224 \times 224$ and input to the visual modules, we remove the images with the shorter side length less than $224$, images with an aspect ratio greater than 3 or less than $0.33$ to prevent the image from changing too much after resize.

2. **Clean object annotations**: we remove all objects that have more than one synset as we found that they are mostly noisy. We also remove objects that have illegal coordinates or exceed the image range.

3. **Construct, clean object label, clean object annotations again**: in Visual Genome, the same object label may have different semantics distinguished by the synsets they are mapped to. To mitigate the noise brought by long-tail distribution, firstly, we count the frequency of each (object label, synset) pair if the mapped synset is not empty. Secondly, for those object labels that aren't mapped to any synset, we regard that it is mapped to the synset that has the highest frequency of the same object label in the first step (if there isn't any same object label, we regard that it is mapped to the empty synset) and add its frequency to the first step. Finally, we remove all the (object label, synset) pairs with frequency less than 2 and also remove all the corresponding objects. To get the final object label name, for the object labels which have more than two types of synsets, we manually check them with help of WordNet and if so, we add some keywords corresponding to the synset behind the object label name as the final name for better differentiation (e.g. (batter, batter.n.01) $\rightarrow$ "batter (baseball player)", (batter, batter.n.02) $\rightarrow$ "batter (semiliquid mixture)"). For other object labels, we directly use the object label name as the final name.

4. **Clean images again**: we remove all the images that don't have any object annotation.

5. **Clean relationship and attribute annotations**: firstly, we remove all the relationships where the corresponding object is removed by step 2 and step 3. Secondly, we count the frequency of each label and remove the relationship/attribute whose frequency is less than 5 or is empty. We don't add additional information in label name as in step 3 as we have checked manually that each label name has exactly one semantic.

6. **Remove unreasonable labels and annotations**: for each label, we ask ChatGPT if it is an reasonable label in Section F.5. For unreasonable labels, we remove them and the corresponding object, attribute, relationship annotations.

7. **Dataset split**: for images in Visual Genome with a coco id, we follow Karpathy's split: images in 'val' and 'restval' are used as the validation set, and images in 'test' are discarded. For images in Visual Genome with a flickr id but no coco id, we match them with images from flickr30k and discard the matched images. All other images are used as the training set.

### F.3 OBJECT ANNOTATION PRE-PROCESSING

The fine-grained object annotations provided in these datasets include bounding box coordinates and category labels for each object. To accommodate varying aspect ratios of object bounding boxes and the requirement for a square input image, following KOSMOS-G (Pan et al., 2023), we crop a new larger square object region $S_i$ that contains the original object region $R_i$, $R_i \subseteq S_i$, with their centers aligned as closely as possible. Since $S_i$ may include not only the object region $R_i$ but also the surrounding object regions, we design a simple but effective strategy to update the object label annotations of $S_i$ by merging the object label annotations of $R_i$ with the object label annotations of surrounding object regions, to improve the quality of object annotations. For each $R_i$ and its corresponding new region $S_i$, we calculate the intersection-over-area (IoA) between $S_i$ and all $R_{j(j \neq i)}$, i.e., $IoA(S_i, R_j) = \frac{Area(S_i \cap R_j)}{Area(R_j)}$. If $IoA(S_i, R_j) \geq 0.8$, we consider $R_j$ as a part of $S_i$ and update the annotations of $R_j$ to the annotations of $S_i$. This strategy can help to include more relevant object regions in the region $S_i$ and improve the quality of object annotations. After that, we remove $S_i$ with duplicate annotations and keep the unique ones.

Since each visual token in our framework corresponds to a $14 \times 14$ pixel region, we remove $S_i$ with edge length less than $28$ pixels or greater than $182$ pixels to ensure that the object region is not too small or too large. We also remove all the images that don't have any bounding box and resize all images to $224 \times 224$ pixels. Finally, for objects in the image, MMGIC provides visual tokens of cropped regions and textual tokens of corresponding fine-grained category labels and location descriptions. The pre-processed object regions in the images are ready to be tokenized and fill in the template in Section F.6.

Specifically, instead of transform bounding box coordinates into textual tokens in the text (Chen et al., 2021; Liu et al., 2023c; Peng et al., 2023) or visual markers in the image (Yao et al., 2024; Shtedritski et al., 2023; Yang et al., 2023), for each object in the image, we directly provide visual tokens for the cropped region and textual tokens for the corresponding category labels. This not only upgrades our data from simple image–text pairs to more complex image–text interleaved data which has shown to be beneficial for vision–language learning (Lin et al., 2023; McKinzie et al., 2024), but also helps to locate and align concepts in the image and in the text by providing both the whole image and object regions.

### F.4   CAPTION SYNTHESIS

Since only partial images are annotated with image captions in these datasets, following BLIP (Li et al., 2022a) and LAION-COCO (Schuhmann et al., 2023), we automatically synthesize captions for all images in the following steps:

1. Generate 10 candidate captions for each image with BLIP-2 (Li et al., 2023b);

2. Rank candidate captions based on image–caption similarity scores calculated by CLIP (Radford et al., 2021);

3. Filter too short ($< 5$ words) or too long ($> 25$ words) captions or captions with low image–caption similarity scores ($< 0.25$);

4. Select the Top-1 caption as the final caption if exists, otherwise repeat the above steps for 10 times. If no caption is selected, we select the caption with the highest similarity score.

### F.5   LABEL DESCRIPTION GENERATION

Label descriptions are corresponding concept descriptions of concrete concepts in the image, which convey understanding about a concept by visually observed details and relevant knowledge. Inspired by the success of previous works (Shen et al., 2022; Yao et al., 2022; Menon & Vondrick, 2023) that introduces label descriptions from WordNet (Miller, 1992), Wiktionary (Meyer & Gurevych, 2012) and LLMs (Brown et al., 2020) to help understand concepts, with the help of GPT-4 (Achiam et al., 2023), we generate label descriptions for fine-grained category labels (object, object attribute, and relationship between objects) and manually check them. We design prompt templates for each type of category labels and carefully provided human-annotated examples. For V3Det, we directly use the label descriptions it provided. For Open Images, Objects365 and Visual Genome, we generate label descriptions for object labels, attribute labels and relationship labels respectively.

Specifically, in system prompt, we first describe the task and add some tips that LLM might focus when generating descriptions. Then we give some examples. In user prompt, we instruct the LLM to generate the description. We also ask LLM to generate 'Invalid' first if the given label is noisy in Visual Genome These labels along with their corresponding annotations will be further removed. The full prompts are shown in Table 17, 18, 19, 20, 21, 22.

### F.6   TEMPLATE

After completing all above steps, for each image sample, we have the following annotations:

- **Coarse-grained concept annotations**: captions and the localized narrative captions.[4]

- **Fine-grained concept annotations:** object labels, attribute labels and relationship labels along with their label descriptions.

- **Region-level annotations**: the object regions of the image obtained in Section F.3 along with their locations and object labels.

We design templates to formulate the above annotations into data samples used in the pre-training stage. There are 2 types of templates: image-first template shown in Table 23 and text-first template shown in Table 24. Each template has 2 parts: image-annotation part and object-annotation part.

---

[4]Only part of images in Open Images have localized narrative captions.

**Image-first template.** For image-annotation part or object-annotation part, we place the corresponding text **after** the image or region, aiming to enable the model to learn visual understanding.

- **Image part**: we first place the image. Secondly, we place the coarse-grained concept annotation including caption and the localized narrative caption. Thirdly, we place image-level fine-grained concept annotations including object labels, attribute labels and relationship labels along with their descriptions. **For object labels**, we first place the label name and then place the label description, each object label name in the image appears exactly once. **For attribute labels**, we first place the attribute label and then place the associated objects, followed by the attribute label description. **For relationship labels**, we first place the relationship label and then place the associated subjects and objects, followed by the relationship label description. [5]

- **Object-annotation part**: we place the object regions of the image with their location descriptions and associated object labels after them.

**Text-first template.** Different from the image-first template, we place the corresponding text **before** the image or region, aiming to enable MLLMs to learn visual generation. Note that we place the captions close to the images, to help MLLMs better learn to utilize caption to generate image.

### F.7 PRE-TRAINING SAMPLE GENERATION

**Pre-tokenize images and regions.** For images and regions, we directly use the visual tokenizer of LaVIT(Jin et al., 2023) to tokenize them into visual token sequences.

**Fill in the template and tokenize data.** There are 4 steps to generate a training sample. For V3Det and Visual Genome, we repeat the process 3 times to include as many regions as possible.

1. **Select a template**: we randomly select the image-first or text-first template with a probability of $0.5$.

2. **Fill in the template with textual annotations**: **For image-annotation part**, we first fill the corresponding text. Then we remove the descriptions with a probability of $0.5$ to prevent model overfitting which might be caused by description repetitions. **For each object-annotation**, we first fill in the region location. Specifically, the $224 \times 224$ square image is divided into a $3 \times 3$ grid, with each cell named sequentially from top to bottom and left to right as Top Left, Top Middle, Top Right, Middle Left, Center, Middle Right, Bottom Left, Bottom Middle, and Bottom Right. The location of a region is designated by the name of the grid cell where its center is located. Then we fill in the object label with the annotations obtained in Section F.3. **To get the final object-annotation part**, we shuffle the regions and place as many regions as possible without exceeding the maximum tokenized length.

3. **Tokenize the data and fill the template with visual annotations**: firstly, we tokenize the data using the tokenizer of with `[IMG]`, `[/IMG]` added as special tokens. Secondly, we replace the positions corresponding to `[IMG]` using the corresponding visual token sequence with `[IMG]` and `[/IMG]` inserted before and after it. Thirdly, we add `` token and `` token to the whole token sequence.

4. **Dealing with samples with token length more than** $2048$: for those samples with object regions, we remove one region a time and go back to step 2 until the token sequence length is less than 2048. For those samples without regions but with descriptions, we go back to step 2 but fill in the template without descriptions. Other samples are discarded.

**Statistics.** Table 13 shows the statistics of the constructed MMGIC. The frequency distribution of object labels, attribute labels and relationship labels are shown in Figure 12 respectively, which illustrates that our label types are broadly distributed. Moreover, only a few labels appear with particularly low frequency. For object labels, attribute labels, and relationship labels, labels with frequency less than 5 accounted for 5%, 1.5%, and 0.3% of the total number of their respective label types, respectively.

---

[5]We design the template in this way to reduce input length and duplication.

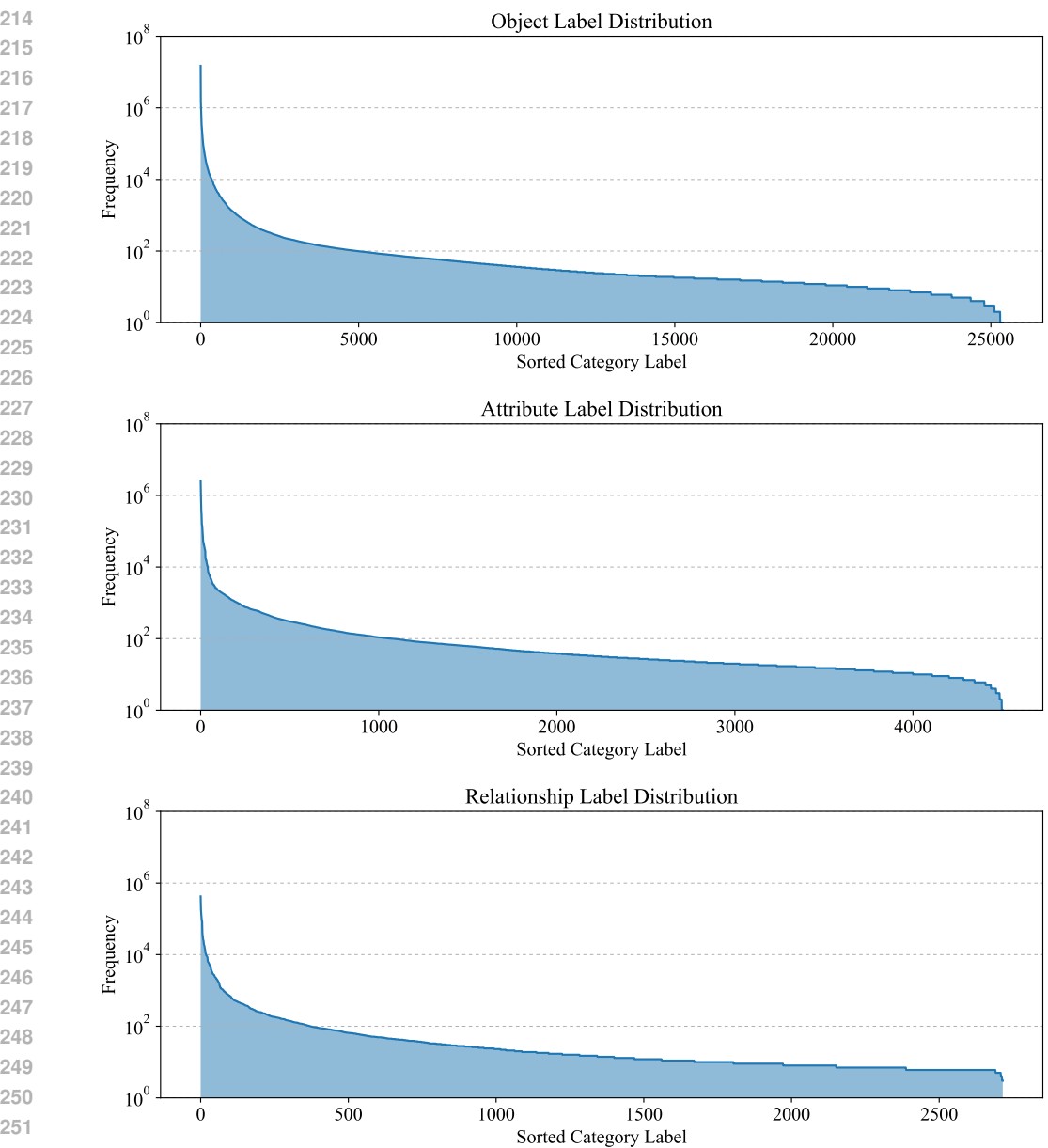

Figure 12: Label frequency distribution of objects, attributes and relationships in MMGıC.

## G  DETAILS OF IC DATASET

To strike a balance between increasing concept breadth, reducing data noise and improving efficiency, we collect several large-scale public image–caption datasets (Ordonez et al., 2011; Sharma et al., 2018; Changpinyo et al., 2021; Schuhmann et al., 2022a;b; Sun et al., 2024), and follow LLaVA (Liu et al., 2023c) to first filter them by the frequency of noun-phrases extracted by spaCy from their given synthesized captions, and then automatically synthesize captions same as MMGıC. We name this dataset as IC, where 52M unique images are collected and selected, almost 15 times more than MMGıC. IC contains 3.19B visual tokens and 5.14B textual tokens.

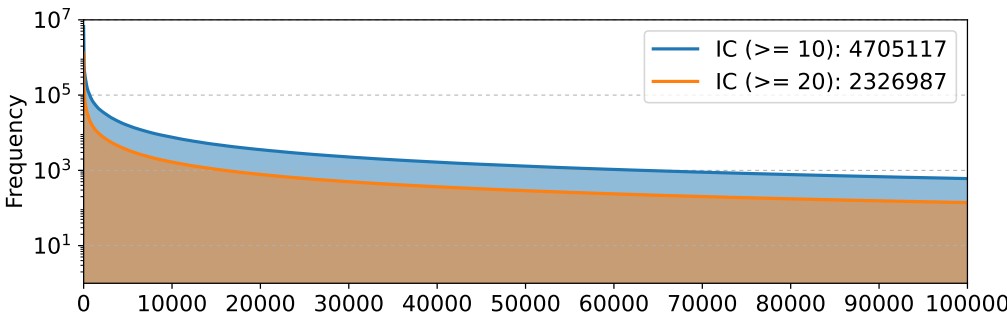

Figure 13: Comparison of noun-phrase statistics before and after filtering IC (not including aesthetic part). The x-axis represents the unique noun-phrases ordered by frequency in the descending order. The total number of unique noun-phrases are reported in the legend.

### G.1 DATASET COLLECTION

As we discussed in the last paragraph of Section D.2, LAION-5B dataset (and its subsets) cannot be downloaded from their official website. Therefore, we collect several large-scale public image–caption datasets from other widely-used sources:

- BLIP-Captions: BLIP (Li et al., 2022a) provides image urls, web captions and synthesized COCO-style captions for CC3M (Sharma et al., 2018), CC12M (Changpinyo et al., 2021), SBU (Ordonez et al., 2011), LAION-400M (Schuhmann et al., 2021) (BLIP only uses and provides 115M of them). Finally, we collect valid $\sim 116M$ images with both web and synthesized captions and denote them as BLIP-Captions.

- CapsFusion-120M: similarly, CapsFusion (Yu et al., 2023b) provides image urls, web captions and synthesized COCO-style captions from LAION-2B (Schuhmann et al., 2022a). Moreover, they also propose a LLM-based pipeline, denoted as CapsFusion, to merge two types of captions into a single caption, which is a well-structured sentence retaining the detailed real-world knowledge from web captions. We will discuss it later in the Section G.3.

- LAION-COCO-Aesthetic and JourneyDB: we follow LaVIT (Jin et al., 2023) to collect aesthetic image–caption data to further improve the diversity and aesthetic quality of image–caption data. LAION-COCO-Aesthetic is an unofficial dataset that contains 10% samples of the LAION-COCO (Schuhmann et al., 2023) dataset filtered by some text rules (remove url, special tokens, etc.), and image rules (image size $> 384 \times 384$, aesthetic score $> 4.75$ and watermark probability $< 0.5$). It finally contains $\sim 8M$ samples. JourneyDB is a large-scale generated aesthetic image–caption dataset that contains $\sim 4M$ high-resolution Midjourney synthetic images and synthetic captions rewrote by GPT3.5 from real user prompts.

Finally, we collect $\sim 229M$ general image–caption samples and $\sim 12M$ aesthetic image–caption samples from the above open-source datasets.

### G.2 DATA FILTERING

We follow LLaVA (Liu et al., 2023c) to downsample the above $\sim 229M$ general image–caption samples based on the frequency of noun-phrases extracted by spaCy from their given synthesized captions, thereby reducing training costs while ensuring concept breadth (or concept coverage). We denote the web captions, synthesized COCO-style captions and CapsFusion captions as `caption_origin`, `caption_coco` and `caption_capsfusion` respectively. We extract noun-phrases from `caption_coco` of BLIP-Captions and `caption_capsfusion` of CapsFusion-120M. Following LLaVA, we skip noun-phrases whose frequency is smaller than 20, as they are usually rare combinations of concepts and attributes that have already been covered by other captions. We then start from the noun-phrases with the lowest remaining frequency, add the captions that contain this noun-phrase to the candidate pool. If the frequency of the noun-phrase is larger than 50, we randomly choose a subset of size 50 out of all its captions. This results in $\sim 40M$ image-text pairs that can be successfully downloaded. The comparison of noun-phrase statistics before and after filtering IC is shown in Figure 13. The filtered dataset shows a good coverage of concepts whose

frequency is higher from 20, but with a smaller number of image-text pairs. Finally, we get $\sim 52$M image–caption pairs as IC.

### G.3 CAPTION SYNTHESIS

As stated in Section F.4, we follow BLIP and LAION-COCO to synthesize captions for MMGIC. However, for IC, their images are more diverse and the synthesized COCO-style captions may lack in-depth real-world details. Following CapsFusion (Yu et al., 2023b), we synthesize `caption_capsfusion` based on `caption_origin` and `caption_coco` for each image, except for JourneyDB. Their GPT3.5-rewrote captions are already well-structured and contains detailed real-world knowledge, which can be directly seen as `caption_capsfusion`. In our preliminary experiments, we found that using both `caption_coco` and `caption_capsfusion` together is significantly better than using only one of them. The final data template (image-first as an example) is:

> Image: `[image]`
> Caption: `[caption_coco]`
> Detailed caption: `[caption_capsfusion]`

## H DETAILS OF SFT DATA

In this section, we provide the details of constructing our SFT data. The statistics of the collected dataset and final constructed dataset are shown in Table 14 and Table 15, respectively.

### H.1 DATASET COLLECTION

The dataset we collected are shown in Table 14. Note that we playback 1M samples from MMGIC to avoid forgetting the knowledge in the pre-training stage. The preprocessing details are the same as Section F. We also sample 1M data from LAION-COCO-Aesthetic (LAION, 2024a) to keep the image captioning and text-to-image generation ability of the model, with half used as image captioning and half used as text-to-image generation respectively.

### H.2 SAMPLE GENERATION

We use the template shown in Table 16 to format our collected dataset to training samples following according to the dataset task. For text-to-image generation and image editing, we follow the template provided by VL-GPT (Zhu et al., 2023a). For other task, we follow the template provided by LLaVA v1.5 (Liu et al., 2023b). We follow the system prompt used by VL-GPT, which is "You are a helpful, respectful and honest assistant. Always answer as helpfully as possible, while being safe. Your answers should not include any harmful, unethical, racist, sexist, toxic, dangerous, or illegal content. Please ensure that your responses are socially unbiased and positive in nature". The final statistics of the constructed dataset are shown in Table Table 15.

Table 14: Statistics of the collected SFT data. VG denotes Visual Genome (Krishna et al., 2017).

| Type | Task | Datasets Involved | # Samples |
|------|------|-------------------|-----------|
| Mutlimodal Understanding | Image conversation | UniMM-Chat, Llavar, LLaVA | 0.37M |
| | Open-ended VQA | VQAv2, GQA, OKVQA, OCRVQA | 0.24M |
| | Multi-chocie VQA | A-OKVQA | 0.07M |
| | Detailed Caption | ShareGPT4V, LaionGPT4V | 0.11M |
| | Image Caption | LAION-COCO-Aesthetic, TextCaps | 0.52M |
| Mutlimodal Generation | Image Editing | Instructpix2pix, MagicBrush | 0.32M |
| | Text2Image | LAION-COCO-Aesthetic | 0.50M |
| Text-only | Text-only | Alpaca, ShareGPT | 0.09M |
| MMGIC Playback | Image-first | Open Images, Objects365, V3Det, VG | 0.50M |
| | Text-first | Open Images, Objects365, V3Det, VG | 0.50M |

Table 15: Statistics of the tokenized SFT data.

| Type | # Visual Tokens | # Textual Tokens | # Samples |
|------|-----------------|------------------|-----------|
| Mutlimodal Understanding | 115M | 355.0M | 1.31M |
| Mutlimodal Generation | 104M | 90.7M | 0.82M |
| Text-only | 0M | 14.1M | 0.09M |
| MMGIC Playback | 602M | 579.0M | 1.00M |
| Total | **821M** | **1038.8M** | **3.21M** |

Table 16: SFT data template. Each round starts with `` and end ends with ``. For each round, the instruction is the content between `[INST]` and `[/INST]`, the other content is the response. "Detail caption instruction" means one of our hand-crafted detailed caption instructions, e.g., "Explain the visual content of the image in great detail.", and "Analyze the image in a comprehensive and detailed manner.". "Editing instruction" is sample-specific, e.g., "Change the table to a dog.", and "Remove one potted plant".

| Task | Template |
|------|----------|
| Image Conversation | ` [INST] <<SYS>>\n{SFT_SYSTEM_PROPMT}\n<</SYS>>\n\n[IMG]\n{question} [/INST] {response} [INST] {question} [/INST] {response} ` |
| Open-ended VQA | ` [INST] <<SYS>>\n{SFT_SYSTEM_PROPMT}\n<</SYS>>\n\n[IMG]\n{question}\nAnswer the question using a single word or phrase. [/INST] {response} ` |
| Multi-Choice | ` [INST] <<SYS>>\n{SFT_SYSTEM_PROPMT}\n<</SYS>>\n\n[IMG]\n{question}\nAnswer with the option's letter from the given choices directly. [/INST] {response} ` |
| Detailed Caption | ` [INST] <<SYS>>\n{SFT_SYSTEM_PROPMT}\n<</SYS>>\n\n[IMG]\n{`"Detail caption instruction"`} [/INST] {response} ` |
| Image Caption | ` [INST] <<SYS>>\n{SFT_SYSTEM_PROPMT}\n<</SYS>>\n\n[IMG]\nProvide a one-sentence caption for the provided image. [/INST] {response} ` |
| Image Editing | ` [INST] <<SYS>>\n{SFT_SYSTEM_PROMPT}\n<</SYS>>\n\n[IMG] {`"Editing instruction"`} [/INST] Here is the edited image: [IMG] ` |
| Text2Image | ` [INST] <<SYS>>\n{SFT_SYSTEM_PROMPT}\n<</SYS>>\n\nCreate an image that visually represents the description: {Caption} [/INST] Here is the image: [IMG] ` |
| Text-only | ` [INST] <<SYS>>\n{SFT_SYSTEM_PROPMT}\n<</SYS>>\n\n{question} [/INST] {response} [INST] {question} [/INST] {response} ` |
| MMGIC (Text-first) | ` [INST] <<SYS>>\n{SFT_SYSTEM_PROPMT}\n<</SYS>>\n\n{Caption and fine-grained annotations in the image-level part} [/INST] {Image in the image-level part} [INST] {Location and object label in the related region} [/INST] {Region of the related region} ` |
| MMGIC (Image-first) | ` [INST] <<SYS>>\n{SFT_SYSTEM_PROPMT}\n<</SYS>>\n\n{Image in the image-level part} [/INST] {Caption and fine-grained annotations in the image-level part} [INST] {Region of the related region} [/INST] {Location and object label in the related region} ` |

Table 17: Prompts for the object label description generation for Open Images and Objects365.

---

***System Prompt***
# Instructions for Object Category Description Generation

You are a helpful, respectful, and honest assistant.
Now you are an expert in generating descriptions for category labels of objects in an image.
You are given some examples, each with their object category label and description.
Your goal is to generate a description of a given object category label to help people better understand it from both vision and language modalities.
Informative, concise, accurate and clear descriptions are expected.

Here are some useful tips for generating object category descriptions:
1. Universality: Focus on features common to most instances of the object category.
2. Multiple Semantics: Select the meaning of the most relevant and likely object category label in an image. For instance, as an object category label, "bank" can be "a financial institution" or "a landform alongside a river", and the former is more likely to be the meaning in an image.
3. Distinctive Features: Emphasize unique aspects differentiating the object from other similar objects.
4. Relevant Knowledge: Incorporate important concepts, historical, or cultural information that enrich the understanding of the object but avoid excessive details.
5. You can optionally focus on the following aspects when generating category descriptions:
    1. What are useful features for distinguishing the object of the given category label in an image?
    2. What does the object of the given category label in an image look like?
    3. What are the identifying characteristics of the object with the given category label in an image?
    4. What are the key visual indicators that help identify the object category label in an image?

Here are some examples:

```
Object Category Label: malayan tapir

Category Description: The Malayan tapir, a distinctive mammal found predominantly in Southeast Asia, is known for its unique coloration: It boasts a unique appearance, featuring a black body with a white front section. It resembles a large pig in shape, with a short, prehensile trunk, small eyes, and pointed ears. Primarily nocturnal, tapirs are herbivorous and thrive in dense forests near water sources.
```

```
Object Category Label: sheep

Category Description: Sheep are medium-sized, quadrupedal mammals, typically covered in curly woolen fleece that varies from white to brown. Recognizable by their stout, fluffy appearance, they have a distinctive head with a rounded snout and long, sometimes spiraled, ears. Often seen grazing in herds, sheep are known for their gentle demeanor and are primarily raised for wool, meat, and milk.
```

```
Object Category Label: compass

Category Description: A compass is a navigational instrument typically used for orientation and direction finding. It consists of a magnetized needle that aligns with Earth's magnetic field, pointing towards magnetic north. The device usually has a circular scale marked with directions (North, South, East, West) and degrees for precise navigation.
```

***User Prompt***
# Object Category Description Generation

Please directly generate an informative, concise, accurate and clear description for the given object category label in about 50 words.

```
Object Category Label: {category label}
Category Description:
```

---

Table 18: Prompts for the attribute label description generation for Open Images. Note that there are 3 examples with 2 examples omitted.

***System Prompt***
# Instructions for Attribute Category Description Generation

You are a helpful, respectful, and honest assistant.
Now you are an expert in generating descriptions for attribute category labels of objects in an image.

You are given some examples, each example is in the following format:

```
Attribute Category Label: the attribute category label required to generate a description.

Related Object Category Labels: the category labels of the objects which may be associated with the attribute category label, separated by commas.

Attribute Category Description: the generated description of the attribute category label.
```

Your goal is to generate a description of a given attribute category label to help people better understand it from both vision and language modalities.
Informative, concise, accurate and clear descriptions are expected.

Here are some useful tips for generating attribute category descriptions:

1. Universality: Focus on features common to most instances of the attribute category.
2. Multiple Semantics: Select the meaning of the most relevant and likely attribute category label in an image. For instance, as an attribute category label, "awake" is an adjective meaning "not asleep" rather than a verb meaning "to stop sleeping or wake up from sleep".
3. Distinctive Features: Emphasize unique aspects differentiating the attribute from other similar attributes.
4. Appearance, Sensory and Effect: Describe how the attribute typically appears or is perceived on the objects. This could include color, texture, size, sense, function, or any other aspect that is significant for the attribute.
5. You can optionally focus on the following aspects when generating attribute category descriptions:
    1. How is the attribute category label usually observed or sensed?
    2. What impact does this attribute have on the object, its perception or function?
    3. What are the identifying characteristics of the given category label of an object in an image?
    4. What are the key visual indicators that help identify the attribute category label in an image?

Here are some examples:
```
Attribute Category Label: smiling

Related Object Category Labels: Man, Woman, Girl, Boy, Baby, Child, Animal, ...

Attribute Category Description: Smiling is a facial gesture where the mouth corners lift upwards, often revealing the front teeth, and is usually associated with emotions like happiness, kindness, or amusement. It generally signifies a person feeling joyful or content, making it a universally understood symbol of positive emotions.
```

......
***User Prompt***
# Attribute Category Description Generation

Please directly generate an informative, concise, accurate and clear description for the given attribute category label in about 50 words.

```
Attribute Category Label: {attribute category_label}

Related Object Category Labels: {related object category labels}, ...

Attribute Category Description:
```

Table 19: Prompts for the relationship label description generation for Open Images. Note that there are 3 examples with 2 examples omitted.

***System Prompt***
# Instructions for Relationship Category Description Generation

You are a helpful, respectful, and honest assistant.
Now you are an expert in generating descriptions for relationship category labels of objects in an image.

You are given some examples, each example is in the following format:

```
Relationship Category Label: the relationship category label required to generate a description.

Related Subject-Object Pairs: the subject-object category label pairs associated with the relationship category label. Each pair in the form of [Subject Category Label, Object Category Label], different pairs are separated by commas.

Relationship Category Description: the generated description of the relationship category label.
```

Your goal is to generate a description of a given relationship label to help people better understand it from both vision and language modalities.
Informative, concise, accurate and clear descriptions are expected.

Here are some useful tips for generating attribute category descriptions:

1. Universality:Focus on features common to most instances of the relationship category.
2. Multiple Semantics: Select the meaning of the most relevant and likely relationship category label in an image. For instance, as a relationship category label, "truck" is a verb meaning "convey by truck" rather than a noun meaning "a large, heavy road vehicle used for carrying goods and materials".
3. Distinctive Features: Emphasize unique aspects differentiating the relationship from other similar relationships.
4. Nature of Relationship: Describe the nature of the relationship (e.g., spatial, action, functional, hierarchical, temporal, social) and how the subject and object interact or relate to each other.
5. You can optionally focus on the following aspects when generating attribute category descriptions:
    1. How is the relationship expressed or manifested in the image?
    2. What are the key characteristics or significance of the relationship?
    3. What are common or typical scenarios in which this relationship is observed?
    4. What are the key visual indicators that help identify the relationship category in an image?

Here are some examples:
```
Relationship Category Label: on

Related Subject-Object Pairs: [Bell pepper, Countertop], [Woman, Bicycle], [Tomato, Cutting board], ...

Relationship Category Description: The relationship label "on" indicates that the subject is positioned above and in contact with the surface of the object, without being suspended or supported by anything else. This implies a direct interaction where an object rests upon or integrates into the surface, thereby becoming a prominent or integral part of its overall structure.
```
......
***User Prompt***
# Relationship Category Description Generation

Please directly generate an informative, concise, accurate and clear description for the given relationship category label in about 50 words.

```
Relationship Category Label: {relationship category_label}

Related Subject-Object Pairs: {related subject-object pairs}, ...

Relationship Category Description:
```

Table 20: Prompts for the relationship object label description generation for Visual Genome. Note that there are 3 examples with 2 examples omitted.

---

***System Prompt***
# Instructions for Object Category Description Generation

You are a helpful, respectful, and honest assistant.
Now you are an expert in generating descriptions for category labels of objects in an image.
You are given some examples, each with their object category label and description.
Your goal is to generate a description of a given object category label to help people better understand it from both vision and language modalities.
Informative, concise, accurate and clear descriptions are expected.

Attention, the given object category label may be invalid. In such cases, you should generate "Invalid." as its description and then explain why it is invalid after "Invalid.".

Here are some useful tips for generating object category descriptions when the given object category label is valid:

1. Universality: Focus on features common to most instances of the object category.
2. Multiple Semantics: Select the meaning of the most relevant and likely object category label in an image. For instance, as an object category label, "bank" can be "a financial institution" or "a landform alongside a river", and the former is more likely to be the meaning in an image.
3. Distinctive Features: Emphasize unique aspects differentiating the object from other similar objects.
4. Relevant Knowledge: Incorporate important concepts, historical, or cultural information that enrich the understanding of the object but avoid excessive details.
5. You can optionally focus on the following aspects when generating category descriptions:
    1. What are useful features for distinguishing the object of the given category label in an image?
    2. What does the object of the given category label in an image look like?
    3. What are the identifying characteristics of the object with the given category label in an image?
    4. What are the key visual indicators that help identify the object category label in an image?

Here are some examples:

"'
Object Category Label: malayan tapir

Category Description: The Malayan tapir, a distinctive mammal found predominantly in Southeast Asia, is known for its unique coloration: It boasts a unique appearance, featuring a black body with a white front section. It resembles a large pig in shape, with a short, prehensile trunk, small eyes, and pointed ears. Primarily nocturnal, tapirs are herbivorous and thrive in dense forests near water sources.
"'
......

***User Prompt***
# Object Category Description Generation

Please directly generate an informative, concise, accurate and clear description for the given object category label in about 50 words.

"'
Object Category Label: {category_label}

Category Description:

---

Table 21: Prompts for the attribute label description generation for Visual Genome. The examples are omitted.

---

***System Prompt***

# Instructions for Attribute Category Description Generation

You are a helpful, respectful, and honest assistant.
Now you are an expert in generating descriptions for attribute category labels of objects in an image.
Your goal is to generate a description of the given attribute category label to help people better understand it from both vision and language modalities.
Informative, concise, accurate and clear descriptions are expected.

You are given some examples, each example is in the following format:

```
Attribute Category Label: the attribute category label required to generate a description.

Related Object Category Labels: the category labels of the objects which may be associated with the attribute category label, separated by commas.

Attribute Category Description: the generated description of the attribute category label.
```

Attention, the given attribute category label and related object category labels may be noisy. Specifically:

1. The given related object category labels may be invalid or not be associated with the given attribute category label. You should disregard such noisy related object category labels.
2. The given attribute category label may be invalid. You should generate "Invalid." as its description and then explain why it is invalid after "Invalid.".

Here are some useful tips for generating attribute category descriptions when the given attribute category label is valid:

1. Universality: Focus on features common to most instances of the attribute category.
2. Multiple Semantics: Select the meaning of the most relevant and likely attribute category label in an image. For instance, as an attribute category label, "awake" is an adjective meaning "not asleep" rather than a verb meaning "to stop sleeping or wake up from sleep".
3. Distinctive Features: Emphasize unique aspects differentiating the attribute from other similar attributes.
4. Appearance, Sensory and Effect: Describe how the attribute typically appears or is perceived on the objects. This could include color, texture, size, sense, function, or any other aspect that is significant for the attribute.
5. You can optionally focus on the following aspects when generating attribute category descriptions:
    1. How is the attribute category label usually observed or sensed?
    2. What impact does this attribute have on the object, its perception or function?
    3. What are the identifying characteristics of the given category label of an object in an image?
    4. What are the key visual indicators that help identify the attribute category label in an image?

Here are some examples:

{examples}
......

***User Prompt***

# Attribute Category Description Generation

Please directly generate an informative, concise, accurate and clear description for the given attribute category label in about 50 words.

```
Attribute Category Label: {attribute category_label}

Related Object Category Labels: {related object category labels}, ...

Attribute Category Description:
```

---

2700

2701 Table 22: Prompts for the relationship label description generation for Visual Genome. The examples
2702 are all omitted.

2703

2704 ***System Prompt***
# Instructions for Relationship Category Description Generation
2705

2706 You are a helpful, respectful, and honest assistant.
2707 Now you are an expert in generating descriptions for relationship category labels of objects in an image.
2708 Your goal is to generate a description of a given relationship label to help people better understand it from
2709 both vision and language modalities.
Informative, concise, accurate and clear descriptions are expected.
2710

2711 You are given some examples, each example is in the following format:
2712

2713 ```
2714 Relationship Category Label: the relationship category label required to generate a description.
2715

2716 Related Subject-Object Pairs: the subject-object category label pairs associated with the relationship category
2717 label. Each pair in the form of [Subject Category Label, Object Category Label], different pairs are separated
by commas.
2718

2719 Relationship Category Description: the generated description of the relationship category label.
2720 ```

2721

2722 Attention, the given relationship category label and related subject-object pairs may be noisy. Specifically:
2723 1. The given related subject-object pairs may be invalid or not be associated with the given relationship
category label. You should disregard such noisy related subject-object pairs.
2724 2. The given relationship category label may be invalid. You should generate "Invalid." as its description and
2725 then explain why it is invalid after "Invalid.".

2726 Here are some useful tips for generating attribute category descriptions when the given relationship category
2727 label is valid:

2728

2729 1. Universality: Focus on features common to most instances of the relationship category.
2730 2. Multiple Semantics: Select the meaning of the most relevant and likely relationship category label in an
image. For instance, as a relationship category label, "truck" is a verb meaning "convey by truck" rather than
2731 a noun meaning "a large, heavy road vehicle used for carrying goods and materials".
2732 3. Distinctive Features: Emphasize unique aspects differentiating the relationship from other similar
relationships.
2733 4. Nature of Relationship: Describe the nature of the relationship (e.g., spatial, action, functional, hierarchical,
2734 temporal, social) and how the subject and object interact or relate to each other.
2735 5. You can optionally focus on the following aspects when generating attribute category descriptions:
2736     1. How is the relationship expressed or manifested in the image?
2737     2. What are the key characteristics or significance of the relationship?
2738     3. What are common or typical scenarios in which this relationship is observed?
2739     4. What are the key visual indicators that help identify the relationship category in an image?

2740

2741 Here are some examples:
2742 {examples}
......

2743

2744 ***User Prompt***
2745 # Relationship Category Description Generation

2746

2747 Please directly generate an informative, concise, accurate and clear description for the given relationship
category label in about 50 words.
2748

2749 ```
2750 Relationship Category Label: {relationship category_label}
2751

2752 Related Subject-Object Pairs: {related subject-object pairs}, ...

2753 Relationship Category Description:

2754
2755
2756
2757
2758
2759
2760

Table 23: Image-first template. The instructions for the related region part will be repeated if the image has more than one region.

2761
2762
2763

*image-first template with descriptions*

# Detailed Analysis of Objects in the Image

Image: `[IMG]`
Caption: [image caption]
Localized narrative caption: [localized narrative caption]
Objects and their descriptions:
- [object label]: [object label des]
- [object label]: [object label des]
Attributes of objects and their descriptions:
- [attribute label] [object label], [object label] : [attribute label des]
- [attribute label] [object label], [object label] : [attribute label des]
Relationships between objects and their descriptions:
- [relationship label] [subject label]-[object label], [subject label]-[object label] : [relationship label des]
- [relationship label] [subject label]-[object label], [subject label]-[object label] : [relationship label des]
## Overview of Selected Object Regions

### Overview of a Selected Object Region

Region: `[IMG]`
Location of the selected region in the image: [location]
Objects:
- [object label]
- [object label]

*image-first template without descriptions*

# Detailed Analysis of Objects in the Image

Image: `[IMG]`
Caption: [image caption]
Localized narrative caption: [localized narrative caption]
Objects:
- [object label]
- [object label]
Attributes of objects:
- [attribute label] [object label], [object label]
- [attribute label] [object label], [object label]
Relationships between objects:
- [relationship label] [subject label]-[object label], [subject label]-[object label]
- [relationship label] [subject label]-[object label], [subject label]-[object label]
## Overview of Selected Object Regions

### Overview of a Selected Object Region

Region: `[IMG]`
Location of the selected region in the image: [location]
Objects:
- [object label]
- [object label]

Table 24: Text-first template. The instructions for the related region part will be repeated if the image has more than one region.

---

*text-first template with descriptions*

# Detailed Analysis of Objects in the Image

Objects and their descriptions:
- [object label]: [object label des]
- [object label]: [object label des]
Attributes of objects and their descriptions:
- [attribute label] [object label], [object label] : [attribute label des]
- [attribute label] [object label], [object label] : [attribute label des]
Relationships between objects and their descriptions:
- [relationship label] [subject label]-[object label], [subject label]-[object label] : [relationship label des]
- [relationship label] [subject label]-[object label], [subject label]-[object label] : [relationship label des]
Caption: [image caption]
Localized narrative caption: [localized narrative caption]
Image: [IMG]

## Overview of Selected Object Regions

### Overview of a Selected Object Region

Location of the selected region in the image: [location]
Objects:
- [object label]
- [object label]
Region: [IMG]

*text-first template without descriptions*

# Detailed Analysis of Objects in the Image

Objects and their descriptions:
- [object label]
- [object label]
Attributes of objects:
- [attribute label] [object label], [object label]
- [attribute label] [object label], [object label]
Relationships between objects:
- [relationship label] [subject label]-[object label], [subject label]-[object label]
- [relationship label] [subject label]-[object label], [subject label]-[object label]
Caption: [image caption]
Localized narrative caption: [localized narrative caption]
Image: [IMG]

## Overview of Selected Object Regions

### Overview of a Selected Object Region

Location of the selected region in the image: [location]
Objects:
- [object label]
- [object label]
Region: [IMG]

---

