# OpenReview forum: "Exploring Multi-Grained Concept Annotations for Multimodal Large Language Models"
_ICLR.cc/2025/Conference — Submitted to ICLR 2025_

### Official Review · Reviewer_VYKS · 2024-10-29

**Soundness:** 3
**Presentation:** 3
**Contribution:** 2
**Rating:** 5
**Confidence:** 4

**Summary:**

The paper introduces the MMGIC dataset with multi-grained concept annotations, significantly improving model understanding and generation abilities, and demonstrates its effectiveness in zero-shot image captioning and generation tasks. It details the data filtering and synthesis process to enhance data quality.

**Strengths:**

1. The introduction of the MMGIC dataset with multi-grained concept annotations is a novel approach that significantly enhances model understanding and generation capabilities, providing richer and more detailed annotation information compared to traditional datasets.
2. The paper details an effective data filtering and synthesis process from multiple public datasets, ensuring high data quality and diversity, which in turn improves model training efficiency and effectiveness.

**Weaknesses:**

1. This paper emphasizes the introduction of a novel dataset. However, the process of image collection and dataset creation is relatively straightforward and lacks significant innovation.
2. The paper mentions several new features of the dataset, but these have not been thoroughly validated or discussed in either theoretical or subsequent practical experiments. Specifically, the proposed label description, which involves designing prompts to generate detailed descriptions via LLMs, is general in nature and unrelated to the images. This could potentially lead to hallucinations during model training.
3. Although this paper is focused on the dataset, a significant portion is dedicated to describing the model architecture. Additionally, the experimental results lack comparative references from other studies, making it difficult to verify the dataset's effectiveness.

**Questions:**

1. Shouldn't you compare your method with others to demonstrate the superiority of your dataset?

2. Compared to others' methods of creating datasets, what are the advantages of your approach?

[1] Chen, Lin, Jinsong Li, Xiaoyi Dong, Pan Zhang, Conghui He, Jiaqi Wang, Feng Zhao, and Dahua Lin. "Sharegpt4v: Improving large multi-modal models with better captions." In: ECCV (2024).

[2] Urbanek, Jack, Florian Bordes, Pietro Astolfi, Mary Williamson, Vasu Sharma, and Adriana Romero-Soriano. "A picture is worth more than 77 text tokens: Evaluating clip-style models on dense captions."  In: CVPR (2024).

---

> ### Author Response · Authors · 2024-11-18
> **Responses for Reviewer VYKS (Part 1/3)**
>
> Thanks for your time, thorough reviews and constructive feedback. We address your concerns as follows.
>
> > Q1: The process of image collection and dataset creation is straightforward and lacks innovation.
>
> - Thanks for your concerns about the innovation of the creation process of MMGiC dataset.
> - **Research Focus**: As mentioned in the title, introduction, and throughout the paper, the research goal of this paper is to **explore** the potential of multi-grained concept annotations in MLLM. Considering the **lack** of such datasets in the field, the primary difficulty of this paper is to construct a large-scale dataset with multi-grained concept annotations for MLLM. This paper **does not overly emphasize** the innovation or novelty of the creation process of MMGiC dataset, as this is **not** our core focus:
>   - This paper collects existing datasets and designs **reasonable** and **effective** data processing and annotation synthesis processes to ensure the **quality** and **diversity** of multi-grained annotations.
>   - Then, this paper designs structured templates to integrate multimodal multi-grained concept annotations into image-text interleaved documents. This helps MLLMs learn annotations across multiple granularities **simultaneously** by leveraging the complex context processing capabilities of MLLMs, and makes MMGiC dataset **applicable** to different MLLM frameworks, without having to add additional components and loss functions to leverage multi-grained annotations.
> - Therefore, the core of the creation process of MMGiC dataset lies in ensuring the **quality**, **diversity**, and **applicability** of multi-grained annotations, which can **fills** the gap in the **lack** of multi-grained concept annotations for MLLMs, and helps this paper to **explore** the potential of multi-grained concept annotations in MLLM in Sec.4, which is also the core contribution of this paper.
>
> > Q2: Lack of validations or discussions on new features of the dataset.
>
> - Thanks for your concerns about the validation and discussion of the new features of MMGiC dataset.
> - The research goal of this paper is to explore the potential of multi-grained concept annotations in MLLM, so this paper conducts experimental validation and discussion of the new features of the dataset around this goal in Sec.4 and App.C.
> - In Sec.4, we conduct detailed validation of MMGiC's new feature, **multi-grained concept annotations**.
>   - **Effectiveness of Multi-grained Annotations**: Sec.4.1 validates the experimental effects of four different annotation combinations on image captioning and image generation tasks, and qualitative analysis based on specific examples, and further conducts experiments in the pre-training and SFT stages in Sec.4.2 and 4.3, demonstrating the potential of multi-grained concept annotations in MLLMs.
>   - **Comparison and Collaboration with IC**: By comparing and collaborating MMGiC with large-scale coarse-grained image-caption dataset IC, we validate and discuss their respective advantages in the **depth** and **breadth** of concept understanding, and further performance improvements brought by their appropriate combination.
>   - **Improvements in Different Dimensions and Inference Stage**: In Sec.4.4, we further focus on SEED-Bench-IMG, a multimodal understanding benchmark, and conduct qualitative and quantitative analysis of the impact of coarse-grained, fine-grained, and multi-grained concept annotations on different understanding dimensions. In App.C.3, we further validate and analyze the performance improvement brought by **automatic-generated** multi-grained concept annotations for images in the evaluation samples during the inference stage.
> - In the appendix,
>   - **Data Quality**: In App.C.1, we conduct statistical analysis of the **data quality** of MMGiC from the perspective of dataset statistics and concept overlap (K-LITE), and demonstrate that compared to traditional coarse-grained image-caption dataset IC, MMGiC has higher quality (more unique noun chunks per sample).
>   - **Image-Text Interleaved Format**: Based on structured templates, multi-grained concept annotations are integrated into image-text interleaved documents. We first demonstrate and analyze the impact of MMGiC dataset on the performance of image editing tasks in App.C.2, and then compare and analyze MMGiC with the widely used image-text interleaved dataset MMC4 in multimodal generation tasks in App.C.6.
>   - This paper also provides discussions on automated annotation synthesis (in App.B.1 and D.3), the **applicability** of MMGiC to different MLLM frameworks (in App.D.2), and comparison of MMGiC and existing image-text interleaved datasets (in App.D.4).
>
> [1] K-LITE: Learning Transferable Visual Models with External Knowledge.

---

> ### Author Response · Authors · 2024-11-18
> **Responses for Reviewer VYKS (Part 2/3)**
>
> > Q3: Hallucinations introduced by image-independent generation of label descriptions.
>
> - Thanks for your concerns about the label description generation in MMGiC dataset.
> - We fully agree with your concerns about the potential hallucinations introduced by the general image-independent generation of label descriptions, which is also an issue that this paper has **particularly** considered when synthesizing annotations for MMGiC dataset.
>   - **Effectiveness of Synthesized Label Descriptions**: As mentioned in Sec.2.2 #135-144, many previous works use WordNet and LLM to obtain general image-independent label descriptions, and successfully **improve** the model's understanding of concepts corresponding to these labels. The experimental results in Tab.1 of Sec.4.1 also show that label descriptions can bring **significant** performance improvements.
>   - **Feasible Synthesis for Common Concepts**: Whether in previous works ([1]) or in the four open-source datasets used in this paper, since the target labels are **common** concepts, LLM can generate label descriptions with good **generality** and rich **visual information** even without seeing the image.
>   - **Disambiguate Polysemous Labels**: It is worth noting that this paper also uses WordNet to **disambiguate** the **polysemous** labels in MMGiC dataset, such as "batter"->"batter (ballplayer)" or "batter (cooking)", to further reduce the ambiguity in the label descriptions, improve the **generality** and **quality**.
>   - **Careful Quality Control**: This paper not only carefully **manual-check** all generated label descriptions to ensure their quality and avoid introducing additional noise or hallucinations, but also carefully designs prompt templates and manual annotation examples (App.F.7) to guide and improve the quality of the annotations generated by LLM. Interestingly, this paper also finds that the powerful GPT-4 can even automatically identify **invalid** noisy labels in the labels with the customized prompt templates, helping us to clean up these invalid annotations.
> - We **have** added a discussion (App.D.5) about the label description generation and hallucination issues in the paper based on our insightful discussions.
>
> [1] Visual Classification via Description from Large Language Models
>
> > Q4: Concerns about the paper structure and lack comparison with other studies to demonstrate the superiority of your dataset?
>
> - Thanks for your concerns about the paper structure and the lack of comparison with other studies.
> - We would like to re-emphasize that the research focus of this paper is to **explore** the potential of multi-grained concept annotations in MLLM. The construction of MMGiC dataset (Sec.2 1.5 pages) and the design of the model architecture (Sec.3 1.5 pages) are both to **fill** the gap in such dataset, and to construct a **general** MLLM framework and a **comparable** and **controllable** experimental setting. Based on this, we can explore the potential of MMGiC in multimodal understanding and multimodal generation (Sec.4 5 pages and App.C 6 pages).
>   - We have reduced the portion of the model architecture section and will further optimize based on your feedback.
> - As for the comparison with other studies:
>   - **Fair and Comparable Baselines**: As mentioned in Sec.4.3 #374-377, the row 0, 1, 2 in Tab.3&4 are three fair and comparable baselines constructed in this paper, the only difference is whether the coarse-grained IC or the multi-grained MMGiC is used. The comparison between the baselines can **fairly** and **reliably** show the potential of MMGiC dataset in MLLM.
>   - **Non-comparable Reference MLLMs**: The other MLLMs presented in gray font are **not comparable** with our baselines, as there are significant differences in training data, training settings, model frameworks, etc., and their computational costs and training data are **far beyond** the baselines in this paper (LoRA Tuning with <8% params vs Full-Param Tuning, even 10 times training data), so they are presented for reference only.
>   - **Effectiveness of MMGiC**: Experimental results of three baselines corresponding to row 0, 1, 2 in Tab.3&4 not only demonstrate the effectiveness of MMGiC in MLLM in a fair and comparable experimental setting, but also achieve **comparable** performance with **far less** computational costs and training data than other MLLMs, and even achieve **better** performance on some metrics, which further shows the effectiveness of MMGiC dataset.
>   - Considering the incomparability, high cost, and difficulty of reproducibility of other MLLMs (see App.D.2), this paper, based on MMGiC dataset and a large-scale (52M) coarse-grained image-caption dataset constructed from widely used open-source datasets, conducts a series of **fair**, **comparable**, and **controllable** experiments on three corresponding baselines in the pre-training and SFT stages, demonstrating the effectiveness of MMGiC in MLLM, which is also the core contribution of this paper.

---

> ### Author Response · Authors · 2024-11-18
> **Responses for Reviewer VYKS (Part 3/3)**
>
> > Q5: Advantages of your approach compared to others' methods of creating datasets.
>
> - Thanks for your concerns about the advantages of our approach compared to others' methods of creating datasets.
> - Compared to the ShareGPT4V (0.1M dense captions automatically generated by GPT-4V) and DCI (7.8K manual-annotated dense captions) datasets you mentioned, the 4M-sized MMGiC dataset constructed for exploring the potential of multi-grained concept annotations in MLLM has the following advantages:
>   - **Multi-Grained Concept Annotations**: MMGiC dataset provides multi-grained concept annotations for each image, including coarse-grained image captions, fine-grained category labels, label annotations, and object region annotations, while the ShareGPT4V and DCI datasets rely on GPT-4V or manually annotated dense captions, which also help improve MLLMs' understanding of concepts, but do not provide explicit **fine-grained aligned annotations** like MMGiC dataset (as shown in Fig.1). This helps MLLMs better locate and learn concepts (Sec.4.1).
>   - **Multimodal Annotations** and **Image-Text Interleaved Documents**: Compared to the textual annotations in ShareGPT4V and DCI datasets, MMGiC dataset provides **multimodal** annotations in text form (image captions, category labels, label descriptions) and visual form (object regions) for each image, and designs structured templates to integrate these multimodal multi-grained concept annotations into **image-text interleaved documents**, enabling MLLMs to learn annotations across multiple granularities **simultaneously** by leveraging the complex context processing capabilities of MLLMs.
>   - **Quality and Hallucinations**:
>     - In Sec.4, the reasonable combination of MMGiC dataset with (noisy) IC dataset can further improve model performance. While in Tab.5 of VILA, when added ShareGPT4V to the SFT data, it can effectively improve the performance of some benchmarks with GPT-4 as the evaluator, but shows some **fluctuations** in performance on other benchmarks, such as TextVQA, MME, SEED-Bench, all show significant **declines**. This indicates that automatic-generated dense captions may introduce some hallucinations.
>     - To ensure the **reliability** of experimental exploration, this paper chooses to build the dataset based on open-source manual-annotated datasets. We design reasonable and effective data processing and synthesis processes to ensure the **quality** and **diversity** of the data, **avoiding** the introduction of additional noise and hallucinations.
>   - **Reproducibility**, **Open-Source**, **Generality**, and **Scalability**:
>     - This paper demonstrates the **feasibility** of MMGiC as pre-training data and directly as SFT data in Sec.4 and App.C.5. The data form of image-text interleaved documents also provides MMGiC with good **generality** (no need to add additional components and loss functions to leverage annotations of different granularities) to different MLLM frameworks.
>     - This paper also promise to open-source code of MMGiC dataset from data processing, annotation synthesis, data construction, and model training and evaluation, providing important resources for future research.
>     - As mentioned in the response to Reviewer `Xnsu`'s Q1, it is **feasible** to further **scale** up MMGiC dataset by building automated systems in the future.
>
> Hope our clarification can solve (parts of) your concerns, and we would greatly appreciate it if you can re-consider the rating.
>
> We will improve our paper based on your feedback and suggestions. Thanks again for your valuable comments.
>
> [1] VILA: On Pre-training for Visual Language Models

---

> ### Author Response · Authors · 2024-11-30
> **Kindly Reminder**
>
> Dear Reviewer VYKS,
>
> We hope this message finds you well.
>
> We understand that you have a busy schedule and greatly appreciate the time and effort you have dedicated to reviewing our work.
>
> As the extended discussion phase is about to close (<3 days), we wanted to kindly follow up to see if you might have any additional feedback or questions regarding our responses to your valuable comments. We are eager to ensure that all your concerns are addressed to your satisfaction.
>
> If there are any areas where you would like further clarification or additional details, please do not hesitate to let us know. We remain at your disposal to assist in any way.
>
> Thank you once again for your support and guidance throughout this process.
>
> Best,
>
> Authors

---

### Official Review · Reviewer_Xnsu · 2024-11-04

**Soundness:** 3
**Presentation:** 3
**Contribution:** 2
**Rating:** 5
**Confidence:** 3

**Summary:**

This research examines how MLLMs perform better when given two types of image information: broad descriptions (like overall image captions) and specific details (such as labeled objects and their locations). By using both levels of information together, MLLMs can better connect visual and linguistic elements, leading to improved performance on tasks that combine vision and language—whether that's describing images, answering questions about them, or creating images from text descriptions.

**Strengths:**

1- The researchers developed MMGIC, a new dataset that provides image annotations at multiple levels of detail. This fills an important need in the field of multimodal learning, where existing datasets often lack such comprehensive labeling.

2- The study's findings are backed by concrete test results, particularly on established benchmarks like POPE and SEED-Bench. By analyzing different combinations of annotation types, the researchers showed that using multiple levels of image descriptions significantly boosts how well MLLMs handle complex tasks that combine visual and language understanding.

**Weaknesses:**

1- Creating large datasets with detailed object-level annotations poses a major challenge, as it demands either extensive manual labor or sophisticated automated systems. While MMGIC makes valuable contributions, its reliance on existing public datasets creates two key limitations: it's difficult to expand the dataset size, and there's less room for automated data collection.

2- The study tests how different combinations of image annotations affect model performance. However, it falls short in explaining why certain combinations are more effective than others. This gap in analysis means we can see which approaches produced better results, but we don't fully understand the underlying reasons for their success.

**Questions:**

Address each of the weaknesses mentioned.

---

> ### Author Response · Authors · 2024-11-18
> **Responses for Reviewer Xnsu (Part 1/2)**
>
> Thanks for your time, thorough reviews and constructive feedback. We address your concerns as follows.
>
> > Q1: Difficult to scale up the dataset size and less room for automated data collection since the reliance on existing datasets.
>
> - Thanks for your concerns about the data scale and automated data collection of MMGiC dataset.
> - We fully agree with your concerns. Yes, MMGiC is a 4M-sized dataset constructed based on existing large-scale open-source human-annotated datasets, to explore the potential of multi-grained concept annotations in MLLM. It is indeed a huge challenge to further expand the data scale by manual annotation or sophisticated automated systems.
>   - **Feasibility of Automated Large-Scale Data Construction**: As discussed in App.B.1 #1092-1115, existing works such as All-Seeing and GLaMM have automatically constructed **large-scale** (even billion-level) open-source datasets with **detailed object-level** annotations to enhance the performance of MLLM on various object-level and grounding tasks. Similar to traditional VLM work, they leverage different types of annotations by adding additional components and loss functions, and achieve improvements on corresponding downstream tasks (object recognition and grounding benchmarks) that benefit from these annotations. Although we observe some hallucinations and noise in these datasets, the performance improvements achieved by these two datasets as preliminary valuable attempts to synthesize large-scale fine-grained annotations, demonstrate the **feasibility** of automated construction of such large-scale datasets.
>   - **Our Research Focus**: Different from above existing works in automated large-scale construction of fine-grained annotations and introduction of additional components and loss functions to leverage them, this paper, for the **first time**, explores the potential of multi-grained concept annotations in a general MLLM framework and a comparable and controllable experimental setting in both multimodal understanding and generation tasks. Furthermore, this paper is not limited to training MLLM with MMGiC dataset only, but explores the comparison and collaboration between MMGiC and large-scale coarse-grained image-caption dataset IC in both pre-training and SFT stages, to demonstrate the **potential** and **effectiveness** of MMGiC dataset. It is worth mentioning that we also demonstrate the **effectiveness** of directly using MMGiC as SFT data in App.C.5.
>   - **Complementary Efforts**: In summary, MMGiC and existing works in automated construction of large-scale fine-grained annotations are **complementary**, and it is **feasible** to further expand the scale of MMGiC dataset by building automated systems in the future. This paper also demonstrates the **effectiveness** of the collaboration between MMGiC and IC, and the **effectiveness** of directly using MMGiC as SFT data, which will provide important references for future work.
>
> [1] The All-Seeing Project: Towards Panoptic Visual Recognition and Understanding of the Open World.
>
> [2] GLaMM: Pixel Grounding Large Multimodal Model.

---

> ### Author Response · Authors · 2024-11-18
> **Responses for Reviewer Xnsu (Part 2/2)**
>
> > Q2: Fall short in explaining the underlying reasons for performance gaps between different combinations of image annotations.
>
> - Thanks for your attention to the underlying reasons for performance gaps.
> - Yes, we totally agree with your concerns. Therefore, this paper, in addition to the cold numbers in tables, through qualitative analysis and in-depth quantitative analysis in both pre-training and SFT stages, to some extent reveals the reasons for the performance differences brought by different combinations of multi-grained concept annotations.
> - **Pre-training Stage**: In Fig.3 and Fig.4 (left) of Sec.4.1, we focus on 3 examples of image captioning and image generation, exploring the reasons behind the performance changes brought by label descriptions and object regions in multi-grained concept annotations. We found that:
>   - (Textual) Label descriptions can help MLLMs better **distinguish** between "accordion" and "electronic keyboard" through **visual details** "pleated bellows and keyboard, box-like" and **related knowledge** "portable", thus improving the performance of MLLMs.
>   - (Visual) Object regions can better **collaborate** with other textual annotations and the complete original image, through the unified data format of image-text interleaved documents and autoregressive discrete training target. This helps MLLMs **ground** the textual annotations to **corresponding** image **regions** in the image, thus correctly **distinguishing** between "laying" and "sitting", and "on top of a toilet" and "on the floor next to a toilet", and significantly improving the performance of MLLMs, especially in "Instance Interaction, Spatial Relation".
> - **SFT Stage**: In Sec.4.4 and Fig.5, we focus on SEED-Bench-IMG, a multimodal understanding benchmark, and conduct qualitative and quantitative analysis, exploring the impact of coarse-grained (CG), fine-grained (FG), and multi-grained (MG) annotations on different understanding dimensions. We found that,
>   - Fig.5 Example $\boxed{1}$: compared to CG, FG helps MLLMs better **locate** and **learn** concepts through label descriptions and object regions, thus better **capturing** visual details and correctly **identifying** these concepts, and achieving better performance in dimensions such as "Instance Identity, Instance Interaction, Spatial Relation".
>   - Fig.5 Example $\boxed{2}$: compared to FG, CG helps MLLMs more **comprehensively** grasp the **holistic** understanding of the image through the global image caption, without focusing too much on local details to ignore the overall scene of the image.
>   - Fig.5 Example $\boxed{3}$: MG can combine the advantages of CG and FG, **integrating** the understanding of the overall scene and local details of the image, thus better understanding and **reasoning** the image, and achieving better performance in dimensions such as "Instance Interaction, Visual Reasoning".
>   - App.C.4 and Fig.9 also provide more qualitative analysis examples.
>   - In addition, to further analyze the reasons behind the performance impact of different annotation combinations on model performance, we **newly** supplement App.C.9 and Fig.10. By visualizing the attention maps of models trained with different annotation combinations, we want to more intuitively "**see**" the differences in their focused image tokens, to understand their differences in capturing local details and overall scenes of concepts.
> - **Inference Stage**: In App.C.3, we further explore in the inference stage, using baselines trained with MMGiC to **automatically generate** multi-grained concept annotations for the images in the evaluation samples. This help MLLMs better **understand** and **reason** the concepts in the images during evaluation, and achieving **further** performance improvements.
>
> Hope our clarification can solve (parts of) your concerns, and we would greatly appreciate it if you can re-consider the rating.
>
> We will improve our paper based on your feedback and suggestions. Thanks again for your valuable comments.

---

> ### Author Response · Authors · 2024-11-30
> **Kindly Reminder**
>
> Dear Reviewer Xnsu,
>
> We hope this message finds you well.
>
> We understand that you have a busy schedule and greatly appreciate the time and effort you have dedicated to reviewing our work.
>
> As the extended discussion phase is about to close (<3 days), we wanted to kindly follow up to see if you might have any additional feedback or questions regarding our responses to your valuable comments. We are eager to ensure that all your concerns are addressed to your satisfaction.
>
> If there are any areas where you would like further clarification or additional details, please do not hesitate to let us know. We remain at your disposal to assist in any way.
>
> Thank you once again for your support and guidance throughout this process.
>
> Best,
>
> Authors

---

### Official Review · Reviewer_TWWp · 2024-11-04

**Soundness:** 2
**Presentation:** 2
**Contribution:** 2
**Rating:** 5
**Confidence:** 4

**Summary:**

To integrate fine-grained concept annotations into Multimodal Large Language models (MLLMs) learning, this paper proposes a new dataset to explore the impact of different data recipes on multimodal comprehension and generation.  Extensive experiments demonstrate that multi-grained concepts do facilitate MLLMs to better locate and learn concepts, aligning vision and language at multiple granularities.

**Strengths:**

1. The proposed dataset with fine-grained annotations can be beneficial to the multimodal community.
2. Authors provided extensive ablation studies to compare different types of fine-grained annotations.
3. The proposed dataset demonstrates reasonably good results on image captioning, retrieval, and image generation tasks.

**Weaknesses:**

1. Considering the rich information of the fine-grained annotations, there seems to be marginal improvements comparing row 3 and row 0 in Table 1.

2. In Table 4, MLLM-MMGIC performs slightly better or even worse compared to the baseline model LaViT-v2-7B. It is a little bit unclear about the effectiveness of the proposed dataset.

3. Considering the large amount of images in IC dataset (i.e., 52M), the improvement of row 1 over row 2 is marginal.

**Questions:**

See the above weakness. The main concern is related to the performance on various tasks. The authors provide a large scale image-text dataset with fine-grained annotations. As this dataset is mainly collected from public object detection datasets, it should be naturally beneficial to VQA /GQA tasks. However, such improvements seem to be marginal on several tasks. For image captioning tasks, the model trained with the proposed dataset perform even worse compared to the baseline. It is difficult to evaluate the quality and effectiveness of the proposed dataset given such results.

---

> ### Author Response · Authors · 2024-11-18
> **Responses for Reviewer TWWp (Part 1/2)**
>
> Thanks for your time, thorough reviews and constructive feedback. We address your concerns as follows.
>
> > Q1: The rich information in fine-grained annotations seems to bring marginal improvements in Table 1 (row 3 vs. row 0).
>
> - Thanks for your attention to Tab.1.
> - **Evaluation Format Preference**:
>   - In fact, the image captioning and image generation tasks in Tab.1 have **natural** advantages for MLLMs trained with coarse-grained annotations, since the evaluation data format of these two tasks is completely **consistent** with the training data format of coarse-grained annotations, i.e., directly generating target captions or images based on given images or captions.
>   - As for the training data format based on multi-grained annotations, such as row 3, contains fine-grained concept annotations, which are significantly different from the evaluation data format. For example, in the image generation task, the training data format of row 3 is to generate target images based on given image captions and fine-grained concept annotations, which is more **complex** and **different**.
> - **Significant Improvements Under Adverse Conditions**:
>   - Even under such **adverse** evaluation conditions, compared to row 0, row 3 with fine-grained annotations has achieved significant improvements in the CIDEr metric of the image captioning task (**+4.66** and **+2.9**), the FID and CLIP-I metrics of the VIST dataset in the image generation task (**+32.28** and **+3.88**), and also achieved certain improvements in the CLIP-T and CLIP-I metrics of the MS-COCO-30K dataset (**+0.76** and **+0.62**), and is close to row 0 in FID (7.36 vs. 7.20).
>   - This **does not** mean that the rich information introduced by fine-grained annotations has not brought significant improvements. On the one hand, the evaluation data format is more favorable for row 0, and on the other hand, the MS-COCO-30K dataset is relatively **easy** and **saturated**, so the magnitude of improvement is not as large as in other datasets, but still has certain improvements.
>     - **Relatively Easy**: the evaluation data is **similar** in style to the images in our training data, all are COCO-style images of common concrete concepts
>     - **Saturated**: for example, the best reference model LaViT-v2-7B uses more than **200M** pre-training data to get 7.10 FID, 31.93 CLIP-T and 71.06 CLIP-V; our row 3 uses **only 4M** MMGiC as pre-training data to get 7.36 FID, 31.57 CLIP-T and 72.24 CLIP-V
>   - Overall, row 3 in Tab.1 has achieved **significant** performance improvements in most metrics compared to row 0.
>
> > Q2: MLLM-MMGiC performs slightly better or even worse compared to the baseline model LaVIT-v2-7B in Table 4.
>
> - Thanks for your attention to Tab.4.
> - **Non-comparable Reference MLLMs**:
>   - As stated in #374-377, the three rows 0, 1, 2 in Tab.4 are the three **fair** and **comparable** baselines constructed in this paper, the **only** difference is whether to use coarse-grained dataset IC or multi-grained dataset MMGiC.
>   - Existing MLLMs such as LaVIT presented in gray font cannot be regarded as a baseline in the experiments of this paper. They are **not comparable** to the baselines in this paper, with significant differences in training data, training settings, model frameworks, and their computational costs and training data are **far beyond** our baselines, hence they are presented for **reference only**.
> - **Detailed Differences between Baselines and LaVIT**: We have specifically discussed the differences between our three baselines and existing MLLMs in App.D.1. Taking LaVIT-v2-7B as an example, except for the differences in model frameworks,
>   - We **freeze** all parameters of visual modules, and most parameters of the LLM. Only a small part of the parameters (<**8%**) including the extended vocabulary and additional LoRA modules, are trained on the 4M MMGiC dataset and 52M IC dataset.
>   - While LaVIT trains **all** parameters, including the LLM and the visual modules, on more than **200M** image-caption pairs, which makes the computational cost and training data of LaVIT **far beyond** our baselines.
> - **Effectiveness of MMGiC Dataset**:
>   - Experimental results of the three baselines corresponding to rows 0, 1, 2 in Tab.4 demonstrate the effectiveness of the 4M MMGiC dataset by comparing the performance of MLLMs trained with MMGiC and IC, and the further performance improvement brought by the collaboration between MMGiC and IC.
>   - In addition, compared to existing MLLMs, our row 1 and row 2 using MMGiC dataset even with **far less** computational costs and training data, achieve **comparable** performance, and even achieve **better** performance on some metrics, which further shows the effectiveness of MMGiC dataset.

---

> ### Author Response · Authors · 2024-11-18
> **Responses for Reviewer TWWp (Part 2/2)**
>
> > Q3: The improvement of row 1 over row 2 bring by IC dataset (52M) is marginal.
>
> - Thanks for your attention to the impact of IC dataset on performance, I assume you mean that the performance improvement bring by IC dataset, i.e., row 2 over row 1 in Tab.3, is not significant.
> - Yes, we fully agree with your point that the performance improvement of IC dataset on **some** tasks is not significant considering its **scale** (52M).
>   - **With or Without IC**: When comparing row 2 and row 1, although row 2 achieves the best performance on the all benchmarks in Tab.3, the performance improvement on different datasets is different: **significant** improvements are achieved on datasets like COCO, NoCaps, VizWiz, MME and MMBench, but **only** 0.13 points on GQA.
>   - **With or Without MMGiC**: Similar phenomena exist when comparing row 2 and row 0: **significant** improvements are achieved on datasets like COCO, NoCaps, GQA, POPE and SEED-Bench, but **only** 0.04 points on VizWiz.
>   - **Respective Advantages**: As stated in #415-424, we attribute this to the fact that MMGiC and IC datasets have respective advantages in the **depth** and **breadth** of concept understanding. Therefore, when compare row 2 and row 1, even IC dataset contains 52M images, the performance improvement is not very significant on tasks that inspect **in-depth** understanding of common concrete concepts, but is significant on tasks that require a **broader** understanding of concepts.
>   - **Further Analysis**: In App.C.1, we further analyze this phenomenon from the perspective of dataset statistics and concept overlap (K-LITE). We found that although the concept overlap between the training data and the downstream evaluation dataset increases **significantly** with the increase of IC dataset (4M -> 52M), the performance is **still lower** than that of the MLLM trained with MMGiC dataset, which has a significantly lower concept overlap. We attribute this to **multi-grained** concept annotations and higher **image quality** of MMGiC. You can find more details and analysis in App.C.1.
>   - **Collaboration**: Most importantly, the collaboration between MMGiC and IC can achieve the **best** performance on all tasks, which further shows the importance of multi-grained concept annotations in MLLM.
> - Therefore, we believe that the **limitations** of the performance improvement of the IC dataset on some tasks, and the **further** performance improvement brought by the combination with MMGiC, further **highlight** the importance of the multi-grained concept annotation dataset MMGiC for future MLLM research.
>
> [1] K-LITE: Learning Transferable Visual Models with External Knowledge
>
> > Q4: Performance improvements seem to be marginal on several tasks. The model trained with MMGiC performs even worse compared to other MLLMs in image captioning tasks.
>
> - Thanks for your attention to task performance.
> - We assume that you are concerned that in Tab.3: (1) the performance improvement of MLLM-MMGiC (row 1) over MLLM-IC (row 0) is not significant on some tasks; (2) the performance of MLLM-MMGiC&IC compared to some existing MLLMs is not superior in the image captioning task.
> - For your concern (1),
>   - **Significant Improvements with Fewer Data**: When comparing our two baselines MLLM-IC and MLLM-MMGiC, as stated in #415-424 of this paper, compared to row 0 using **52M** IC data, row 1 using **only 4M** MMGiC data has achieved even **significantly** better results on many tasks (such as COCO, NoCaps, POPE, SEED-Bench), which demonstrates the **quality** and **effectiveness** of MMGiC dataset. In addition, in Tab.4, row 1 has achieved **significantly** better performance than row 0 on all 5 metrics of multimodal generation benchmarks.
>   - **Further Improvements with Collaboration**: Row 2 using both MMGiC and IC data has achieved the **best** performance on the all benchmarks in Tab.3, and even outperforms some existing MLLMs on some tasks, even though their **far higher** computational costs and training data.
> - For your concern (2),
>   - **Non-comparable Reference MLLMs**: As we replied in your Q2, existing MLLMs presented in gray font in Tab.3 are **not comparable** to the three baselines in this paper, with computational costs and training data **far beyond** our baselines.
>   - **Data Contamination in Reference MLLMs**: It is worth noting that in Tab.3, taking image captioning task as an example, VL-GPT-I and SEED-LLaMA-I, both **directly** use COCO Caption dataset in the training or SFT data, while for the **reliability** of evaluation, this paper **did not** use it in either the pre-training or SFT stages, which may be one of the reasons for the performance difference between SEED-LLaMA-I and our baselines on COCO and NoCaps.
>
> Hope our clarification can solve (parts of) your concerns, and we would greatly appreciate it if you can re-consider the rating.
>
> We will improve our paper based on your feedback and suggestions. Thanks again for your valuable comments.

---

> ### Author Response · Authors · 2024-11-30
> **Kindly Reminder**
>
> Dear Reviewer TWWp,
>
> We hope this message finds you well.
>
> We understand that you have a busy schedule and greatly appreciate the time and effort you have dedicated to reviewing our work.
>
> As the extended discussion phase is about to close (<3 days), we wanted to kindly follow up to see if you might have any additional feedback or questions regarding our responses to your valuable comments. We are eager to ensure that all your concerns are addressed to your satisfaction.
>
> If there are any areas where you would like further clarification or additional details, please do not hesitate to let us know. We remain at your disposal to assist in any way.
>
> Thank you once again for your support and guidance throughout this process.
>
> Best,
>
> Authors

---

### Official Review · Reviewer_cquC · 2024-11-04

**Soundness:** 3
**Presentation:** 3
**Contribution:** 2
**Rating:** 5
**Confidence:** 5

**Summary:**

The paper introduces a new dataset, Multimodal Multi-Grained Concept Annotations (MMGIC), aimed at enhancing Multimodal Large Language Models (MLLMs) in vision-language tasks by integrating both coarse-grained and fine-grained concept annotations. MMGIC is designed to address this limitation by including fine-grained annotations, such as object labels and object regions, alongside traditional captions, to better align vision and language across multiple granularities. The proposed approach leverages MMGIC within a general MLLM framework and evaluates its effectiveness on 12 multimodal comprehension and generation benchmarks.

**Strengths:**

The paper introduces MMGIC, a multimodal dataset with multi-grained concept annotations that integrates both coarse-grained (e.g., captions) and fine-grained (e.g., object labels, regions) annotations. This is a novel approach to enrich MLLMs, allowing them to achieve more nuanced vision-language alignment across different granularities.

**Weaknesses:**

**Writing and Clarity:** While I appreciate the authors’ efforts in extensively detailing the effectiveness of the proposed dataset, the writing is difficult to follow. There are too many unnecessary explanations, making the core claim of the paper unclear. The main focus seems to be on the dataset, yet the structure and writing style do not align well with a typical dataset paper, leading to a somewhat disorganized presentation.

**Dataset Quality:** As this is a dataset paper, it would be helpful to include a thorough evaluation of the dataset’s quality. Although large models like GPT are powerful to generate captions for local areas in the proposed dataset, they are known to suffer from hallucination issues. Could this dataset potentially exacerbate such problems, and how do you mitigate this risk?

**Untenable argument:** The importance of fine-grained understanding is well recognized. However, I question the point made in line 63 regarding the use of next-token prediction as a means to facilitate image understanding. Given the scale of the proposed dataset, a more effective loss function tailored to enhance comprehension might be a better choice than a simple autoregressive objective, which may not fully leverage the dataset’s potential.

**Questions:**

See weaknesses.

---

> ### Author Response · Authors · 2024-11-18
> **Responses for Reviewer cquC (Part 1/2)**
>
> Thanks for your time, thorough reviews and constructive feedback. We address your concerns as follows.
>
> > Q1: Writing and Clarity. The structure and writing style do not align well with a typical dataset paper.
>
> - Thanks for your comments. We apologize for the confusion caused by the structure and writing style of this paper.
> - **Research Focus**: As mentioned in the title, introduction, and throughout the paper, the research goal of this paper is to **explore** the potential of multi-grained concept annotations in MLLM, and the construction of the dataset is the **foundation** for this exploration.
> - **Paper Structure**:
>   - To this end, this paper briefly introduces how we construct MMGiC dataset in Sec.2 (1.5 pages) to fill the gap in such dataset, and briefly introduces the baseline framework and experimental settings we adopt in Sec.3 (1.5 pages).
>   - Then, based on the foundation of the first two sections, we detail how we explore different data recipes of multi-grained concept annotations in the pre-training and SFT stages in Sec.4 (5 pages), and compare, combine, and analyze the performance of MMGiC and IC (large-scale coarse-grained image-caption dataset) on 12 multimodal understanding and generation benchmarks to explore and demonstrate the potential of multi-grained concept annotations in MLLM.
>   - We also supplement more exploratory experiments and analysis in App.C (6 pages).
> - Therefore, this paper is not a dataset paper, but an **experimental exploration** paper that **explores** the **potential** of multi-grained concept annotations in improving MLLM's multimodal understanding and multimodal generation capabilities.
> - We **have** optimized Sec.2&3 based on your suggestions to make the paper more clear and organized, and we will further highlight the research focus of the paper based on your feedback.
>
> > Q2: Dataset Quality. Generated captions for local areas may introduce hallucination issues.
>
> - Thanks for your attention to the quality of the dataset.
> - **No Captions for Local Areas**: In fact, it is precisely because of the consideration of the hallucination problem in the generation of annotations by large models that this paper **did not** use large models like GPT to generate captions for **local areas**.
> - **Annotation Synthesis**: As shown in Fig.1 and Sec.2.2 of this paper, MMGiC dataset only includes the following two types of annotations generated by large models:
>   1. **Image Captions**: Based on the widely used image captioning pipeline (LAION COCO), BLIP-2 is used to generate captions for all images (Fig.1 $\boxed{1}$, **not** including local areas, more details in App.F.4), CLIP is used to select the best results in multiple rounds of generation, and then the data is manually spot-checked to ensure the quality of final generated captions;
>   2. **Label Descriptions**: GPT-4 is used to generate label descriptions for all category labels (Fig.1 $\boxed{2}$, more details in App.F.5), we improve the quality of GPT-4 generated annotations through human-written prompt templates and human-annotated examples, and ensure the quality of generated annotations through polysemous label disambiguation (manually with the help of WordNet) and manually check all generated annotations. We further discuss our efforts to avoid hallucination problems in label description generation in the Q3 of Reviewer `VYKS`.
> - In addition, the remaining fine-grained annotations other than image captions and label descriptions are derived from manual annotations of our collected dataset. We further discuss the efforts and explorations of annotation synthesis in MMGiC dataset in App.D.3, and supplement the data processing details we designed to improve the quality of human annotations in four public datasets in App.F.2.
> - We fully agree with your point that to **avoid** the impact of hallucinations in generated data on the exploration conclusions of multi-grained concept annotations, we have made every effort to **avoid** hallucination problems in the construction of MMGiC dataset, and ensure **data quality** in multiple ways.
>
> [1] LAION COCO: 600M synthetic captions from Laion2B-en
>
> [2] BLIP-2: Bootstrapping Language-Image Pre-training with Frozen Image Encoders and Large Language Models
>
> [3] CLIP: Contrastive Language-Image Pre-training

---

> ### Author Response · Authors · 2024-11-18
> **Responses for Reviewer cquC (Part 2/2)**
>
> > Q3: Untenable argument. The importance of fine-grained understanding is well recognized.
>
> - Thanks for your attention to the argument of this paper.
> - Yes, we fully agree with your point, and we also emphasize this point in #47-49 of Sec.1 and Sec.5.
> - We respectfully clarify that the core argument of this paper is **comprehensive and in-depth exploration** of MMGiC in MLLM.
>   - Whether in traditional VLM work or in the recent MLLM work, this paper, for the **first time**, explores and demonstrates the potential of multi-grained concept annotations in a general MLLM framework and a comparable and controllable experimental setting.
>   - Through comparison, combination and analysis of the performance of MMGiC and IC on 12 multimodal **understanding** and **generation** benchmarks during the pre-training and SFT stages, exploring and demonstrating the potential of multi-grained concept annotations in MLLM.
>   - This paper not only demonstrates the respective advantages of MMGiC and IC in the **depth** and **breadth** of concept representation in a fair and comparable experimental setting, but also shows that their appropriate combination can **further** improve performance, and provides detailed experiments and in-depth analysis in the main body and appendix.
>
> > Q4: Untenable argument. Question the use of next-token prediction as a means to facilitate image understanding. A tailored loss function for multi-grained annotations may be more effective than a simple autoregressive objective.
>
> - Thanks for your suggestion on the choice of loss function.
> - We fully agree with your point that in addition to next-token prediction, adding a tailored loss function for fine-grained concept annotations **may** more fully leverage the potential of MMGiC dataset (we also provided some relevant discussion on the details of loss functions in the end of App.E.1).
> - However, our core argument is **not** to explore what loss function can better leverage the potential of MMGiC, and the next-token prediction is also **not** sub-optimal, but **common and effective** for both image understanding and the use of fine-grained annotations.
>   - **Feasibility of Next-Token Prediction**:
>     - **Image Understanding**: Using next-token prediction as a learning target to promote image understanding is a **common practice** in VLM (OFA, UNIFIED-IO) and MLLM (SEED-LLaMA, Chameleon, Emu3) work, and this paper follows this general loss function. The experiments of previous work and this paper have shown the **effectiveness** of this loss function in promoting image understanding.
>     - **Leveraging Fine-Grained Annotations**: No matter in VLM (OFA, UNIFIED-IO, Pix2seq) or MLLM (Kosmos-2, Ferret, MM1.5), there have been many works leverage fine-grained annotations by directly using next-token prediction as a learning target, and have achieved good results in various downstream tasks.
>   - **Fair Comparison and Applicability**:
>     - As stated in #203-207, this paper **avoids** introducing additional components and loss functions (such as bounding box prediction), ensuring the **fair** comparison of baselines trained with MMGiC (image captions and fine-grained annotations) and IC (only image captions).
>     - This paper integrates multi-grained concept annotations into image-text interleaved documents through the structured template, so that MMGiC and IC can be explored in a fair and comparable way under a general MLLM framework and an autoregressive loss function, to ensures the **applicability** and **comparability** of the experimental results in different MLLM frameworks.
> - We hope to explore a tailored loss function for fine-grained concept annotations in future work to consider its impact on leveraging the potential of MMGiC. Interestingly, MM1.5 found that introducing referring and grounding task fine-tuning data (fine-grained annotations in the traditional VLM format) will **weaken** the performance of MLLM to some extent on general multimodal understanding tasks, which reminds us to pay attention to its impact on the performance of different types of tasks in future explorations.
>
> Hope our clarification can solve (parts of) your concerns, and we would greatly appreciate it if you can re-consider the rating.
>
> We will improve our paper based on your feedback and suggestions. Thanks again for your valuable comments.
>
> [1] OFA: Unifying Architectures, Tasks, and Modalities Through a Simple Sequence-to-Sequence Learning Framework
>
> [2] Unified-IO: A Unified Model for Vision, Language, and Multi-Modal Tasks
>
> [3] SEED-LLaMA: Making LLaMA SEE and Draw with SEED Tokenizer
>
> [4] Chameleon: Mixed-Modal Early-Fusion Foundation Models
>
> [5] Emu3: Next-Token Prediction is All You Need
>
> [6] Pix2seq: A Language Modeling Framework for Object Detection
>
> [7] Kosmos-2: Grounding Multimodal Large Language Models to the World
>
> [8] Ferret: Refer and Ground Anything Anywhere at Any Granularity
>
> [9] MM1.5: Methods, Analysis & Insights from Multimodal LLM Fine-tuning.

---

> ### Author Response · Authors · 2024-11-30
> **Kindly Reminder**
>
> Dear Reviewer cquC,
>
> We hope this message finds you well.
>
> We understand that you have a busy schedule and greatly appreciate the time and effort you have dedicated to reviewing our work.
>
> As the extended discussion phase is about to close (<3 days), we wanted to kindly follow up to see if you might have any additional feedback or questions regarding our responses to your valuable comments. We are eager to ensure that all your concerns are addressed to your satisfaction.
>
> If there are any areas where you would like further clarification or additional details, please do not hesitate to let us know. We remain at your disposal to assist in any way.
>
> Thank you once again for your support and guidance throughout this process.
>
> Best,
>
> Authors

---

### Author Response · Authors · 2024-11-18
**General Response for Reviewers**

Dear Reviewers, here, we hope to highlight the key conclusions of this paper to help you better understand our research focus and contribution.

The main focus of this paper is to **explore** the impact and potential of multi-grained concept annotations (**MMGIC**) on the multimodal understanding and generation capabilities of MLLM during the pre-training and SFT stages.
The construction of the dataset and model framework is a necessary **foundation** for our **exploration**.

The key conclusions of this paper include:

1. The multi-grained concept annotations provided by **MMGiC** can **complement** each other in image-text interleaved documents, and **improve** the multimodal **understanding** and **generation** capabilities of MLLM.
   - Sec.4.1, Pre-training Stage: To explore the impact of different annotation combinations in MMGiC on MLLM, we explore the performance of different data recipes. Based on the structured template and an autoregressive discrete MLLM framework, different annotations of MMGiC can be **integrated** into image-text interleaved documents and **facilitate** multimodal alignment across **multiple granularities simultaneously**, and **enhance** its multimodal understanding and generation capabilities.
   - Sec.4.4, SFT Stage: To further explore the impact of coarse-grained, fine-grained, and multi-grained concept annotations in MMGiC on MLLM, we **in-depth** analyze and explore different understanding dimensions and find that: fine-grained annotations can improve MLLM's ability to **recognize** and **locate** concepts, and **identify** relationships between concepts more accurately; coarse-grained annotations can improve MLLM's ability to grasp the **holistic** understanding of the image; and multi-grained annotations can **further** enhance MLLM's multimodal understanding capabilities through the **collaborative complementarity** of annotations at different granularities.

2. **MMGiC** and traditional large-scale coarse-grained image-caption data, **IC**, have their **respective** advantages in the **depth** and **breadth** of concept understanding, and can further **improve** the performance of MLLM through effective **collaboration**.
   - Sec.4.2, Pre-training Stage: To compare the impact of MMGiC and IC on the performance, the potential of their collaboration, and the way of their collaboration, we construct IC dataset based on open-source data to compare performance and explore different collaboration strategies. We find that MLLM trained with **only 4M** MMGiC significantly **outperforms** MLLM trained with **52M** IC in image captioning and image generation tasks; an appropriate curriculum learning strategy, i.e., first learning low-quality large-scale IC data and then **high-quality** MMGiC data, can help MLLM achieve **optimal** average performance.
   - Sec.4.3, SFT Stage: We fine-tune our three baselines (pre-trained with IC, MMGiC, and both) with SFT data, and explore their performance on **12** multimodal understanding and generation benchmarks. MLLMs trained with MMGiC and IC have their **respective** advantages in the **depth** and **breadth** of concept understanding, and their effective **collaboration** can further improve performance.

In addition, we not only provide qualitative analysis in Sec.4 (Fig.3, Fig.4, Fig.5), but also supplement **more** exploratory experiments and analysis in App.C (6 pages), including **concept overlap** analysis, results and analysis on **image editing** tasks, **self-generated** multi-grained concept annotations in the **inference** stage, the **effectiveness** of directly using MMGiC as SFT data, etc.

In our **newly** submitted version, based on your feedback and suggestions,
- we optimize the Sec.2&3 to make the paper more clear and further highlight the research focus of this paper based on the Q1 of Reviewer `cquC`.
- we add visualizations of the attention maps in App.C.9 and Fig.10 to more intuitively "**see**" the differences between models trained with different annotation combinations based on the Q2 of Reviewer `Xnsu`.
- we add a discussion (App.D.5) about our efforts to avoid hallucination problems in label description generation based on the Q3 of Reviewer `VYKS`.

---

### Meta-Review · Area_Chair_mpvS · 2024-12-22

**Metareview:**

This paper present a new dataset called MMGiC, aimed at improving the fine-grained image understanding for multimodla LLMs. The authors curated a number of object detection and scene graph datasets and convert the region annotations to training data for MLLMs. In addition to the dataset, the authors also proposed an autoregressive discrete MLLM framework which can not only generate textual tokens but also visual tokens for image decoding and generation.The authors evaluated the trained model for both multimodal understanding and generation, and showed superior performance to the model only trained with image caption data.

There are some strengths as mentioned by reviewers which the ACs also agreed. Concretely, the authors introduced a new dataset which contain both global image description and fine-grained regional annotaitons. The results for models trained on this dataset show superior performance to baselines. However, the ACs also share some concerns with the reviewers that: 1) this work lacks an in-depth analysis about how the different combination of dataset affects the final performance; 2)  After the SFT, the model pretrained with the proposed data does not show much improvements over the baseline (i.e., using MLLM-IC data for pretraining). especially on the understanding tasks. There is no clear explanation about this from the authors.

Due to limisted discussions between authors and reviewers, the ACs read the submission and all authors' responses. Overall, the ACs agreed that some of the comments from reviewers are subjective and made without careful understanding of the detailes. Ruling the out, in the meanwhile, many drawbacks pointed out by the reviewers are valid and align with the overall quality of this work. As such, the ACs think the overall quality of this work does not reach the bar of this venue, and suggest the authors further enhance the presentation and convey more detailed analysis to support the ambitious goal of improving both multimodal understanding and generation.

**Additional Comments On Reviewer Discussion:**

During the rebuttal period, there were no discussions between reviewers and authors. Later, reviewers raised their scores and the final average rating becomes 5.0, but still under the bar for this venue. Furthermore, reviewer cquC made a conclusive comment to appreciate the revision of the submission but still cast a lot concern about the novelty and contribution of this work to the community. Given this, the ACs spend time to read the submission and responses from reviewers and had a discussion on this submission.

After the discusssion, the ACs appreciated the effort from the authors to address concerns raised by reviewers and also the engagement during the rebuttal period. However, the ACs also agreed many points made by reviewers about the weakeness of the work, while being aware of some of them are not grounded and derived from misunderstanding on this submission. Take all these into account, the ACs think this work still has many weaknesses and require more work to polish the presentation and make the experiments more solid.

---

### Decision · Program_Chairs · 2025-01-22

Reject